# Optimization, Generalization and Differential Privacy Bounds for Gradient Descent on Kolmogorov–Arnold Networks

Puyu Wang[1]  Junyu Zhou[2]  Philipp Liznerski[1]  Marius Kloft[1]

## Abstract

Kolmogorov–Arnold Networks (KANs) have recently emerged as a structured alternative to standard MLPs, yet a principled theory for their training dynamics, generalization, and privacy properties remains limited. In this paper, we analyze gradient descent (GD) for training two-layer KANs and derive general bounds that characterize their training dynamics, generalization, and utility under differential privacy (DP). As a concrete instantiation, we specialize our analysis to logistic loss under an NTK-separable assumption, where we show that *polylogarithmic* network width suffices for GD to achieve an optimization rate of order $1/T$ and a generalization rate of order $1/n$, with $T$ denoting the number of GD iterations and $n$ the sample size. In the private setting, we characterize the noise required for $(\epsilon, \delta)$-DP and obtain a utility bound of order $\sqrt{d}/(n\epsilon)$ (with $d$ the input dimension), matching the classical lower bound for general convex Lipschitz problems. Our results imply that polylogarithmic width is not only sufficient but also *necessary* under differential privacy, revealing a qualitative gap between non-private (sufficiency only) and private (necessity also emerges) training regimes. Experiments further illustrate how these theoretical insights can guide practical choices, including network width selection and early stopping.

## 1. Introduction

Kolmogorov–Arnold Networks (KANs) (Liu et al., 2025b) have recently emerged as a structured alternative to standard multilayer perceptrons (MLPs), replacing fixed pointwise

[1]Department of Computer Science, RPTU Kaiserslautern-Landau, Kaiserslautern, Germany [2]Catholic University of Eichstätt-Ingolstadt, Ingolstadt, Germany. Correspondence to: Junyu Zhou <Junyu.Zhou@ku.de>.

*Proceedings of the 43rd International Conference on Machine Learning*, Seoul, South Korea. PMLR 306, 2026. Copyright 2026 by the author(s).

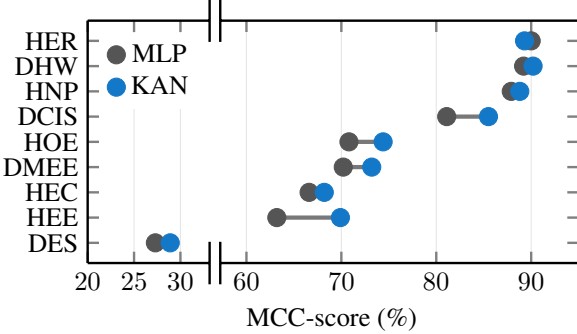

*Figure 1.* Comparison of MLP and KAN performance on genomic sequence classification. Each point pair corresponds to a benchmark dataset from Table 1 of Cherednichenko & Poptsova (2025), reporting Matthews correlation coefficient (higher is better) for a baseline MLP model and its KAN-based counterpart. Most KAN points lie to the right of their MLP counterparts, indicating improved predictive performance when replacing MLP modules with KAN layers.

nonlinearities with learnable univariate edge functions. This architectural bias has led to strong empirical performance in tasks where interactions can be effectively represented through univariate components. Such settings arise in scientific computing, physics-informed learning, time series forecasting, and molecular and biological modeling (Li et al., 2025a; Cherednichenko & Poptsova, 2025; Shukla et al., 2024; Patra et al., 2025; Wang et al., 2025d; Vaca-Rubio et al., 2024). In genomic sequence classification, for instance, replacing MLP modules with KAN layers has been shown to improve predictive performance on several benchmark datasets (see Figure 1).

At the same time, KANs raise new theoretical questions: their training dynamics depend on a large collection of coupled univariate function parameters, and standard analyses developed for MLPs do not directly apply. In particular, the interaction between width, optimization dynamics, and statistical generalization in KANs remains poorly understood.

Most existing theoretical analyses of KANs are algorithm-independent, focusing on expressivity and approximation-theoretic guarantees rather than the behavior of concrete training algorithms (Zhang & Zhou, 2025; Liu et al., 2025a; Li et al., 2025b; Wang et al., 2025c). As a result, they do not yield guarantees for the iterates produced by gradient

descent (GD), nor do they explain how optimization dynamics interact with generalization. A notable exception is Gao & Tan (2025), who show convergence of GD for two-layer KANs under strong positive-definiteness assumptions on the associated neural tangent kernel (NTK), which require network widths that scale polynomially in problem parameters. However, corresponding generalization guarantees for GD-trained KANs remain largely unexplored.

In addition to optimization and generalization, privacy considerations introduce an additional layer of complexity for training KANs. As models are increasingly trained on sensitive data in domains such as biology and medicine, it becomes essential to understand how privacy-preserving training procedures affect their performance. Differential privacy (DP) (Dwork et al., 2014) provides a rigorous framework for limiting the influence of individual data points, and gradient-based methods with additive noise, such as differentially private gradient descent (DP-GD), are standard tools for achieving such guarantees (Abadi et al., 2016; Song et al., 2013; Wang et al., 2022). However, existing theoretical analyses of private gradient methods for neural networks predominantly focus on MLPs (Romijnders & Koskela, 2025; Wang et al., 2025a; Shi et al., 2026).

Establishing theoretical guarantees for GD and its differentially private variant (DP-GD) on KANs is technically challenging. Along the optimization trajectory, the curvature of the loss depends sensitively on how far the parameters move from initialization, while DP-GD introduces stochastic perturbations at every iteration over a nonconvex landscape. To address these challenges, we develop a unified framework that combines self-consistent curvature control for GD with a trajectory-wise sensitivity and stability analysis of the projected DP-GD recursion. This framework enables guarantees for (DP-)GD optimization, generalization, and privacy utility in KAN training. Our main contributions can be summarized as follows.

- *Reference-point framework for GD and DP-GD.* We develop a unified reference-point analysis for training two-layer KANs with GD and DP-GD, yielding bounds on the optimization risk (training loss), the population risk, and the trajectory-averaged population risk under $(\epsilon, \delta)$-DP (Theorems 4.3, 4.11, and 4.15).
- *Optimization and fast-rate generalization.* Under NTK separability and logistic loss, we show that *polylogarithmic width suffices* for GD to achieve an optimization risk $\widetilde{\mathcal{O}}(1/T)$ (Theorem 4.9) and a fast-rate population risk bound $\widetilde{\mathcal{O}}(1/n)$ (Theorem 4.12).
- *Differential privacy and utility.* We provide an explicit Gaussian noise calibration ensuring $(\epsilon, \delta)$-DP for DP-GD on KANs and establish a privacy–utility tradeoff with an admissible width regime. Under NTK separability, this yields a utility bound $\widetilde{\mathcal{O}}(\sqrt{d}/(n\epsilon))$ for the trajectory-

averaged population risk, matching the classical convex Lipschitz lower bound (Theorem 4.16).
- *Sharp width characterization.* In the NTK-separable setting, we show that *polylogarithmic width* is not only sufficient but essentially necessary for DP-GD to attain the desired utility, revealing a qualitative gap between non-private GD and DP-GD.
- *Theory-guided practical implications, validated empirically.* Our theory provides guidance for selecting the width $m$ and the iteration number $T$. For GD, it predicts diminishing returns from increasing $m$ and $T$ beyond moderate values. In contrast, under a fixed privacy budget, it suggests that DP-GD benefits from moderate widths and early stopping. These qualitative trends are supported by the experiments in Section 5.

**Organization:** Section 2 introduces KANs and the model studied in this work. Sections 3 and 4 present our main results. Section 5 reports experimental findings. Related work and conclusion are given in Sections 6 and 7.

## 2. Kolmogorov–Arnold Networks: A Primer

This section provides background on KANs and introduces the concrete model studied in this work.

### 2.1. Motivation: Kolmogorov–Arnold Representation

Kolmogorov–Arnold type representation results (Kolmogorov, 1963; Arnol'd, 1957; Braun & Griebel, 2009) show that broad classes of multivariate functions admit representations as superpositions of univariate functions combined with simple aggregation operations. This perspective motivates neural architectures that place learnable univariate components on network edges, rather than relying on a fixed pointwise nonlinearity shared across units. KANs, introduced by Liu et al. (2025b), embody this idea by explicitly parameterizing edge functions, thereby inducing a structural bias that differs from standard MLPs.

Formally, a KAN is a feedforward network that maps an input vector $\mathbf{x} = (x_1, \ldots, x_{n_0}) \in \mathbb{R}^{n_0}$ to a sequence of hidden representations $\mathbf{x}_\ell = (x_{\ell,1}, \ldots, x_{\ell,n_\ell}) \in \mathbb{R}^{n_\ell}$ across layers $\ell = 0, \ldots, L$, with $\mathbf{x}_0 = \mathbf{x}$ and $n_\ell$ denoting the width of layer $\ell$. Each edge $(\ell, i) \to (\ell+1, j)$ carries a learnable univariate function $\phi_{\ell,j,i} : \mathbb{R} \to \mathbb{R}$, and the hidden units are computed as

$$x_{\ell+1,j} = \sum_{i=1}^{n_\ell} \phi_{\ell,j,i}(x_{\ell,i}), \qquad \ell = 0, \ldots, L-1. \quad (1)$$

In practice, the edge functions are parameterized using finite-dimensional function classes such as basis expansions. We defer the specific parameterization to Section 2.3.

## 2.2. Empirical Context and Applications

We briefly summarize empirical observations and application settings that motivate the study of KANs.

**Architectural inductive bias and interpretability.** KANs introduce an architectural inductive bias by replacing fixed pointwise activations with learnable univariate edge functions, which can be particularly effective when the target mapping exhibits low-dimensional structure, smoothness, or compositional interactions. This design aligns naturally with settings in which multivariate relationships admit structured decompositions into simpler components. In addition, the explicit parameterization of edge functions can support interpretability in practice, as individual components may be visualized and inspected directly (Erdmann et al., 2025; Ranasinghe et al., 2024).

**Application landscape.** Building on this architectural inductive bias, empirical studies report competitive performance of KAN-based models on a variety of structured learning tasks (Liu et al., 2025b; Somvanshi et al., 2025); see Figure 1 for an illustrative example in genomic sequence classification. In practice, KAN variants have often been explored as modular replacements for MLP components within larger architectures. Representative examples include convolutional KAN layers for vision tasks (Bodner et al., 2024; Ferdaus et al., 2024), temporal KAN designs for time-series forecasting (Genet & Inzirillo, 2024), and Transformer architectures in which the feedforward sub-network is replaced by KAN layers (Wang et al., 2025d). KAN components have also been combined with diffusion models in several recent works (Xiong et al., 2025; Su & Yang, 2025; Qiu et al., 2025).

## 2.3. Two-layer KANs with B-splines

We study a two-layer KAN with a single hidden layer of width $m$ and a scalar output, in which the univariate edge functions are parameterized using B-spline bases.

**Parameter representation.** For analysis, we collect all trainable coefficients into a single parameter vector. For each hidden unit $j \in [m]$, the first-layer spline coefficients $\{a_{i,j,k}\}_{i \in [d], k \in [p]}$ are arranged into a vector $\mathbf{a}_j \in \mathbb{R}^{dp}$ using a fixed ordering of the index pair $(i, k)$, and we set $\mathbf{a} = (\mathbf{a}_1, \ldots, \mathbf{a}_m) \in \mathbb{R}^{mdp}$. Similarly, the second-layer coefficients $\{c_{j,k}\}_{k \in [p]}$ are collected into vectors $\mathbf{c}_j \in \mathbb{R}^p$ and stacked as $\mathbf{c} = (\mathbf{c}_1, \ldots, \mathbf{c}_m) \in \mathbb{R}^{mp}$. The complete parameter vector is thus $\Theta = (\mathbf{a}, \mathbf{c}) \in \mathbb{R}^{mp(d+1)}$.

**Two-layer KAN model.** Each univariate edge function is parameterized using a B-spline basis $\{b_k\}_{k=1}^p$, and a bounded activation function $\sigma : \mathbb{R} \to \mathbb{R}$ (e.g., $\tanh$ or sigmoid) is applied after the first layer. The spline basis functions are defined on an interval covering all values

encountered during training and satisfy the boundedness conditions in Assumption 4.1. For an input $\mathbf{x}_0 = \mathbf{x} \in \mathbb{R}^d$, with input dimension $d$ and hidden width $m$, the two-layer KAN model is given by

$$f_\Theta(\mathbf{x}) = \frac{1}{\sqrt{m}} \sum_{j=1}^m \sum_{k=1}^p c_{j,k} \, b_k(x_{1,j}), \qquad (2)$$

where the hidden units take the form

$$x_{1,j} = \sigma\Big(\frac{1}{\sqrt{d}} \sum_{i=1}^d \sum_{k=1}^p a_{i,j,k} \, b_k(x_{0,i})\Big), \quad j \in [m].$$

This model closely follows the two-layer KAN of Gao & Tan (2025), adding an explicit $1/\sqrt{d}$ normalization in the first layer to stabilize pre-activations as $d$ grows and simplify the Hessian analysis. The $1/\sqrt{m}$ factor is the standard output normalization to control the scale as $m$ increases.

## 2.4. Toward a Theoretical Understanding

For the two-layer KAN model defined above, key theoretical questions remain open. In particular, it is unclear when gradient-based methods optimize the model reliably, how the resulting predictors generalize, and how these guarantees interact with privacy constraints. This motivates our theoretical study of optimization, generalization, and differential privacy for KANs.

## 3. Overview of Our Results

Our formal results are presented in Section 4 in a general form. In this section, we specialize to the logistic loss and an NTK-separable regime to highlight the main implications and intuitions. We use $\widetilde{\mathcal{O}}(\cdot)$ to suppress logarithmic factors and write $a \asymp b$ if $a$ and $b$ are of the same order.

### 3.1. Optimization Bound

We run GD with a constant step size $\eta > 0$ for $T$ iterations. The following theorem provides the convergence behavior of GD for KANs. For readability, we suppress the explicit dependence of the required width on problem parameters and defer these details to Section 4.

**Theorem 3.1** (Informal version of Theorem 4.9)**.** *Under the NTK separability assumption with margin $\gamma > 0$, if the network width is polylogarithmic, i.e., $m \geq \mathrm{polylog}(n, T)$, then with high probability (w.h.p.) over the random initialization, a two-layer KAN trained by GD achieves an optimization risk (training loss) at most*

$$\widetilde{\mathcal{O}}\Big(\frac{1}{\gamma^2 \eta T}\Big).$$

Remarkably, a *polylogarithmic* network width already *suffices* for GD to attain an $\mathcal{O}(1/T)$ optimization rate.

## 3.2. Generalization Bound

Building on our optimization guarantee, we establish the first algorithm-dependent generalization bound specifically for GD-trained KANs, achieving a fast $\mathcal{O}(1/n)$ rate up to the separability margin and logarithmic factors.

**Theorem 3.2** (Informal version of Theorem 4.12). *Under the NTK separability assumption with margin $\gamma > 0$, if the network width is polylogarithmic, and $\eta T \gtrsim n$, then w.h.p. over the random initialization, a two-layer KAN trained by GD achieves the expected population risk at most*

$$\widetilde{\mathcal{O}}\Big(\frac{1}{\gamma^4 n}\Big),$$

*where the expectation is taken with respect to (w.r.t.) the draw of the training data.*

Moreover, our result reveals an implicit regularization bias toward solutions that remain close to initialization. See Section 4.3 for further discussion.

## 3.3. Differentially Private Gradient Descent

Finally, we study DP-GD, where privacy imposes additional width constraints, and give a sharp utility bound.

**Theorem 3.3** (Informal version of Theorem 4.16). *Under the NTK separability assumption with margin $\gamma > 0$, for $\eta T \asymp \frac{\gamma^2 n\epsilon}{\sqrt{d}}$ and a polylogarithmic width $m \asymp \mathrm{polylog}(n)$, w.h.p. over the random initialization, the $(\epsilon, \delta)$-DP variant of GD algorithm (Algorithm 1) achieves an expected population risk, averaged over $T$ iterates, bounded by*

$$\widetilde{\mathcal{O}}\Big(\frac{\sqrt{d}}{\gamma^4 n\epsilon}\Big).$$

Our result also characterizes an admissible width regime with an essentially matching lower bound, showing that a polylogarithmic width is necessary for DP-GD to achieve the desired utility bound. See Section 4.4 for details.

## 3.4. Comparison to Prior Work

We refer to Theorems 4.9, 4.12 and 4.16 for the precise statements of our results and highlight here the main points of departure from existing theory.

**Optimization bounds.** The work most closely related to our optimization analysis is Gao & Tan (2025), which studies GD for two-layer B-spline KANs in regression. Under a positive-definiteness assumption on the expected NTK Gram matrix $G^\infty$ (i.e., $\lambda_{\min}(G^\infty) > 0$), they prove global linear convergence $(1 - \frac{\eta}{2}\lambda_{\min}(G^\infty))^T$ but require network widths that scale polynomially in the problem parameters. In contrast, we analyze GD for classification under an NTK-separability condition. As noted by Nitanda et al. (2019), this assumption is weaker than a Gram-matrix

positive-definiteness condition and can be satisfied in many cases due to the universality of the neural tangent models. This yields sublinear convergence $\widetilde{\mathcal{O}}(\frac{1}{\eta T \gamma^2})$ already in a polylogarithmic-width regime.

**Generalization bounds.** Most existing generalization results for KANs are *algorithm-independent* (Zhang & Zhou, 2025; Li et al., 2025b; Liu et al., 2025a). By comparison, we provide the *first algorithm-dependent* generalization bound $\widetilde{\mathcal{O}}(1/\gamma^4 n)$ for GD-trained KANs under NTK separability.

**Utility bounds for DP-GD.** To the best of our knowledge, this is the first utility analysis of DP gradient-based methods for KANs. A minimax lower bound tailored to KANs under DP is not currently available. We therefore compare to the classical lower bound for general convex Lipschitz problems (Bassily et al., 2019), which is widely used as a generic baseline in the DP optimization literature. Our utility bound $\widetilde{\mathcal{O}}(\frac{\sqrt{d}}{n\epsilon})$ matches this baseline up to the separability margin and logarithmic factors.

# 4. Preliminaries and Main Results

We begin by describing our problem setup and then present our main results. Proofs are provided in Appendix C–E.

## 4.1. Preliminaries

We consider the following empirical minimization problem for a two-layer KAN classifier $f_\Theta$ parameterized by $\Theta \in \mathbb{R}^{mp(d+1)}$ (see (2)):

$$\min_{\Theta \in \mathbb{R}^{mp(d+1)}} \mathcal{L}_S(\Theta) = \frac{1}{n} \sum_{i=1}^n \ell(y_i f_\Theta(\mathbf{x}_i))$$

given a training data set $S = \{z_i = (\mathbf{x}_i, y_i)\}_{i=1}^n$ where each $z_i$ is independently drawn from the population distribution $\mathcal{P}$. Here, we call $\mathcal{L}_S(\Theta)$ the optimization risk (training loss). Denote $\|\cdot\|_2$ the standard Euclidean norm for vectors and the operator norm for matrices. Throughout the paper, we assume $\|\mathbf{x}_i\|_2 \leq 1$, the binary labels $y_i \in \{-1, +1\}$ and $\ell(\cdot)$ be a nonnegative convex loss function. Define the population risk $\mathcal{L}(\Theta) = \mathbb{E}_{(\mathbf{x},y)\sim\mathcal{P}}[\ell(yf_\Theta(\mathbf{x}))]$.

For notational convenience, we identify the concatenation $[\mathbf{a}^\top, \mathbf{c}^\top]^\top$ with the ordered pair $(\mathbf{a}, \mathbf{c})$. We train $\Theta(k) = (\mathbf{a}(k), \mathbf{c}(k))$ by GD with a step size $\eta > 0$:

$$\mathbf{a}(k+1) = \mathbf{a}(k) - \eta\, \partial_\mathbf{a} \mathcal{L}_S(\Theta(k)),$$

$$\mathbf{c}(k+1) = \mathbf{c}(k) - \eta\, \partial_\mathbf{c} \mathcal{L}_S(\Theta(k)).$$

Following Gao & Tan (2025), we use the standard Gaussian initialization:

$$\mathbf{a}(0) \sim \mathcal{N}(0, \mathbf{I}_{mdp}) \text{ and } \mathbf{c}(0) \sim \mathcal{N}(0, \mathbf{I}_{mp}). \quad (3)$$

We impose two standard assumptions (Gao & Tan, 2025; Taheri et al., 2025). Basis functions with degree exceeding

three (e.g., cubic B-spline basis) and commonly used transformation functions (e.g., sigmoid and hyperbolic tangent) can be chosen so that the following assumption holds.

**Assumption 4.1.** Assume $\sigma$ satisfies $|\sigma(u)| \leq B_\sigma$, $|\sigma'(u)| \leq B'_\sigma$, and $|\sigma''(u)| \leq B''_\sigma$ for all $u \in \mathbb{R}$. Further, assume $\{b_k\}_{k=1}^p$ satisfy $|b_k(v)| \leq B_b$, $|b'_k(v)| \leq B'_b$, and $|b''_k(v)| \leq B''_b$ for all $v \in \text{range}(\sigma)$.

**Assumption 4.2.** Assume the nonnegative convex loss $\ell$ : $\mathbb{R} \to \mathbb{R}_+$ satisfies $|\ell'(u)| \leq B'_\ell$, and $\ell''(u) \leq B''_\ell$ for all $u \in \mathbb{R}$. Further, assume $\ell$ is self-bounded with $\alpha_\ell > 0$, i.e., $|\ell'(u)| \leq \alpha_\ell \ell(u)$ for all $u \in \mathbb{R}$.

The logistic loss naturally satisfies Assumption 4.2 with $B'_\ell = 1$, $B''_\ell = 1/4$ and $\alpha_\ell = 1$. Unless otherwise specified, we set $B'_\ell = B''_\ell = \alpha_\ell = 1$ in the sequel. Throughout the paper, we assume that Assumptions 4.1 and 4.2 hold.

### 4.2. Optimization Guarantees of GD

We now present our main results on the optimization bounds. Our bounds are stated relative to a reference point $\Theta^*$ (e.g., an interpolating solution). For any $\Theta^* \in \mathbb{R}^{mp(d+1)}$, define its *reference-point complexity* by

$$\mathfrak{C}_S(\Theta^*) = 2\eta T \mathcal{L}_S(\Theta^*) + \|\Theta(0) - \Theta^*\|_2^2,$$

and further denote its expected counterpart by $\mathfrak{C}(\Theta^*) = \mathbb{E}_S[\mathfrak{C}_S(\Theta^*)]$ and write $\Lambda_{\Theta^*} = \|\Theta(0) - \Theta^*\|_2$ for the distance from the initialization. Let $C_{\sigma,b} \geq 0$ be a generic constant depending only on the quantities specified in Assumption 4.1. Define $\rho_\ell := C_{\sigma,b} p^3$, which plays the role of a (trajectory-dependent) smoothness parameter for $\mathcal{L}_S$. An explicit expression for $\rho_\ell$ is provided in the appendix. We use $a \gtrsim b$ (resp. $a \lesssim b$) to denote $a \geq Cb$ (resp. $a \leq Cb$) for an absolute constant $C > 0$. Since $p$ is fixed in our parameterization, we suppress its dependence in our results.

Let $\{\Theta(k)\}_{k=0}^T$ be the iterates produced by GD with step size $\eta$, and let $\Theta(T)$ be the output after $T$ iterations. The following theorem gives the convergence behavior of GD.

**Theorem 4.3** (Optimization – General bound). *Let $\Theta^*$ be a reference point. Assume $\eta \leq \min\{1/\rho_\ell, 1\}$ and $\Lambda_{\Theta^*}^2 \geq 4\max\{\eta T \mathcal{L}_S(\Theta^*), \eta \mathcal{L}_S(\Theta(0))\}$. If $m \gtrsim (\log(m/\delta) + \Lambda_{\Theta^*}^2)\Lambda_{\Theta^*}^4$, then with probability at least $1 - \delta$ over the randomness of the initialization, it holds that*

$$\mathcal{L}_S(\Theta(T)) \leq \frac{1}{T}\sum_{k=1}^T \mathcal{L}_S(\Theta(k)) \leq \frac{\mathfrak{C}_S(\Theta^*)}{\eta T}.$$

*Furthermore, for all $k \in [T-1]$, with probability at least $1 - \delta$ over initialization, it holds that*

$$\|\Theta(k+1) - \Theta^*\|_2 \leq \sqrt{2}\Lambda_{\Theta^*}, \quad \|\Theta(k+1) - \Theta(0)\|_2 \leq 3\Lambda_{\Theta^*}.$$

*Discussion of Results.* Beyond the $\mathcal{O}(\mathfrak{C}_S(\Theta^*)/T)$ convergence rate, Theorem 4.3 also shows that the GD iterates

remain in a controlled neighborhood of both the initialization $\Theta(0)$ and the reference point $\Theta^*$. As we show below, this *stay-in-a-ball* property is the cornerstone of our analysis: it allows us to establish that the training loss $\mathcal{L}_S$ is locally smooth and almost convex along the GD trajectory, which in turn yields the convergence guarantees.

*Remark* 4.4 (Key proof idea). A main challenge in proving Theorem 4.3 is that the curvature of the training loss $\mathcal{L}_S$ is *trajectory-dependent*: explicit Hessian calculations show that both the minimum and maximum eigenvalues depend on the evolving deviation $\|\Theta(k) - \Theta(0)\|_2$. To make the curvature control self-consistent, we develop a *double induction* that simultaneously bounds (i) the cumulative training loss $\eta\sum_{s=1}^k \mathcal{L}_S(\Theta(s))$ and (ii) the distances $\|\Theta(k) - \Theta^*\|_2$ and $\|\Theta(k) - \Theta(0)\|_2$. These coupled bounds ensure that all iterates stay inside a region where the Hessian eigenvalues are uniformly controlled, thereby certifying local smoothness and almost-convexity along the trajectory.

**Risks under Realizability.** To clarify the dependency of $\Theta^*$ in Theorem 4.3, we impose a realizability assumption, which posits the existence of a model near the initialization that achieves arbitrarily small training error. Similar assumptions have been considered in (Schliserman & Koren, 2022; Taheri & Thrampoulidis, 2024). We verify this assumption in Theorem 4.9.

**Assumption 4.5** (Realizability). Assume that for almost all draws of $S \sim \mathcal{P}^n$ and for any sufficiently small $\epsilon > 0$, the set $\{\Theta : \mathcal{L}_S(\Theta) \leq \epsilon\}$ is non-empty, and define $g(\epsilon) := \inf\{\|\Theta - \Theta(0)\|_2 : \mathcal{L}_S(\Theta) \leq \epsilon\}$. Assume moreover that the infimum is attained, i.e., there exists $\Theta^\epsilon$ such that $\mathcal{L}_S(\Theta^\epsilon) \leq \epsilon$ and $g(\epsilon) = \|\Theta^\epsilon - \Theta(0)\|_2$.

*Remark* 4.6 (Milder condition). In Lemma C.6, we show that Assumption 4.5 can be relaxed as follows: for any sufficiently small $\epsilon > 0$, there exists $\Theta^\epsilon$ such that $\mathcal{L}_S(\Theta^\epsilon) \leq \epsilon$. This requirement is strictly weaker than Assumption 4.5.

**Theorem 4.7** (Optimization under Realizability). *Suppose Assumption 4.5 holds. Let $\eta \lesssim \min\{g^2(1), g^2(1)(\mathcal{L}_S(\Theta(0)))^{-1}\}$ be a constant, and assume $m \gtrsim (\log(m/\delta) + g^2(1/T))g^4(1/T)$. Then, with probability at least $1 - \delta$ over the randomness of the initialization,*

$$\mathcal{L}_S(\Theta(T)) \leq \frac{1}{T}\sum_{k=1}^T \mathcal{L}_S(\Theta(k)) \lesssim \frac{\eta + g^2\left(\frac{1}{T}\right)}{\eta T}.$$

The above optimization performance depends explicitly on the realizability profile $g$. If $g(\epsilon) \lesssim \log(1/\epsilon)$, then choosing $\eta T \asymp n$ yields $\mathcal{L}_S(\Theta(T)) \lesssim \frac{\log^2(n)}{n}$ under $m \gtrsim \log^6(n)$ (up to logarithmic factors in $1/\delta$).

**Connection to NTK Separability.** We next connect realizability to a separability condition formulated in terms of the NTK features at initialization. As shown in Nitanda et al.

(2019), this separability assumption is weaker than the positivity assumption on the Gram-matrix of NTK considered in the literature (Arora et al., 2019; Du et al., 2019; Gao & Tan, 2025). Let $\langle \cdot, \cdot \rangle$ denote the dot product.

**Assumption 4.8** (NTK separability)**.** Let $\gamma \in (0, 1]$. Assume there exists $\Theta_0 \in \mathbb{R}^{mp(d+1)}$ with $\|\Theta_0\|_2 = 1$ such that $y_i \langle \nabla f_{\Theta(0)}(\mathbf{x}_i), \Theta_0 \rangle \geq \gamma, \forall i \in [n]$.

For concreteness, we specialize to the logistic loss in the following theorem, which shows that with polylogarithmic width, GD attains a $\widetilde{\mathcal{O}}(1/T)$ optimization rate.

**Theorem 4.9** (Optimization under NTK separability)**.** *Suppose Assumption 4.8 holds. Let $\ell$ be the logistic loss. Let $\eta \lesssim 1$ be a constant, assume $m \gtrsim \log(m/\delta)\big(\log^6(T) + \log^3(n/\delta)\big)/\gamma^6$. Then, with probability at least $1 - \delta$ over the randomness of the initialization, it holds that*

$$\mathcal{L}_S(\Theta(T)) \leq \frac{1}{T} \sum_{k=1}^{T} \mathcal{L}_S(\Theta(k)) \lesssim \frac{\log^2(T) + \log\left(\frac{n}{\delta}\right)}{\gamma^2 \eta T}.$$

### 4.3. Generalization Guarantees of GD

To derive generalization (population risk) guarantees for GD, it suffices to control the generalization gap $\mathcal{L}(\Theta(T)) - \mathcal{L}_S(\Theta(T))$ and then combine it with our optimization results (Theorem 4.3). We bound the generalization gap via on-average argument stability (Lei & Ying, 2020). Without loss of generality, we assume $\Lambda_{\Theta^*} \geq 1$ (otherwise we may replace it by 1).

**Theorem 4.10** (Generalization gap via on-average argument stability)**.** *Let $\widetilde{S}$ be an i.i.d. copy of $S$, and let $\Theta^*$ be a reference point independent of $S$. Assume $\eta \leq \min\{1/2\rho_\ell, 1\}$, $\Lambda_{\Theta^*}^2 \geq 8 \max\big\{\eta T\big(\mathcal{L}_S(\Theta^*) + \mathcal{L}_{\widetilde{S}}(\Theta^*)\big), \eta\big(\mathcal{L}_S(\Theta(0)) + \mathcal{L}_{\widetilde{S}}(\Theta(0))\big)\big\}$, and $m \gtrsim \big(\log\left(\frac{m}{\delta}\right) + \Lambda_{\Theta^*}^2\big)\Lambda_{\Theta^*}^4$. Then, with probability at least $1 - \delta$ over the random initialization,*

$$\mathbb{E}_S\big[\mathcal{L}(\Theta(T)) - \mathcal{L}_S(\Theta(T))\big] \lesssim \frac{\eta \Lambda_{\Theta^*}^2}{n} \mathbb{E}_S\Big[\sum_{t=0}^{T} \mathcal{L}_S(\Theta(t))\Big].$$

*Here, the expectation is taken over the draw of $S$.*

Theorem 4.10 shows that the generalization gap is controlled by the cumulative training loss along the GD trajectory. Consequently, whenever GD achieves small training loss, it also enjoys a small generalization gap.

Combining Theorem 4.10 with the optimization bound in Theorem 4.3 yields the following generalization guarantee.

**Theorem 4.11** (Generalization – General bound)**.** *Suppose the assumptions of Theorems 4.10 hold. Then, with probability at least $1 - \delta$ over the randomness of the initialization,*

$$\mathbb{E}_S\big[\mathcal{L}(\Theta(T))\big] \lesssim \big(\frac{\Lambda_{\Theta^*}^2}{n} + \frac{1}{\eta T}\big)\mathfrak{C}(\Theta^*).$$

*Discussion of Results.* Theorem 4.11 has two immediate implications. First, we can certify a suitable reference point $\Theta^*$ such that $\Lambda_{\Theta^*}^2 \leq \mathfrak{C}_S(\Theta^*) \lesssim \big(\log^2(T) + \log(n/\delta)\big)/\gamma^2$, for instance under the NTK separability assumption. Consequently, taking $\eta T \asymp n$ and $m \gtrsim \text{polylog}(n, \delta^{-1})$, a polylogarithmic width already *suffices* to achieve the fast $\widetilde{\mathcal{O}}(1/n)$ rate for the population risk. Second, our theorem highlights an *implicit regularization* effect of GD. Since the bound is stated in terms of $\mathfrak{C}(\Theta^*)$, we may choose $\Theta^* \in \arg\min_\Theta \big\{\mathcal{L}(\Theta) + \frac{1}{2\eta T}\|\Theta - \Theta(0)\|_2^2\big\}$. Thus, our analysis is informative whenever $\inf_\Theta \big\{\mathcal{L}(\Theta) + \frac{1}{2\eta T}\|\Theta - \Theta(0)\|_2^2\big\}$ is small, favoring predictors with low generalization risk that remain close to initialization, consistent with prior observations on implicit bias in overparameterized learning (Oymak & Soltanolkotabi, 2019).

We instantiate Theorem 4.11 under NTK separability, where an fast rate in $n$ up to logarithmic factors is obtained. We also provide population risk bound under realizability assumption in Theorem D.5.

**Theorem 4.12** (Generalization under NTK separability)**.** *Suppose Assumption 4.8 holds. Let $\ell$ be logistic loss. Let $\eta \lesssim 1$ be a constant step size. Assume that $\eta T \gtrsim n$ and $m \gtrsim \log(\frac{m}{\delta})\big(\log^6(T) + \log^3(n/\delta)\big)/\gamma^6$. Then, with probability at least $1 - \delta$ over the randomness of the random initialization, it holds that $\mathbb{E}_S\big[\mathcal{L}(\Theta(T))\big] \lesssim \frac{\log^4(n) + \log^2(n/\delta)}{\gamma^4 n}$.*

### 4.4. Differentially Private Gradient Descent

To enable training KANs on sensitive datasets, we study a differentially private variant of GD (DP-GD). We first present DP-GD with an explicit choice of the Gaussian noise variance ensuring $(\epsilon, \delta)$-DP, and then establish its privacy and utility (population risk) guarantees.

We begin by recalling DP (Dwork et al., 2006). Two datasets $S$ and $S'$ are neighboring if they differ in one data point.

**Definition 4.13** (Differential privacy)**.** We say that a randomized algorithm $\mathcal{A}$ satisfies $(\epsilon, \delta)$-DP if, for any two neighboring datasets $S$ and $S'$ and any measurable set $E$ in the output space of $\mathcal{A}$, it holds $\mathbb{P}(\mathcal{A}(S) \in E) \leq e^\epsilon \mathbb{P}(\mathcal{A}(S') \in E) + \delta$. We say $\mathcal{A}$ satisfies $\epsilon$-DP if $\delta = 0$.

We ensure DP by adding Gaussian noise to the gradients at each iteration and projecting the noisy updates onto bounded domains. The noise variances are chosen according to the $\ell_2$-sensitivity (Definition E.2) of the per-iteration gradients.

Specifically, let $\mathbf{a}(0) \in \mathbb{R}^{mdp}$ and $\mathbf{c}(0) \in \mathbb{R}^{mp}$ be the initialization of DP-GD generated according to (3). For $R > 0$, let $\mathcal{B}(\bar{u}, R)$ denote the closed Euclidean ball centered at $\bar{u}$ with radius $R$. We define the parameter domains $\Omega_{\mathbf{a}} = \mathcal{B}(\mathbf{a}(0), R_1)$ and $\Omega_{\mathbf{c}} = \mathcal{B}(\mathbf{c}(0), R_2)$ for some constants $R_1, R_2 > 0$, and let $\text{Proj}_\Omega(\cdot)$ denote the Euclidean projection onto $\Omega$. Rather than using gradient clipping, we

control sensitivity by constraining the iterates via projection. Given dataset $S$, step size $\eta$, and privacy parameters $(\epsilon, \delta)$. For $k = 0, \ldots, T - 1$, let $\widetilde{\Theta}(k) = (\mathbf{a}(k), \mathbf{c}(k))$, DP-GD updates $\mathbf{a}(k)$ and $\mathbf{c}(k)$ via

$$\mathbf{a}(k+1) = \text{Proj}_{\Omega_{\mathbf{a}}}\big(\mathbf{a}(k) - \eta\big(\partial_{\mathbf{a}}\mathcal{L}_S(\widetilde{\Theta}(k)) + \mathbf{b}_1(k)\big)\big), \quad (4)$$

$$\mathbf{c}(k+1) = \text{Proj}_{\Omega_{\mathbf{c}}}\big(\mathbf{c}(k) - \eta\big(\partial_{\mathbf{c}}\mathcal{L}_S(\widetilde{\Theta}(k)) + \mathbf{b}_2(k)\big)\big). \quad (5)$$

Here, $\mathbf{b}_1(k) \in \mathbb{R}^{mdp}$ and $\mathbf{b}_2(k) \in \mathbb{R}^{mp}$ are independent Gaussian vectors given by

$$\mathbf{b}_1(k) \overset{\text{i.i.d.}}{\sim} \mathcal{N}(0, \sigma_1^2 \mathbf{I}_{mdp}) \text{ and } \mathbf{b}_2(k) \overset{\text{i.i.d.}}{\sim} \mathcal{N}(0, \sigma_2^2 \mathbf{I}_{mp}).$$

with $\sigma_1^2 = C_1 \tilde{\sigma}^2$ and $\sigma_2^2 = C_2 \tilde{\sigma}^2$. Here, $\tilde{\sigma}^2 = T\big(1 + \frac{\log(2T/\delta)}{\epsilon}\big)(n^2\epsilon)^{-1}$, $C_1 = 8(B'_\ell B'_\sigma B'_b B_b)^2 p^2 \big(4\sqrt{p} + \frac{2\sqrt{\log(2/\delta)} + R_2}{\sqrt{m}}\big)^2$, and $C_2 = 8(B'_\ell B_b)^2 p$. The detailed procedure is summarized in Algorithm 1 (see Appendix E), and its privacy guarantee is stated below.

**Theorem 4.14** (Privacy guarantee). *Algorithm 1 satisfies $(\epsilon, \delta)$-DP.*

Unlike non-private GD, DP-GD injects additive Gaussian noise into the gradient at every iteration. Consequently, the iterates follow a stochastic trajectory and monotonic decrease of the training loss is generally not guaranteed in this nonconvex setting. We therefore study the averaged population risk along the DP-GD trajectory.

Fix any reference point $\Theta^* = (\mathbf{a}^*, \mathbf{c}^*)$. We assume $\mathbf{a}^* \in \Omega_{\mathbf{a}}$, $\mathbf{c}^* \in \Omega_{\mathbf{c}}$, and that $\Theta^*$ is independent of $S$. $\widetilde{\Lambda}_{\Theta^*} = \|\widetilde{\Theta}(0) - \Theta^*\|_2$. Let $\bar{\rho} = C_{\sigma,b}\, p^2 \big(p + \frac{R_2^2}{m}\big)$ and $R = R_1 + R_2$.

**Theorem 4.15** (Utility guarantee – General bound). *Let $\{\widetilde{\Theta}(k)\}_{k=1}^T$ be produced by Algorithm 1. Assume $\|\widetilde{\Theta}(0) - \Theta^*\|_2^2 \geq C \max\{\eta T\big(\mathcal{L}_S(\Theta^*) + \mathcal{L}_{\widetilde{S}}(\Theta^*)\big), \eta\big(\mathcal{L}_S(\widetilde{\Theta}(0)) + \mathcal{L}_{\widetilde{S}}(\widetilde{\Theta}(0))\big)\}$ and $\eta \leq \min\{1/3\bar{\rho}, 1\}$. If $m \gtrsim \big(\log(\frac{m}{\delta}) + R^2\big) \max\big\{R^4 + \widetilde{\Lambda}_{\Theta^*}^4, \big(\frac{\eta T \log(T/\delta)}{n\epsilon}\big)^2\big\}$ and*

$$m \lesssim (n\epsilon)^4 \big(d^2(\eta T)^4(\log(m/\delta) + R^2)\log^2(T/\delta)\big)^{-1},$$

*then with probability at least $1 - \delta$ over the randomness of the initialization, $\frac{1}{T}\sum_{k=1}^T \mathbb{E}_{S,\mathcal{A}}\big[\mathcal{L}(\widetilde{\Theta}(k))\big]$ is controlled by*

$$\Big(\frac{1}{\eta T} + \frac{m^{\frac{1}{4}}}{n}\Big)\widetilde{\Lambda}_{\Theta^*}^2 + \big(1 + \frac{\eta T}{n}\big)\frac{m\eta T d \log(\frac{T}{\delta})}{n^2\epsilon^2} + \delta.$$

*Discussion of Results.* A key contribution of our DP analysis is a *width characterization* for private KAN training: we prove matching upper and lower bounds on an admissible width regime. In particular, under the NTK separability setting (see Theorem 4.16), a polylogarithmic scaling of $m$ is not only sufficient but also essentially necessary for DP-GD to attain a utility rate $\mathcal{O}(\sqrt{d}/n\epsilon)$, matching the classical

convex Lipschitz lower bound in its dependence on $d$, $n$, and $\epsilon$ in Bassily et al. (2019).

We next specialize Theorem 4.15 to the NTK-separable setting, obtaining a utility rate of order $\mathcal{O}(\sqrt{d}/(n\epsilon))$.

**Theorem 4.16** (Utility under NTK Separability). *Let Assumption 4.8 holds. Assume $\eta \lesssim 1$ be a constant, $\delta \lesssim \frac{\sqrt{d}}{n\epsilon}$, $\eta T \asymp \frac{\gamma^2 n\epsilon}{\sqrt{d}\log^{5/2}(n/\delta)}$ and $m \asymp \log^6(n/\delta)/\gamma^6$. Then, with probability at least $1 - \delta$ over initialization,*

$$\frac{1}{T}\sum_{k=1}^T \mathbb{E}_{S,\mathcal{A}}\big[\mathcal{L}(\widetilde{\Theta}(k))\big] \lesssim \log^6\big(\frac{n}{\delta}\big)\frac{\sqrt{d}}{\gamma^4 n\epsilon}.$$

# 5. Practical Implications and Experiments

In this section, we empirically validate our theoretical results and translate them into concrete practical guidance for achieving optimal inference performance.

## 5.1. Practical Implications for Training KANs Using GD

Our theory provides explicit recommendations for choosing the network width $m$ and the number of iterations $T$, identifying critical regimes for both GD and DP-GD.

**Training and test accuracy saturate beyond critical width.** Both the optimization bound (Theorem 4.9) and the generalization bound (Theorem 4.12) indicate that increasing the width $m$ improves accuracy only up to a critical threshold $m_{\text{crit}}$ (typically on the order of $\log(n)$). Beyond this regime, both training and test accuracy plateau and may exhibit fluctuations, suggesting that further increasing the network width is unnecessary and offers limited benefit.

**Training accuracy may continue to improve with training iterations.** The optimization bound furthermore predicts that training accuracy may continue to improve as the number of GD iterations $T$ increases. However, this suggests that additional iterations primarily benefit optimization rather than generalization.

**Test accuracy saturates with increasing training iterations.** In contrast, the generalization bound indicates that test accuracy saturates beyond a critical threshold $T_{\text{crit}}$ and may even deteriorate. Consequently, running GD for substantially longer yields at best marginal additional generalization gains.

**Private utility is maximized when width and training iterations lie in an admissible range.** The utility bound of Theorem 4.16 provides explicit guidance for choosing the network width $m$ and the number of training iterations $T$ under a fixed privacy budget. In particular, the bound suggests using a polylogarithmic width and an effective training horizon $T$ on the order of $n\epsilon/\sqrt{d}$ (up to logarithmic factors and problem-dependent constants) to achieve favorable utility.

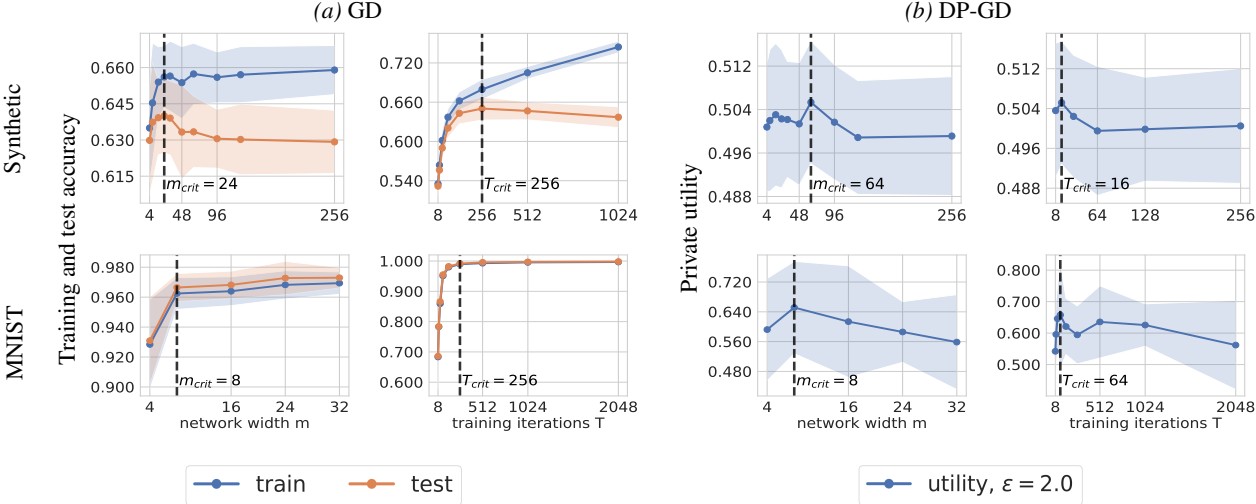

*Figure 2.* (a) Training and test accuracy as a function of the width $m$ (with $T$ fixed) and the number of iterations $T$ (with $m$ fixed) for GD, and (b) private utility as a function of $m$ and $T$ for DP-GD, on synthetic logistic data (top row) and MNIST (bottom row). In (a), the vertical dashed lines indicate empirically observed change points beyond which increasing the width $m$ yields diminishing or flat training and test accuracy, while increasing the number of training iterations $T$ primarily improves training accuracy but leads to diminishing or flat test accuracy, consistent with the theoretical bounds. In (b), the dashed lines indicate observed turning points beyond which private utility degrades due to the amplification or accumulation of privacy noise, in line with the private utility bound. Together, these results suggest selecting width and training duration within the admissible regimes identified by the theory.

This implies that the width $m$ should be sufficiently large to enable learning, but not so large that it excessively amplifies the injected noise. Similarly, the number of iterations $T$ should be large enough to benefit optimization, yet small enough to avoid excessive accumulation of privacy noise, thereby motivating early stopping.

### 5.2. Empirical Validation

Figure 2 reports the accuracy of GD and DP-GD on synthetic logistic data and MNIST, supporting the practical implications of our theory. Experimental details and hyperparameter settings are provided in Appendix F.

Figure 2(a) shows that, on both datasets, increasing the width $m$ yields rapid gains in training and test accuracy in the small-width regime (up to approximately $m_{\mathrm{crit}} \approx 24$), after which accuracy diminishes and the performance curves become largely flat, consistent with the theoretical bounds. As the number of training iterations $T$ increases, training accuracy improves (or remains near its peak), whereas test accuracy stabilizes much earlier. In particular, once $T$ reaches a moderate scale (around $T_{\mathrm{crit}} \approx 256$), further iterations provide little additional improvement in test accuracy.

Figure 2(b) shows that for fixed privacy budget $\epsilon = 2$, private utility (test accuracy at the last iterate) increases with either $m$ or $T$ up to an observed turning point (up to approximately $m_{\mathrm{crit}} \approx 64$ and $T_{\mathrm{crit}} \approx 64$), after which performance degrades, highlighting the importance of moderate widths and early stopping. On MNIST, private utility peaks around $m \approx 8$ and $T \approx 64$. Overall, the empirical trends align well

with our theoretical implications.

### 5.3. Separability Validation

To further examine the practical achievability of Assumption 4.8, we conduct an additional empirical study on synthetic datasets that are not linearly separable in the input space. For KAN, we use the same two-layer model as in (2). As a baseline, we consider a parameter-matched two-layer ReLU network of the form

$$f_{\mathrm{ReLU}}(x) = \frac{1}{\sqrt{h}} \sum_{j=1}^{h} c_j \, \mathrm{ReLU}\left(\frac{\langle \mathbf{a}_j, \mathbf{x} \rangle}{\sqrt{d}}\right),$$

with Gaussian initialization $\mathbf{a}_j \sim \mathcal{N}(0, \mathbf{I}_d)$ and $c_j \sim \mathcal{N}(0, 1)$. For a KAN with hidden width $m$, we set the ReLU width to $h = pm$, where $p$ is the number of spline basis functions, so that the two models have the same number of parameters.

We consider three structured nonlinear datasets. In the *multi-interval one-dimensional* dataset, the label is determined by a non-monotone rule on a single coordinate: points with $x_1$ in two outer intervals are assigned one label, while points in a middle interval are assigned the other. In the *XOR-gap* dataset, the label is given by the sign pattern of two active coordinates, with a margin gap excluding points near the coordinate axes. In the *checkerboard-gap* dataset, the label is determined by the parity of the signs of several active coordinates, again with a gap around the decision boundaries. In all cases, the remaining coordinates are sampled as nuisance dimensions.

_Table 1._ Empirical NTK margin on nonlinear synthetic data.

| Model | Data | $m = 8$ | $m = 64$ | $m = 512$ |
|-------|------|---------|----------|-----------|
| ReLU | Multi-int. | 0.0151 | 0.0191 | 0.0197 |
| KAN | Multi-int. | **0.0400** | **0.0883** | **0.1027** |
| ReLU | XOR | 0.0226 | 0.0285 | 0.0284 |
| KAN | XOR | **0.0243** | **0.0503** | **0.0599** |
| ReLU | Checker. | 0.0060 | 0.0104 | 0.0111 |
| KAN | Checker. | **0.0206** | **0.0451** | **0.0538** |

For each model and width, we compute the empirical NTK margin at random initialization by solving the corresponding max-margin problem in the tangent feature space,

$$\max_{\|\Theta\|_2 \leq 1} \min_{i \in [n]} y_i \langle \nabla f_{\Theta(0)}(x_i), \Theta \rangle.$$

We repeat the experiment over multiple random initializations and report the mean margin. Table 1 summarizes the results for representative widths $m \in \{8, 64, 512\}$. On all three nonlinear datasets, the empirical NTK margin of KAN is positive and increases with width. Moreover, it is consistently larger than that of the parameter-matched ReLU network, with the gap becoming more pronounced at larger widths. While these results do not provide a complete characterization of Assumption 4.8, they offer additional evidence that NTK separability is practically plausible for structured nonlinear data aligned with the KAN architecture.

## 6. Related Work

There are only few theoretical results analyzing the training dynamics of KAN under gradient methods. In this section, we discuss related work on KANs and differentially private optimization, and position our results relative to existing guarantees. Additional discussion on analyses for MLPs is deferred to Appendix A.

**Gradient-based analysis for KANs.** Besides Gao & Tan (2025), Wang et al. (2025c) studies expressiveness and GD dynamics via spectral bias in a simplified setting, which is orthogonal to our guarantees. Most existing generalization results for KANs are *algorithm-independent*, e.g., Zhang & Zhou (2025); Li et al. (2025b); Liu et al. (2025a) rely on uniform-convergence-style arguments and therefore do not yield bounds tailored to GD iterates.

**Utility analysis for DP-GD on MLPs.** Rigorous utility analysis for DP-GD and DP-SGD on MLPs remain limited (Bu et al., 2023; Romijnders & Koskela, 2025; Shi et al., 2026; Wang et al., 2025a). Wang et al. (2025a) derive utility bounds for smooth three-layer MLPs in an over-scaled regime, and explicitly leave the standard $1/\sqrt{m}$ scaling as an open problem. Other works rely on convex reformulations or alternative threat models (Romijnders & Koskela, 2025), or provide task-specific evidence that noise can act as an implicit regularizer (Shi et al., 2026). There are also

works on DP optimization for nonconvex objectives, which mainly focus on stationarity or empirical objective gaps (Zhang et al., 2017; Wang et al., 2019; Bassily et al., 2021; Lowy et al., 2024).

## 7. Conclusion and Limitations

We provide optimization, generalization, and private utility bounds for training KANs with (DP-)GD, identifying regimes of width and training duration beyond which performance saturates or degrades. These insights offer concrete guidance for choosing model size and training horizons and are supported by experiments on synthetic data and MNIST. Several limitations point to future work: extending the analysis beyond two-layer KANs, relaxing the smoothness assumption on $\sigma$ to cover non-smooth activations such as ReLU, and generalizing our guarantees from GD to SGD.

## Impact Statement

This paper advances the theoretical understanding of training Kolmogorov–Arnold Networks under differential privacy. Our results provide explicit $(\epsilon, \delta)$-DP guarantees and corresponding utility bounds, which can support the principled use of gradient-based training on sensitive datasets (e.g., in biology and medicine) by limiting the influence of any single individual's record. At the same time, our bounds make explicit the privacy–utility trade-off and the dependence of performance on design choices such as width and early stopping. As with any DP method, these guarantees mitigate but do not eliminate privacy risks, and practitioners should select privacy parameters $(\epsilon, \delta)$ and validate models in the intended deployment context.

## Acknowledgment

PW acknowledges support by the Alexander-von-Humboldt Foundation through a Humboldt Research Fellowship. MK acknowledges support by the DFG through FOR 5359 (ID 459419731), TRR 375 (ID 511263698), SPP 2298 (ID 464252197 and 441826958), and SPP 2331 (ID 441958259), by the Carl-Zeiss Foundation through the initiative AI-Care, and by the BMFTR award 01IS24071A.

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

# Appendix

## A. Further Related work for MLPs

Most existing optimization and generalization results on MLPs focuses on *fully-connected* architectures. A central tool is the NTK (Jacot et al., 2018), which has enabled global convergence analyses and, in the lazy-training regime, sharp statistical rates in various settings (Arora et al., 2019; Cao & Gu, 2019; Allen-Zhu et al., 2019; Zou et al., 2020; Chen et al., 2021; Nguyen & Mücke, 2024; Nitanda & Taiji, 2021), often requiring widths that scale polynomially with $n$. In parallel, generalization has been studied via uniform convergence using capacity measures such as Rademacher complexity and covering numbers (Bartlett et al., 2017; Frei et al., 2023; Ji & Telgarsky, 2020; Nitanda et al., 2019; Lei et al., 2026; Li et al., 2025c; Zhou et al., 2024; 2026). More recently, there has been growing interest in generalization analysis based on algorithmic stability (Richards & Kuzborskij, 2021; Lei et al., 2022; Taheri et al., 2025; Taheri & Thrampoulidis, 2024; Wang et al., 2025b), which provides algorithm-dependent generalization bounds directly tied to the training dynamics.

## B. Useful Lemmas

In this section, we introduce several useful lemmas that will be used in the proofs.

**Lemma B.1** (Concentration of the norm of Sub-Gaussian vectors (Wainwright, 2019))**.** *Let $\mathbf{u} \in \mathbb{R}^s$ be a centered Sub-Gaussian vector with parameter $\sigma^2$, i.e., $\mathbb{E}[u_i] = 0$ and $\mathbb{E}[e^{\lambda u_i}] \leq e^{\lambda^2 \sigma^2 / 2}$ for all $\lambda > 0$ and $i \in [s]$. Then with probability at least $1 - \delta$ for $\delta \in (0, 1)$, it holds that*

$$\|\mathbf{u}\|_2 \leq 4\sigma\sqrt{s} + 2\sigma\sqrt{\log(\frac{1}{\delta})}.$$

**Corollary B.2.** *With probability at least $1 - \delta$ over initialization $\mathbf{c}(0)$, it holds that*

$$\|\mathbf{c}(0)\|_2 \leq 4\sqrt{pm} + 2\sqrt{\log(\frac{2}{\delta})} \qquad and \qquad \max_{i \in [m]} \|\mathbf{c}_i(0)\|_2 \leq 4\sqrt{p} + 2\sqrt{\log(\frac{2m}{\delta})}. \tag{6}$$

*Proof.* Note $\mathbf{c}(0) \sim \mathcal{N}(0, \mathbf{I}_{mp})$. Applying Lemma B.1 twice with $\mathbf{u} = \mathbf{c}(0)$, $s = mp$ and $\sigma^2 = 1$ and with $\mathbf{u} = \mathbf{c}_i(0)$, $s = p$ and $\sigma^2 = 1$ over all $i \in [m]$, respectively, we can obtain the desried results. □

**Lemma B.3.** *Let $s \in \mathbb{N}$ and $\mathbf{A}_i \in \mathbb{R}^{m_i \times n_i}$ for $i \in [s]$. Define the diagonal block matrix*

$$\mathbf{B} = \begin{bmatrix} \mathbf{A}_1 & \cdots & \mathbf{0} \\ \vdots & \ddots & \vdots \\ \mathbf{0} & \cdots & \mathbf{A}_s \end{bmatrix} \in \mathbb{R}^{\sum_{i=1}^s m_i \times \sum_{i=1}^s n_i}$$

*Then, it holds that*

$$\|\mathbf{B}\|_2 = \max_{i \in [s]} \|\mathbf{A}_i\|_2.$$

*Proof.* Since the operator norm of any PSD matrix $\mathbf{A}$ is equal to the associated largest eigenvalue, i.e., $\|\mathbf{A}\|_2 = \lambda_{\max}(\mathbf{A})$. Then, it holds that

$$\|\mathbf{B}\|_2 = \sqrt{\|\mathbf{B}^\top \mathbf{B}\|_2} = \sqrt{\lambda_{\max}(\mathbf{B}^\top \mathbf{B})} = \max_{i \in [s]} \sqrt{\lambda_{\max}(\mathbf{A}_i^\top \mathbf{A}_i)} = \max_{i \in [s]} \sqrt{\|\mathbf{A}_i^\top \mathbf{A}_i\|_2} = \max_{i \in [s]} \|\mathbf{A}_i\|_2,$$

which completes the proof. □

Denote $[\Theta_1, \Theta_2] = \{\alpha\Theta_1 + (1 - \alpha)\Theta_2 : \alpha \in [0, 1]\}$ as the line segment between $\Theta_1$ and $\Theta_2$.

**Lemma B.4** (Local quasi-convexity property (Taheri & Thrampoulidis, 2024))**.** *Suppose $G : \mathbb{R}^d \to \mathbb{R}$ be a second-order differentiable function satisfying $\lambda_{\min}(\nabla^2 G(\Theta)) \geq -\kappa G(\Theta)$. Let $\Theta_1, \Theta_2 \in \mathbb{R}^d$ be two arbitrary points with distance $\|\Theta_1 - \Theta_2\|_2 \leq D \leq \sqrt{2/\kappa}$. Let $\tau := (1 - D^2\kappa/2)^{-1}$. Then,*

$$\max_{\mathcal{V} \in [\Theta_1, \Theta_2]} G(\mathcal{V}) \leq \tau \max\{G(\Theta_1), G(\Theta_2)\}.$$

## C. Proofs for Optimization.

**Matrix form of** $f_\Theta$: For any scalar $v \in \mathbb{R}$, we denote $\mathbf{h}(v) = [b_1(v), \ldots, b_p(v)]^\top \in \mathbb{R}^p$. For $s \in \mathbb{N}$ and a vector $\mathbf{u} = [u_1, \ldots, u_s]^\top \in \mathbb{R}^s$, we denote $\mathbf{h}(\mathbf{u}) = [\mathbf{h}(u_1)^\top, \ldots, \mathbf{h}(u_s)^\top]^\top \in \mathbb{R}^{sp}$.

To proof Proposition C.2, we need the following results for gradients and Hessians. To better understand the gradients and Hessians, we introduce the matrix multiplication form of $f_\Theta$. Denote $\mathbf{A} = [\mathbf{a}_1, \ldots, \mathbf{a}_m]^\top \in \mathbb{R}^{m \times dp}$, $f_\Theta$ can be rewritten as

$$f_\Theta(\mathbf{x}) = \frac{1}{\sqrt{m}} \mathbf{c}^\top \mathbf{h}\Big(\sigma\big(\frac{1}{\sqrt{d}} \mathbf{A}\mathbf{h}(\mathbf{x})\big)\Big).$$

Let $C_{\sigma,b}$ be a constant that depend solely on $\sigma, b$.

**Lemma C.1** (Gradient and Hessian). *Let $\delta \in (0,1)$. Suppose (6) and Assumptions 4.1 and 4.2 hold. It holds for any* $\Theta = (\mathbf{a}, \mathbf{c})$ *and any* $\mathbf{x} \in \mathcal{X}$ *that*

$$\big\|\nabla f_\Theta(\mathbf{x})\big\|_2 \leq C_{\sigma,b}\, p \left( \sqrt{p} + \sqrt{\frac{\log(1/\delta)}{m}} + \max_{i \in [m]} \|\mathbf{c}_i(0) - \mathbf{c}_i\|_2 \right)$$

*and*

$$\big\|\nabla^2 f_\Theta(\mathbf{x})\big\|_2 \leq \frac{C_{\sigma,b}\, p^{\frac{3}{2}} \big(\sqrt{p} + \sqrt{\log(m/\delta)} + \max_{i \in [m]} \|\mathbf{c}_i(0) - \mathbf{c}_i\|_2\big)}{\sqrt{m}}.$$

*Proof.* We will control $\|\nabla f_\Theta(\mathbf{x})\|_2$ and $\|\nabla^2 f_\Theta(\mathbf{x})\|_2$ separately. It's obvious that

$$\big\|\partial_\mathbf{c} f_\Theta(\mathbf{x})\big\|_2 = \frac{1}{\sqrt{m}} \big\|\mathbf{h}\big(\sigma\big(\frac{1}{\sqrt{d}} \mathbf{A}\mathbf{h}(\mathbf{x})\big)\big)\big\|_2 \leq \sqrt{p} B_b. \tag{7}$$

For all $v \in \mathbb{R}$, we denote $\mathbf{h}'(v) = [b_1'(v), \ldots, b_p'(v)]^\top \in \mathbb{R}^p$ and $\mathbf{h}''(v) = [b_1''(v), \ldots, b_p''(v)]^\top \in \mathbb{R}^p$. Define

$$\mathbf{u}(\mathbf{x}) = \sigma\big(\frac{1}{\sqrt{d}} \mathbf{A}\mathbf{h}(\mathbf{x})\big) \quad \text{and} \quad \mathbf{D}(\mathbf{x}) = \mathrm{diag}\big(\sigma'\big(\frac{1}{\sqrt{d}} \mathbf{a}_i^\top \mathbf{h}(\mathbf{x})\big)\big)_{i=1}^m \in \mathbb{R}^{m \times m}. \tag{8}$$

Note that $f_\Theta(\mathbf{x}) = \frac{1}{\sqrt{m}} \mathbf{c}^\top \mathbf{h}(\mathbf{u}(\mathbf{x}))$,

$$\frac{\partial \mathbf{h}(\mathbf{u}(\mathbf{x}))}{\partial \mathbf{u}(\mathbf{x})} = \begin{bmatrix} \mathbf{h}'(u_1(\mathbf{x})) & \mathbf{0} & \cdots & \mathbf{0} \\ \mathbf{0} & \mathbf{h}'(u_2(\mathbf{x})) & \cdots & \mathbf{0} \\ \vdots & \vdots & \ddots & \vdots \\ \mathbf{0} & \mathbf{0} & \mathbf{0} & \mathbf{h}'(u_m(\mathbf{x})) \end{bmatrix} \in \mathbb{R}^{mp \times m}$$

and

$$\frac{\partial \mathbf{u}(\mathbf{x})}{\partial \mathbf{a}_i} = \begin{bmatrix} \mathbf{0} \\ \vdots \\ \frac{1}{\sqrt{d}} \sigma'\big(\frac{1}{\sqrt{d}} \mathbf{a}_i^\top \mathbf{h}(\mathbf{x})\big) \mathbf{h}(\mathbf{x})^\top \\ \vdots \\ \mathbf{0} \end{bmatrix} \in \mathbb{R}^{m \times pd}.$$

According to the chain rule, for any $i \in [m]$, it holds that

$$\partial_{\mathbf{a}_i} f_\Theta(\mathbf{x}) = \frac{\partial f_\Theta(\mathbf{x})}{\partial \mathbf{h}(\mathbf{u}(\mathbf{x}))} \frac{\partial \mathbf{h}(\mathbf{u}(\mathbf{x}))}{\partial \mathbf{u}(\mathbf{x})} \frac{\partial \mathbf{u}(\mathbf{x})}{\partial \mathbf{a}_i} = \frac{1}{\sqrt{md}} \langle \mathbf{c}_i, \mathbf{h}'(u_i(\mathbf{x})) \rangle \sigma'\big(\frac{1}{\sqrt{d}} \mathbf{a}_i^\top \mathbf{h}(\mathbf{x})\big) \mathbf{h}(\mathbf{x})^\top.$$

Recall that $\mathbf{a} = \mathrm{Vec}\big(\{\mathbf{a}_i\}_{i=1}^m\big) \in \mathbb{R}^{mpd}$ is the vectorization of $\mathbf{a}$. It holds that

$$\partial_\mathbf{a} f_\Theta(\mathbf{x}) = \frac{1}{\sqrt{md}} \mathrm{Vec}\Big(\big\{\sigma'\big(\frac{1}{\sqrt{d}} \mathbf{a}_i^\top \mathbf{h}(\mathbf{x})\big) \langle \mathbf{c}_i, \mathbf{h}'(u_i(\mathbf{x})) \rangle \mathbf{h}(\mathbf{x})\big\}_{i=1}^m\Big) \in \mathbb{R}^{mpd},$$

and its associated $\|\cdot\|_2$-norm can be controlled by

$$\|\partial_{\mathbf{a}} f_\Theta(\mathbf{x})\|_2 = \frac{1}{\sqrt{md}} \Big( \sum_{i=1}^m \sigma'\big(\frac{1}{\sqrt{d}}\mathbf{a}_i^\top \mathbf{h}(\mathbf{x})\big)^2 \big|\langle \mathbf{c}_i, \mathbf{h}'(u_i(\mathbf{x}))\rangle\big|^2 \|\mathbf{h}(\mathbf{x})\|_2^2 \Big)^{\frac{1}{2}}$$

$$\leq B'_\sigma B_b \sqrt{\frac{p}{m}} \Big( \sum_{i=1}^m \big|\langle \mathbf{c}_i, \mathbf{h}'(u_i(\mathbf{x}))\rangle\big|^2 \Big)^{\frac{1}{2}}$$

$$\leq B'_\sigma B_b B'_b \sqrt{\frac{p^2}{m}} \Big( \sum_{i=1}^m \|\mathbf{c}_i\|_2^2 \Big)^{\frac{1}{2}}$$

$$\leq B'_\sigma B_b B'_b \sqrt{\frac{p^2}{m}} \big( \|\mathbf{c}(0)\|_2 + \|\mathbf{c}(0) - \mathbf{c}\|_2 \big) \tag{9}$$

$$\leq B'_\sigma B_b B'_b p \Big( 4\sqrt{p} + 2\sqrt{\frac{\log(\frac{2}{\delta})}{m}} + \max_{i\in[m]} \|\mathbf{c}_i(0) - \mathbf{c}_i\|_2 \Big), \tag{10}$$

where the last inequality used (6). Combining the estimates for $\|\partial_{\mathbf{c}} f_\Theta(\mathbf{x})\|_2$ (see (7)) and $\|\partial_{\mathbf{a}} f_\Theta(\mathbf{x})\|_2$ (see (10)) and the fact $\nabla f_\Theta(\mathbf{x}) = [\partial_{\mathbf{c}} f_\Theta(\mathbf{x})^\top, \partial_{\mathbf{a}} f_\Theta(\mathbf{x})^\top]^\top$, it holds that

$$\big\| \nabla f_\Theta(\mathbf{x}) \big\|_2 \leq C_{\sigma,b}\, p \Big( \sqrt{p} + \sqrt{\frac{\log(\frac{1}{\delta})}{m}} + \max_{i\in[m]} \|\mathbf{c}_i(0) - \mathbf{c}_i\|_2 \Big). \tag{11}$$

Now, we turn to estimate the Hessian of $f_\Theta(\mathbf{x})$. Note that

$$\partial_{\mathbf{a}}^2 f_\Theta(\mathbf{x}) = \begin{bmatrix} \partial_{\mathbf{a}_1}^2 f_\Theta(\mathbf{x}) & \cdots & \mathbf{0} \\ \vdots & \ddots & \vdots \\ \mathbf{0} & \cdots & \partial_{\mathbf{a}_m}^2 f_\Theta(\mathbf{x}) \end{bmatrix} \in \mathbb{R}^{mpd \times mpd},$$

where

$$\partial_{\mathbf{a}_i}^2 f_\Theta(\mathbf{x}) = \frac{1}{d\sqrt{m}} \Big( \sigma''\big(\frac{1}{\sqrt{d}}\mathbf{a}_i^\top \mathbf{h}(\mathbf{x})\big) \langle \mathbf{c}_i, \mathbf{h}'\big(\sigma\big(\frac{1}{\sqrt{d}}\mathbf{a}_i^\top \mathbf{h}(\mathbf{x})\big)\big) \rangle$$

$$+ \big(\sigma'\big(\frac{1}{\sqrt{d}}\mathbf{a}_i^\top \mathbf{h}(\mathbf{x})\big)^2 \langle \mathbf{c}_i, \mathbf{h}''\big(\sigma\big(\frac{1}{\sqrt{d}}\mathbf{a}_i^\top \mathbf{h}(\mathbf{x})\big)\big) \rangle \big) \Big) \mathbf{h}(\mathbf{x})\mathbf{h}(\mathbf{x})^\top \in \mathbb{R}^{pd \times pd}.$$

Denoting $\mathbf{w}_i = \sigma''\big(\frac{1}{\sqrt{d}}\mathbf{a}_i^\top \mathbf{h}(\mathbf{x})\big) \mathbf{h}'\big(\sigma\big(\frac{1}{\sqrt{d}}\mathbf{a}_i^\top \mathbf{h}(\mathbf{x})\big)\big) + \sigma'\big(\frac{1}{\sqrt{d}}\mathbf{a}_i^\top \mathbf{h}(\mathbf{x})\big)^2 \mathbf{h}''\big(\sigma\big(\frac{1}{\sqrt{d}}\mathbf{a}_i^\top \mathbf{h}(\mathbf{x})\big)\big)$. We can rewrite $\partial_{\mathbf{a}_i}^2 f_\Theta(\mathbf{x})$ as

$$\partial_{\mathbf{a}_i}^2 f_\Theta(\mathbf{x}) = \frac{1}{d\sqrt{m}} \langle \mathbf{c}_i, \mathbf{w}_i \rangle \mathbf{h}(\mathbf{x})\mathbf{h}(\mathbf{x})^\top.$$

Note $\partial_{\mathbf{a}}^2 f_\Theta(\mathbf{x})$ is the block diagonal matrix. From Lemma B.3 we know

$$\big\| \partial_{\mathbf{a}}^2 f_\Theta(\mathbf{x}) \big\|_2 = \max_{i\in[m]} \big\| \partial_{\mathbf{a}_i}^2 f_\Theta(\mathbf{x}) \big\|_2 = \max_{i\in[m]} \sup_{\|\mathbf{v}\|_2=1} \big| \mathbf{v}^\top \partial_{\mathbf{a}_i}^2 f_\Theta(\mathbf{x})\, \mathbf{v} \big|$$

$$= \frac{1}{d\sqrt{m}} \max_{i\in[m]} \sup_{\|\mathbf{v}\|_2=1} \big| \langle \mathbf{c}_i, \mathbf{w}_i \rangle \langle \mathbf{h}(\mathbf{x}), \mathbf{v} \rangle^2 \big| \leq \frac{1}{d\sqrt{m}} \max_{i\in[m]} \|\mathbf{c}_i\|_2 \|\mathbf{w}_i\|_2 \|\mathbf{h}(\mathbf{x})\|_2^2$$

$$\leq \frac{1}{d\sqrt{m}} \|\mathbf{h}(\mathbf{x})\|_2^2 \max_{i\in[m]} \Big[ \|\mathbf{w}_i\|_2 \big( \|\mathbf{c}_i(0)\|_2 + \|\mathbf{c}_i(0) - \mathbf{c}_i\|_2 \big) \Big]$$

$$\leq B_b^2 \frac{p}{\sqrt{m}} \max_{i\in[m]} \Big[ \|\mathbf{w}_i\|_2 \big( \|\mathbf{c}_i(0)\|_2 + \|\mathbf{c}_i(0) - \mathbf{c}_i\|_2 \big) \Big]$$

$$\leq B_b^2 \frac{p}{\sqrt{m}} \max_{i\in[m]} \|\mathbf{w}_i\|_2 \Big( 4\sqrt{p} + 2\sqrt{\log(\frac{m}{\delta})} + \max_{i\in[m]} \|\mathbf{c}_i(0) - \mathbf{c}_i\|_2 \Big)$$

$$\leq B_b^2 (B''_\sigma B'_b + B'^2_\sigma B''_b) \frac{p^{\frac{3}{2}}}{\sqrt{m}} \Big( 4\sqrt{p} + 2\sqrt{\log(\frac{m}{\delta})} + \max_{i\in[m]} \|\mathbf{c}_i(0) - \mathbf{c}_i\|_2 \Big), \tag{12}$$

where the second equality used the fact that $\partial_{\mathbf{a}_i}^2 f_\Theta(\mathbf{x})$ is symmetric, the first inequality used Cauchy-Schwarz inequality, the third inequality used Assumption 4.1 that $\sup_{t\in\mathbb{R}}|b(t)| \le B_b$, the last second inequality used (6), and in the last inequality we controlled $\|\mathbf{w}_i\|_2$ by using Assumption 4.1 as follows

$$\|\mathbf{w}_i\|_2 = \left\|\sigma''\left(\frac{1}{\sqrt{d}}\mathbf{a}_i^\top \mathbf{h}(\mathbf{x})\right)\mathbf{h}'\left(\sigma\left(\frac{1}{\sqrt{d}}\mathbf{a}_i^\top \mathbf{h}(\mathbf{x})\right)\right) + \sigma'\left(\frac{1}{\sqrt{d}}\mathbf{a}_i^\top \mathbf{h}(\mathbf{x})\right)^2 \mathbf{h}''\left(\sigma\left(\frac{1}{\sqrt{d}}\mathbf{a}_i^\top \mathbf{h}(\mathbf{x})\right)\right)\right\|_2$$
$$\le \|\sigma''\|_\infty \sqrt{p}\|b'\|_\infty + \|\sigma'\|_\infty^2 \sqrt{p}\|b''\|_\infty \le \sqrt{p}(B_\sigma'' B_b' + B_\sigma'^2 B_b'').$$

Now, we turn to estimate $\|\partial_{\mathbf{c}}\partial_{\mathbf{a}} f_\Theta(\mathbf{x})\|_2$. Note $\partial_{\mathbf{c}}\partial_{\mathbf{a}} f_\Theta(\mathbf{x})$ is a block diagonal matrix which has the form

$$\partial_{\mathbf{c}}\partial_{\mathbf{a}} f_\Theta(\mathbf{x}) = \begin{bmatrix} \partial_{\mathbf{c}_1}\partial_{\mathbf{a}_1} f_\Theta(\mathbf{x}) & \cdots & \mathbf{0} \\ \vdots & \ddots & \vdots \\ \mathbf{0} & \cdots & \partial_{\mathbf{c}_m}\partial_{\mathbf{a}_m} f_\Theta(\mathbf{x}) \end{bmatrix} \in \mathbb{R}^{mpd \times mp},$$

where $\partial_{\mathbf{c}_i}\partial_{\mathbf{a}_i} f_\Theta(\mathbf{x}) = \frac{1}{\sqrt{md}}\sigma'\left(\frac{1}{\sqrt{d}}\mathbf{a}_i^\top \mathbf{h}(\mathbf{x})\right)\mathbf{h}(\mathbf{x})\mathbf{h}'(u_i(\mathbf{x}))^\top \in \mathbb{R}^{pd \times p}$. From Lemma B.3, we know

$$\left\|\partial_{\mathbf{c}}\partial_{\mathbf{a}} f_\Theta(\mathbf{x})\right\|_2 = \max_{i\in[m]}\left\|\partial_{\mathbf{c}_i}\partial_{\mathbf{a}_i} f_\Theta(\mathbf{x})\right\|_2$$
$$= \frac{1}{\sqrt{md}}\max_{i\in[m]}\sup_{\|\mathbf{u}\|_2=\|\mathbf{v}\|_2=1}\sigma'\left(\frac{1}{\sqrt{d}}\mathbf{a}_i^\top \mathbf{h}(\mathbf{x})\right)\langle\mathbf{h}(\mathbf{x}),\mathbf{u}\rangle\langle\mathbf{h}'(u_i(\mathbf{x})),\mathbf{v}\rangle$$
$$\le \frac{1}{\sqrt{md}}\|\sigma'\|_\infty\|\mathbf{h}(\mathbf{x})\|_2\max_{i\in[m]}\|\mathbf{h}'(u_i(\mathbf{x}))\|_2 \le \frac{B_\sigma' B_b B_b' p}{\sqrt{m}}, \tag{13}$$

where the last inequality used Assumption 4.1.

Note the Hessian of $f_\Theta(\mathbf{x})$ has the form

$$\nabla^2 f_\Theta(\mathbf{x}) = \begin{bmatrix} \partial_{\mathbf{c}}^2 f_\Theta(\mathbf{x}) & \partial_{\mathbf{c}}\partial_{\mathbf{a}} f_\Theta(\mathbf{x})^\top \\ \partial_{\mathbf{c}}\partial_{\mathbf{a}} f_\Theta(\mathbf{x}) & \partial_{\mathbf{a}}^2 f_\Theta(\mathbf{x}) \end{bmatrix} = \begin{bmatrix} \mathbf{0} & \partial_{\mathbf{c}}\partial_{\mathbf{a}} f_\Theta(\mathbf{x})^\top \\ \partial_{\mathbf{c}}\partial_{\mathbf{a}} f_\Theta(\mathbf{x}) & \partial_{\mathbf{a}}^2 f_\Theta(\mathbf{x}) \end{bmatrix}.$$

Combining the above estimates for $\|\partial_{\mathbf{c}}\partial_{\mathbf{a}} f_\Theta(\mathbf{x})\|_2$ and $\|\partial_{\mathbf{a}}^2 f_\Theta(\mathbf{x})\|_2$ and the fact that $\nabla^2 f_\Theta(\mathbf{x})$ is symmetric, we know

$$\left\|\nabla^2 f_\Theta(\mathbf{x})\right\|_2 = \sup_{\|\mathbf{v}\|_2=1}\left|\mathbf{v}^\top\nabla^2 f_\Theta(\mathbf{x})\mathbf{v}\right| = \sup_{\|(\mathbf{v}_1,\mathbf{v}_2)\|_2=1}\left|2\mathbf{v}_1^\top\partial_{\mathbf{c}}\partial_{\mathbf{a}} f_\Theta(\mathbf{x})\mathbf{v}_2 + \mathbf{v}_2^\top\partial_{\mathbf{a}}^2 f_\Theta(\mathbf{x})\mathbf{v}_2\right|$$
$$\le 2\left\|\partial_{\mathbf{c}}\partial_{\mathbf{a}} f_\Theta(\mathbf{x})\right\|_2 + \left\|\partial_{\mathbf{a}}^2 f_\Theta(\mathbf{x})\right\|_2 \tag{14}$$
$$\le \frac{C_{\sigma,b}\, p^{\frac{3}{2}}\left(\sqrt{p} + \sqrt{\log(\frac{m}{\delta})} + \max_{i\in[m]}\left\|\mathbf{c}_i(0) - \mathbf{c}_i\right\|_2\right)}{\sqrt{m}}.$$

The proof is complete. $\qquad\square$

Building on Lemma C.1, we establish in the following upper and lower bounds on the largest and smallest eigenvalues of $\nabla^2\mathcal{L}_S(\Theta)$, respectively. These bounds depend on the quantity $\max_{i\in[m]}\|\mathbf{c}_i(0) - \mathbf{c}_i\|_2$. We will show that this term remains controlled along the gradient descent trajectory. Consequently, the loss function $\ell(yf_\Theta(\mathbf{x}))$ is weakly convex and smooth with respect to the variable $\Theta$.

**Proposition C.2** (Smoothness and Curvature). *Let $\delta \in (0,1)$. Suppose (6) and Assumptions 4.1 and 4.2 hold. Assume $m \gtrsim \max\{\log(m/\delta), p\}$. It holds for any $\Theta$ and any training dataset $S$, that*

$$\lambda_{\min}\left(\nabla^2\mathcal{L}_S(\Theta)\right) \ge -\frac{C_{\sigma,b}\, p^{\frac{3}{2}}\left(\sqrt{\log(\frac{m}{\delta})} + \sqrt{p} + R_{\mathbf{c}}\right)}{\sqrt{m}}\mathcal{L}_S(\Theta)$$

*and*

$$\lambda_{\max}\left(\nabla^2\mathcal{L}_S(\Theta)\right) \le C_{\sigma,b}\, p^2\left(p + R_{\mathbf{c}}^2\right),$$

*where $R_{\mathbf{c}} = \max_{i\in[m]}\|\mathbf{c}_i(0) - \mathbf{c}_i\|_2$.*

*Proof.* The gradient of loss is given as

$$\nabla \ell(y f_\Theta(\mathbf{x})) = \ell'(y f_\Theta(\mathbf{x})) y \nabla f_\Theta(\mathbf{x}).$$

For the Hessian of loss, note that

$$\nabla^2 \ell(y f_\Theta(\mathbf{x})) = \ell''(y f_\Theta(\mathbf{x})) \nabla f_\Theta(\mathbf{x}) \nabla f_\Theta(\mathbf{x})^\top + \ell'(y f_\Theta(\mathbf{x})) y \nabla^2 f_\Theta(\mathbf{x}).$$

Since $\ell$ is convex, $\ell''(a) \geq 0$ for all $a \in \mathbb{R}$. Then, $\ell''(y f_\Theta(\mathbf{x})) \nabla f_\Theta(\mathbf{x}) \nabla f_\Theta(\mathbf{x})^\top$ is a PSD matrix. By further noting that $|\ell'(a)| \leq G_\ell = 1$ and $|\ell''(a)| \leq L_\ell = 1$ for all $a \in \mathbb{R}$, we have

$$-|\ell'(y f_\Theta(\mathbf{x}))| \big\| \nabla^2 f_\Theta(\mathbf{x}) \big\|_2 \leq \lambda_{\min}\big(\nabla^2 \ell(y f_\Theta(\mathbf{x}))\big) \leq \lambda_{\max}\big(\nabla^2 \ell(y f_\Theta(\mathbf{x}))\big) \leq \big\| \nabla f_\Theta(\mathbf{x}) \big\|_2^2 + \big\| \nabla^2 f_\Theta(\mathbf{x}) \big\|_2. \quad (15)$$

Plugging the estimates of $\big\| \nabla f_\Theta(\mathbf{x}) \big\|_2$ and $\big\| \nabla^2 f_\Theta(\mathbf{x}) \big\|_2$ (see Lemma C.1) back into (15) and noting that $|\ell'(y f_\Theta(\mathbf{x}))| \leq \ell(y f_\Theta(\mathbf{x}))$, we know

$$\lambda_{\min}\big(\nabla^2 \ell(y f_\Theta(\mathbf{x}))\big) \geq -\frac{C_{\sigma,b}\, p^{\frac{3}{2}}\big(\sqrt{\log(\frac{m}{\delta})} + \sqrt{p} + R_\mathbf{c}\big)}{\sqrt{m}} \ell(y f_\Theta(\mathbf{x})),$$

and

$$\lambda_{\max}\big(\nabla^2 \ell(y f_\Theta(\mathbf{x}))\big) \leq C_{\sigma,b}\, p^2 \Big(p + \sqrt{\frac{\log(\frac{m}{\delta})}{m}} + R_\mathbf{c}^2\Big)$$

with $R_\mathbf{c} = \max_{i \in [m]} \|\mathbf{c}_i(0) - \mathbf{c}_i\|_2$.

By applying the same argument to each sample loss and using that the empirical loss is their average, we obtain,

$$\lambda_{\min}\big(\nabla^2 \mathcal{L}_S(\Theta)\big) \geq -\frac{C_{\sigma,b}\, p^{\frac{3}{2}}\big(\sqrt{\log(\frac{m}{\delta})} + \sqrt{p} + R_\mathbf{c}\big)}{\sqrt{m}} \mathcal{L}_S(\Theta),$$

and from the condition $m \gtrsim \max\{\log(m/\delta), p\}$ we have

$$\lambda_{\max}\big(\nabla^2 \mathcal{L}_S(\Theta)\big) \leq C_{\sigma,b}\, p^2 \big(p + R_\mathbf{c}^2\big).$$

The proof is complete. $\qquad \square$

In the subsequent proof, we work with vectorized quantities, and therefore $\Theta \in \mathbb{R}^{mp(d+1)}$. Let $\{\Theta(k)\}_{k \in [T]}$ be produced by GD with $T$ iterations. By using self-bounding property of the loss $\ell$, we have following estimates of $\mathbf{a}$ and $\mathbf{c}$.

**Lemma C.3.** *Let $\delta \in (0, 1)$. Suppose Assumptions 4.1 and 4.2 hold. Let $\{\Theta(k)\}_{k \in [T]}$ be produced by GD. The following statements hold true.*

*(a) For all $k \in [T - 1]$, we have*

$$\max_{j \in [m]} \big\| \mathbf{c}_j(k+1) - \mathbf{c}_j(0) \big\|_2 \leq \frac{\eta B_b \sqrt{p}}{\sqrt{m}} \sum_{t=0}^{k} \mathcal{L}_S(\Theta(t)).$$

*(b) Let $\delta \in (0, 1)$, assume $m \gtrsim \log(m/\delta)$ and (6) holds. Then, for all $k \in [T - 1]$, we have*

$$\max_{j \in [m]} \big\| \mathbf{a}_j(k+1) - \mathbf{a}_j(0) \big\|_2 \leq \frac{\eta B'_\sigma B_b B'_b p}{\sqrt{m}} \Big(4\sqrt{p} + 2\sqrt{\log(\frac{2m}{\delta})} + \frac{\eta B_b \sqrt{p}}{\sqrt{m}} \sum_{s=0}^{T} \mathcal{L}_S(\Theta(s))\Big) \sum_{t=0}^{k} \mathcal{L}_S(\Theta(t)).$$

*Proof.* We first prove part (a) of the lemma. For any $j \in [m]$, by the GD update rule and the self-bounding property of the loss $\ell$, we have

$$\begin{aligned}
\big\| \mathbf{c}_j(k+1) - \mathbf{c}_j(k) \big\|_2 = \eta \Big\| \frac{\partial \mathcal{L}_S(\Theta(k))}{\partial \mathbf{c}_j} \Big\|_2 &\leq \frac{\eta}{n} \sum_{i=1}^{n} \big|\ell'(y_i f_{\Theta(k)}(\mathbf{x}_i)) y_i\big| \Big\| \frac{\partial f_{\Theta(k)}(\mathbf{x}_i)}{\partial \mathbf{c}_j} \Big\|_2 \\
&= \frac{\eta}{\sqrt{m}} \frac{1}{n} \sum_{i=1}^{n} \big|\ell'(y_i f_{\Theta(k)}(\mathbf{x}_i))\big| \Big\| \mathbf{h}\Big(\frac{1}{\sqrt{d}} \sigma(\mathbf{a}_j^\top \mathbf{h}(\mathbf{x}))\Big) \Big\|_2 \\
&\leq \frac{\eta B_b \sqrt{p}}{\sqrt{m}} \frac{1}{n} \sum_{i=1}^{n} \big|\ell'(y_i f_{\Theta(k)}(\mathbf{x}_i))\big| \leq \frac{\eta B_b \sqrt{p}}{\sqrt{m}} \mathcal{L}_S(\Theta(k)), \quad (16)
\end{aligned}$$

where the second inequality used $\sup_{v \in \mathbb{R}} |b_k(v)| \leq B_b$, and the last inequality follows from the self-bounding property $|\ell'(v)| \leq \ell(v)$ for all $v \in \mathbb{R}$.

Furthermore, by summing over iterations, we obtain

$$\left\|\mathbf{c}_j(k+1) - \mathbf{c}_j(0)\right\|_2 \leq \sum_{t=0}^{k} \left\|\mathbf{c}_j(t+1) - \mathbf{c}_j(t)\right\|_2 \leq \frac{\eta B_b \sqrt{p}}{\sqrt{m}} \sum_{t=0}^{k} \mathcal{L}_S(\Theta(t)). \tag{17}$$

Now we consider part (b) of the lemma. Using the same notation as (8), we denote $u_j(\mathbf{x}) = \sigma(\frac{1}{\sqrt{d}}\mathbf{a}_j^\top \mathbf{h}(\mathbf{x}))$ for any $j \in [m]$ and $\mathbf{x} \in \mathcal{X}$. According to the update rule, $|\ell'(v)| \leq \ell(v)$ and the above inequality, we know for all $j \in [m]$ and $k = 0, \ldots, T-1$, it holds that

$$\begin{aligned}
\left\|\mathbf{a}_j(k+1) - \mathbf{a}_j(k)\right\|_2 &\leq \frac{\eta}{\sqrt{md}}\left\|\frac{1}{n}\sum_{i=1}^{n} \ell'(y_i f_{\Theta(k)}(\mathbf{x}_i))y_i \sigma'\left(\frac{1}{\sqrt{d}}\mathbf{a}_j^\top \mathbf{h}(\mathbf{x}_i)\right) \langle \mathbf{c}_j, \mathbf{h}'(u_j(\mathbf{x}_i))\rangle \mathbf{h}'(u_j(\mathbf{x}_i))\right\|_2 \\
&\leq \frac{\eta}{\sqrt{md}}\frac{1}{n}\sum_{i=1}^{n} \left|\ell'(y_i f_{\Theta(k)}(\mathbf{x}_i))\right| \|\sigma'\|_\infty \|\mathbf{c}_j\|_2 \|\mathbf{h}'(u_j(\mathbf{x}_i))\|_2 \|\mathbf{h}'(u_j(\mathbf{x}_i))\|_2 \\
&\leq \frac{\eta B'_\sigma B_b B'_b p}{\sqrt{m}} \|\mathbf{c}_j(k)\|_2 \mathcal{L}_S(\Theta(k)) \\
&\leq \frac{\eta B'_\sigma B_b B'_b p}{\sqrt{m}} \left(\|\mathbf{c}_j(0)\|_2 + \|\mathbf{c}_j(k) - \mathbf{c}_j(0)\|_2\right)\mathcal{L}_S(\Theta(k)) \\
&\leq \frac{\eta B'_\sigma B_b B'_b p}{\sqrt{m}} \left(4\sqrt{p} + 2\sqrt{\log(\frac{2m}{\delta})} + \frac{\eta B_b \sqrt{p}}{\sqrt{m}} \sum_{t=0}^{k} \mathcal{L}_S(\Theta(t))\right)\mathcal{L}_S(\Theta(k)),
\end{aligned}$$

where we have used Cauchy-Schwarz inequality in the second inequality, and in the third inequality we have used Assumption 4.1 and the self-boundedness property $|\ell'(v)| \leq \ell(v)$, and in the last inequality we have used (6) and (17).

Hence, it holds for all $j \in [m]$ that

$$\begin{aligned}
\left\|\mathbf{a}_j(k+1) - \mathbf{a}_j(0)\right\|_2 &\leq \sum_{t=0}^{k} \left\|\mathbf{a}_j(t+1) - \mathbf{a}_j(t)\right\|_2 \\
&\leq \frac{\eta B'_\sigma B_b B'_b p}{\sqrt{m}} \left(4\sqrt{p} + \sqrt{\log\left(\frac{2m}{\delta}\right)} + \frac{\eta B_b \sqrt{p}}{\sqrt{m}} \sum_{s=0}^{T} \mathcal{L}_S(\Theta(s))\right) \sum_{t=0}^{k} \mathcal{L}_S(\Theta(t)).
\end{aligned}$$

The proof is complete. $\qquad\square$

For any reference point $\Theta^* \in \mathbb{R}^{mp(d+1)}$, let $c_{\max} = C_{\sigma,b}\, p^{\frac{3}{2}}\left(\sqrt{\log(\frac{m}{\delta})} + \sqrt{p} + 3\|\Theta(0) - \Theta^*\|_2\right)$, and $\rho_\ell = C_{\sigma,b}\, p^2\big(p + \max\{\frac{\eta^2 B_b^2 p}{m}\|\Theta(0) - \Theta^*\|_2^4, 3\|\Theta(0) - \Theta^*\|_2^2\}\big)$.

Observe that the only source of randomness in Theorem 4.3 comes from the initialization, and the conclusion holds on the event (6). Therefore, we equivalently restate Theorem 4.3 on the event (6), replacing the statement "with probability at least $1 - \delta$" by the condition (6).

**Theorem C.4** (Restatement of Theorem 4.3). *Let* $\delta \in (0,1)$. *Suppose* (6) *and Assumptions 4.1 and 4.2 hold. For any reference point* $\Theta^* \in \mathbb{R}^{mp(d+1)}$ *satisfies* $\|\Theta(0) - \Theta^*\|_2^2 \geq 4\max\{\eta T \mathcal{L}_S(\Theta^*), \eta \mathcal{L}_S(\Theta(0))\}$, *assume* $m \geq C_{\sigma,b}\, p^3(\log(m/\delta) + p + \|\Theta(0) - \Theta^*\|_2^2)\|\Theta(0) - \Theta^*\|_2^4$, *and* $\eta \leq \min\{1/\rho_\ell, 1\}$, *then we have*

$$\mathcal{L}_S(\Theta(T)) \leq \frac{1}{T}\sum_{k=1}^{T} \mathcal{L}_S(\Theta(k)) \leq 2\mathcal{L}_S(\Theta^*) + \frac{1}{\eta T}\left\|\Theta(0) - \Theta^*\right\|_2^2.$$

*Furthermore, for all* $k \in [T-1]$, *it holds that*

$$\|\Theta(k+1) - \Theta^*\|_2 \leq \sqrt{2}\|\Theta(0) - \Theta^*\|_2 \quad and \quad \|\Theta(k+1) - \Theta(0)\|_2 \leq 3\|\Theta(0) - \Theta^*\|_2.$$

*Proof.* We use the induction strategy to prove the following statements. It holds for any $k \in [T]$ that

$$\eta \sum_{s=1}^{k} \mathcal{L}_S(\Theta(s)) \leq \frac{7}{5}\eta k \mathcal{L}_S(\Theta^*) + \frac{3}{5}\|\Theta(0) - \Theta^*\|_2^2 + \frac{\eta}{5}\mathcal{L}_S(\Theta(0)),$$

$$\|\Theta(k) - \Theta^*\|_2 \leq \sqrt{2}\|\Theta(0) - \Theta^*\|_2 \quad \text{and} \quad \|\Theta(k) - \Theta(0)\|_2 \leq 3\|\Theta(0) - \Theta^*\|_2.$$

Note that the statements hold with $k = 0$. Here, we take the conventional notation $\sum_{s=1}^{0} = 0$. We assume that the above statements hold for all $t \in [k]$ and will prove that they also hold for $t = k + 1$.

For any fixed $\Theta^* = (\mathbf{a}^*, \mathbf{c}^*)$, $t \in [k]$ and $\alpha \in [0, 1]$, we denote $\Theta_{\alpha,t} = \alpha\Theta(t) + (1-\alpha)\Theta^*$ and $\mathbf{c}_{\alpha,t} = \alpha\mathbf{c}(t) + (1-\alpha)\mathbf{c}^*$ as the convex combination of the reference point and the output of the $t$-th iteration. According to the induction assumption, we know $R_{\mathbf{c}} = \max_{i \in [m]} \|\mathbf{c}_i(0) - [\mathbf{c}_{\alpha,t}]_i\|_2 \leq \max\{\|\mathbf{c}(t) - \mathbf{c}(0)\|_2, \|\mathbf{c}^* - \mathbf{c}(0)\|_2\} \leq \max\{\|\Theta(t) - \Theta(0)\|_2, \|\Theta^* - \Theta(0)\|_2\} \leq 3\|\Theta(0) - \Theta^*\|_2$. Then, from Proposition C.2 we know the least eigenvalue of $\nabla^2 \mathcal{L}_S(\Theta_{\alpha,t})$ can be controlled as follows.

$$\lambda_{\min}\left(\nabla^2 \mathcal{L}_S(\Theta_{\alpha,t})\right) \geq -\frac{c_{\max}}{\sqrt{m}}\mathcal{L}_S(\Theta_{\alpha,t}). \tag{18}$$

where $c_{\max} = C_{\sigma,b}\, p^{\frac{3}{2}}\left(\sqrt{\log(\frac{m}{\delta})} + \sqrt{p} + 3\|\Theta(0) - \Theta^*\|_2\right)$.

Then from Taylor's theorem and (18), we know that there exists a $\alpha \in [0, 1]$ and the associated $\Theta_{\alpha,t}$ such that with probability at least $1 - \delta$, it holds that

$$\mathcal{L}_S(\Theta^*) = \mathcal{L}_S(\Theta(t)) + \langle \nabla \mathcal{L}_S(\Theta(t)), \Theta^* - \Theta(t)\rangle + \frac{1}{2}\left(\Theta(t) - \Theta^*\right)^{\top} \nabla^2 \mathcal{L}_S(\Theta_{\alpha,t})\left(\Theta(t) - \Theta^*\right)$$

$$\geq \mathcal{L}_S(\Theta(t)) + \langle \nabla \mathcal{L}_S(\Theta(t)), \Theta^* - \Theta(t)\rangle - \frac{c_{\max}}{2\sqrt{m}}\mathcal{L}_S(\Theta_{\alpha,t})\|\Theta(t) - \Theta^*\|_2^2.$$

It then follows that

$$\mathcal{L}_S(\Theta(t)) \leq \mathcal{L}_S(\Theta^*) - \langle \nabla \mathcal{L}_S(\Theta(t)), \Theta^* - \Theta(t)\rangle + \frac{c_{\max}}{2\sqrt{m}}\max_{\alpha \in [0,1]} \mathcal{L}_S(\Theta_{\alpha,t})\|\Theta(t) - \Theta^*\|_2^2. \tag{19}$$

Now, we turn to estimate the largest eigenvalue of $\nabla^2 \mathcal{L}_S(\Theta_{\alpha,t})$. For all $t \in [k]$, from Proposition C.2, Lemma C.3 and the induction assumption on the upper bound of $\eta \sum \mathcal{L}_S(\Theta(t))$, it holds that

$$\lambda_{\max}\left(\nabla^2 \mathcal{L}_S(\Theta(t+1))\right)$$

$$\leq C_{\sigma,b}\, p^2\left(p + \max_{i \in [m]}\{\mathbf{c}_i(0) - \mathbf{c}_i(t+1)\}^2\right)$$

$$\leq C_{\sigma,b}\, p^2\left(p + \frac{B_b^2 p}{m}\left(\eta \sum_{s=0}^{t} \mathcal{L}_S(\Theta(s))\right)^2\right)$$

$$\leq C_{\sigma,b}\, p^2\left(p + \frac{B_b^2 p}{m}\left(\frac{7}{5}\eta k \mathcal{L}_S(\Theta^*) + \frac{3}{5}\|\Theta(0) - \Theta^*\|_2^2 + \frac{\eta}{5}\mathcal{L}_S(\Theta(0))\right)^2\right).$$

Note that the condition $\|\Theta(0) - \Theta^*\|_2^2 \geq 4\max\{\eta T \mathcal{L}_S(\Theta^*), \eta \mathcal{L}_S(\Theta(0))\}$ implies the estimate $\frac{7}{5}\eta T \mathcal{L}_S(\Theta^*) + \frac{7}{10}\|\Theta(0) - \Theta^*\|_2^2 + \frac{6\eta}{5}\mathcal{L}_S(\Theta(0)) \leq \|\Theta(0) - \Theta^*\|_2^2$. Then, the above inequality can be further bounded as follows

$$\lambda_{\max}\left(\nabla^2 \mathcal{L}_S(\Theta(t+1))\right) \leq C_{\sigma,b}\, p^2\left(p + \frac{B_b^2 p}{m}\|\Theta(0) - \Theta^*\|_2^4\right) \leq C_{\sigma,b} p^3 =: \rho_\ell,$$

where the last inequality used the condition $m \gtrsim \|\Theta(0) - \Theta^*\|_2^4$. Then we know $\ell(\Theta(t+1))$ is $\rho_\ell$-strongly smooth through the trajectory of $\{\Theta(t)\}_{t \in [k]}$. From the descent lemma (see e.g., (Taheri & Thrampoulidis, 2024)), we know if $\eta \leq 1/\rho_\ell$, it holds for any $t \in [k]$ that

$$\mathcal{L}_S(\Theta(t+1)) \leq \mathcal{L}_S(\Theta(t)) - \frac{\eta}{2}\|\nabla \mathcal{L}_S(\Theta(t))\|_2^2. \tag{20}$$

Combining the above inequality and (19), we know

$$\mathcal{L}_S(\Theta(t+1))$$
$$\leq \mathcal{L}_S(\Theta^*) - \langle \nabla \mathcal{L}_S(\Theta(t)), \Theta^* - \Theta(t) \rangle - \frac{\eta}{2} \|\nabla \mathcal{L}_S(\Theta(t))\|_2^2 + \frac{c_{\max}}{2\sqrt{m}} \max_{\alpha \in [0,1]} \mathcal{L}_S(\Theta_{\alpha,t}) \|\Theta(t) - \Theta^*\|_2^2$$
$$= \mathcal{L}_S(\Theta^*) + \frac{1}{2\eta} \left( \|\Theta(t) - \Theta^*\|_2^2 - \|\Theta(t+1) - \Theta^*\|_2^2 \right) + \frac{c_{\max}}{2\sqrt{m}} \max_{\alpha \in [0,1]} \mathcal{L}_S(\Theta_{\alpha,t}) \|\Theta(t) - \Theta^*\|_2^2, \tag{21}$$

where the last equality used the fact $\nabla \mathcal{L}_S(\Theta(t)) = \frac{1}{\eta}(\Theta(t) - \Theta(t+1))$ and the identity $2\langle \mathbf{u}, \mathbf{v} \rangle = \|\mathbf{u}\|_2^2 + \|\mathbf{v}\|_2^2 - \|\mathbf{u} - \mathbf{v}\|_2^2$.

Now, we turn to estimate $\max_{\alpha \in [0,1]} \mathcal{L}_S(\Theta_{\alpha,t})$. From the the induction assumption $\|\Theta(t) - \Theta^*\|_2 \leq \sqrt{2}\|\Theta(0) - \Theta^*\|_2$ for all $t = 0, \ldots, k$, the condition $m \geq 64 c_{\max}^2 \|\Theta(0) - \Theta^*\|_2^4$ (implied by $m \gtrsim p^3(\log(m/\delta) + p + \|\Theta(0) - \Theta^*\|_2^2)\|\Theta(0) - \Theta^*\|_2^4$), we know

$$\|\Theta(t) - \Theta^*\|_2^2 \leq \frac{\sqrt{m}}{4c_{\max}}. \tag{22}$$

Therefore, Lemma B.4 with $D^2 = \frac{\sqrt{m}}{4c_{\max}}$ and $\kappa = \frac{c_{\max}}{\sqrt{m}}$ (hence $D^2 \kappa \leq 1/4$) implies, for all $t = 0, \ldots, k$

$$\max_{\alpha \in [0,1]} \mathcal{L}_S(\Theta_{\alpha,t}) \leq \frac{4}{3} \max \left\{ \mathcal{L}_S(\Theta(t)), \mathcal{L}_S(\Theta^*) \right\} \leq \frac{4}{3} \left( \mathcal{L}_S(\Theta(t)) + \mathcal{L}_S(\Theta^*) \right).$$

Plugging the above inequality and (22) back into (21) yields, for any $t = 0, \ldots, k$, we have

$$\mathcal{L}_S(\Theta(t+1)) \leq \mathcal{L}_S(\Theta^*) + \frac{1}{2\eta} \left( \|\Theta(t) - \Theta^*\|_2^2 - \|\Theta(t+1) - \Theta^*\|_2^2 \right) + \frac{2c_{\max}}{3\sqrt{m}} \left( \mathcal{L}_S(\Theta(t)) + \mathcal{L}_S(\Theta^*) \right) \|\Theta(t) - \Theta^*\|_2^2$$
$$\leq \mathcal{L}_S(\Theta^*) + \frac{1}{2\eta} \left( \|\Theta(t) - \Theta^*\|_2^2 - \|\Theta(t+1) - \Theta^*\|_2^2 \right) + \frac{1}{6} \left( \mathcal{L}_S(\Theta(t)) + \mathcal{L}_S(\Theta^*) \right) \tag{23}$$

Summing over $t \in \{0, \ldots, k\}$ yields

$$\sum_{t=1}^{k+1} \mathcal{L}_S(\Theta(t)) \leq (k+1)\mathcal{L}_S(\Theta^*) + \frac{1}{2\eta} \|\Theta(0) - \Theta^*\|_2^2 + \frac{1}{6} \sum_{t=0}^{k} \left( \mathcal{L}_S(\Theta(t)) + \mathcal{L}_S(\Theta^*) \right)$$
$$= \frac{7}{6}(k+1)\mathcal{L}_S(\Theta^*) + \frac{1}{2\eta} \|\Theta(0) - \Theta^*\|_2^2 + \frac{1}{6} \sum_{t=0}^{k} \mathcal{L}_S(\Theta(t)).$$

It then follows that

$$\eta \sum_{t=1}^{k+1} \mathcal{L}_S(\Theta(t)) \leq \frac{7}{5} \eta (k+1) \mathcal{L}_S(\Theta^*) + \frac{3}{5} \|\Theta(0) - \Theta^*\|_2^2 + \frac{\eta}{5} \mathcal{L}_S(\Theta(0)),$$

which completes the first part of the induction.

It remains to show $\|\Theta(k+1) - \Theta^*\|_2 \leq \sqrt{2}\|\Theta(0) - \Theta^*\|_2$ and $\|\Theta(k+1) - \Theta(0)\|_2 \leq 3\|\Theta(0) - \Theta^*\|_2$. Note we have the estimate

$$\|\Theta(k+1) - \Theta^*\|_2^2 \leq \|\Theta(k) - \Theta^*\|_2^2 + \frac{7}{3}\eta \mathcal{L}_S(\Theta^*) + \frac{\eta}{3}\mathcal{L}_S(\Theta(k))$$
$$\leq \|\Theta(0) - \Theta^*\|_2^2 + \frac{7}{3}\eta k \mathcal{L}_S(\Theta^*) + \frac{\eta}{3} \sum_{t=0}^{k} \mathcal{L}_S(\Theta(t))$$
$$\leq \frac{6}{5}\|\Theta(0) - \Theta^*\|_2^2 + \frac{14}{5}\eta k \mathcal{L}_S(\Theta^*) + \frac{\eta}{15}\mathcal{L}_S(\Theta(0))$$
$$\leq 2\|\Theta(0) - \Theta^*\|_2^2, \tag{24}$$

where the first inequality used (23) with $t = k$, and in the second inequality we used the first inequality recursively, and the last second inequality is according to the induction assumption $\eta \sum_{s=1}^{k} \mathcal{L}_S(\Theta(s)) \leq \frac{7}{5}\eta k \mathcal{L}_S(\Theta^*) + \frac{3}{5}\|\Theta(0) - \Theta^*\|_2^2 + \frac{\eta}{5}\mathcal{L}_S(\Theta(0))$, and in the last inequality we have used $\|\Theta(0) - \Theta^*\|_2^2 \geq 4 \max\{\eta T \mathcal{L}_S(\Theta^*), \eta \mathcal{L}_S(\Theta(0))\}$ and $\eta \leq 1$.

The second part of the induction is proved by noting that

$$\|\Theta(k+1) - \Theta(0)\|_2 \leq \|\Theta(k+1) - \Theta^*\|_2 + \|\Theta^* - \Theta(0)\|_2 \leq 3\|\Theta(0) - \Theta^*\|_2.$$

Finally, note (20) implies $\mathcal{L}_S(\Theta(t+1)) \leq \mathcal{L}_S(\Theta(t))$ holds for all $t \in [T-1]$, then

$$\mathcal{L}_S(\Theta(T)) \leq \frac{1}{T} \sum_{k=1}^{T} \mathcal{L}_S(\Theta(k)) \leq \frac{6}{5}\mathcal{L}_S(\Theta^*) + \frac{3}{5\eta T}\|\Theta(0) - \Theta^*\|_2^2 + \frac{1}{5T}\mathcal{L}_S(\Theta(0)) \tag{25}$$

$$\leq 2\mathcal{L}_S(\Theta^*) + \frac{1}{\eta T}\|\Theta(0) - \Theta^*\|_2^2,$$

which in the last inequality we have used the condition $\|\Theta(0) - \Theta^*\|_2^2 \geq 4\eta \mathcal{L}_S(\Theta(0))$. The proof is completed. $\qquad \square$

**Assumption C.5** (Weak Realizability). For any $\epsilon > 0$, there exists $\Theta^\epsilon$ such that $\mathcal{L}_S(\Theta^\epsilon) \leq \epsilon$.

**Lemma C.6.** *Suppose Assumption C.5 holds. For any $\epsilon > 0$, one can define a non-increasing function $g : \mathbb{R}_+ \to \mathbb{R}_+$ such that there exists $\Theta^\epsilon$ satisfying*

$$\mathcal{L}_S(\Theta^\epsilon) \leq \epsilon \quad and \quad \|\Theta^\epsilon - \Theta(0)\|_2 = g(\epsilon).$$

*As a corollary, Assumption C.5 implies Assumption 4.5.*

*Proof.* Define $\Omega_n := \{\Theta : \mathcal{L}_S(\Theta) \leq 1/n\}$ and $\Omega_0 := \mathbb{R}^{mp(d+1)}$. Then, $\Omega_n \subset \Omega_{n-1}$ for all $n \in \mathbb{N}$. Further, by the continuity of $\mathcal{L}_S(\cdot)$, we know each $\Omega_n$ is closed. Then, there exists $\Theta_n$ satisfying $\Theta_n = \arg\min_{\Theta \in \Omega_n} \|\Theta - \Theta(0)\|_2$. For any $\epsilon$, there exists $n \in \mathbb{N}$ such that $\epsilon \in [\frac{1}{n}, \frac{1}{n-1})$ (we take $\frac{1}{0} = \infty$). Setting $\Theta^\epsilon = \Theta_n$, we know $\mathcal{L}_S(\Theta^\epsilon) = \mathcal{L}_S(\Theta_n) \leq \frac{1}{n} \leq \epsilon$. Then, for any $\epsilon$ and the associated $n$, we know

$$g(\epsilon) = \|\Theta_n - \Theta(0)\|_2 = \min_{\Theta \in \Omega_n} \|\Theta - \Theta(0)\|_2.$$

Now we show $g(\epsilon)$ is non-increasing. Specifically, for any $0 < \epsilon_1 \leq \epsilon_2$. For the case $\epsilon_1, \epsilon_2 \in (\frac{1}{n+1}, \frac{1}{n}]$ for some $n$, we know $g(\epsilon_1) = g(\epsilon_2) = \min_{\Theta \in \Omega_n} \|\Theta - \Theta(0)\|_2$. For the case $\frac{1}{n_1+1} < \epsilon_1 \leq \frac{1}{n_1} \leq \frac{1}{n_2+1} < \epsilon_2 \leq \frac{1}{n_2}$ for some $n_1 > n_2$, we know

$$g(\epsilon_1) = \min_{\Theta \in \Omega_{n_1}} \|\Theta - \Theta(0)\|_2 \geq \min_{\Theta \in \Omega_{n_2}} \|\Theta - \Theta(0)\|_2 = g(\epsilon_2),$$

where the inequality follows from $\Omega_{n_1} \subset \Omega_{n_2}$. This completes the proof of the first part of the lemma.

Note that the condition $\|\Theta^\epsilon - \Theta(0)\|_2 = g(\epsilon)$ implies $\|\Theta^\epsilon - \Theta(0)\|_2 \leq g(\epsilon)$. Then, Assumption 4.5 holds. $\qquad \square$

Now, we give the proof of Theorem 4.7 as follows. Observe that the only source of randomness in Theorem 4.7 comes from the initialization, and the conclusion holds on the event (6). Therefore, we may equivalently restate Theorem 4.7 on the event (6), replacing the statement "with probability at least $1 - \delta$" by the condition (6).

**Theorem C.7** (Restatement of Theorem 4.7). *Let (6) and Assumptions 4.1, 4.2 and 4.5 hold. Assume $\eta \leq \min\{g^2(1), g^2(1)(\mathcal{L}_S(\Theta(0)))^{-1}\}$, $m \gtrsim \left(\log(m/\delta) + g^2(\frac{1}{T})\right)g^4(\frac{1}{T})$. Then, it holds that $\|\Theta(k) - \Theta^{\frac{1}{T}}\|_2^2 \leq 4g^2(\frac{1}{T})$ and*

$$\mathcal{L}_S(\Theta(T)) \leq \frac{1}{T} \sum_{k=1}^{T} \mathcal{L}_S(\Theta(k)) \lesssim \frac{2\eta + g^2(\frac{1}{T})}{\eta T}.$$

*Proof.* Assumption 4.5 with $\epsilon = 1/T$ implies that there exists a non-increasing function $g : \mathbb{R}_+ \mapsto \mathbb{R}_+$ such that there exists $\Theta^{1/T}$ with

$$\mathcal{L}_S(\Theta^{1/T}) \leq \frac{1}{T} \quad and \quad \|\Theta(0) - \Theta^{1/T}\|_2 = g\left(\frac{1}{T}\right) \geq g(1).$$

Recall that we denote $\Lambda_{\Theta^{1/T}} = \|\Theta(0) - \Theta^{1/T}\|_2$ and $\mathfrak{C}_S(\Theta^{1/T}) = 2\eta T + \|\Theta(0) - \Theta^{1/T}\|_2^2$. Then, the conditions $m \gtrsim (\log(m/\delta) + g^2(1/T))g^4(1/T)$ and $\eta \lesssim \min\{g^2(1), g^2(1)(\mathcal{L}_S(\Theta(0)))^{-1}\}$ ensure that all the conditions in Theorem 4.3 are satisfied. Applying Theorem 4.3 with $\Theta^* = \Theta^{1/T}$ implies that

$$\mathcal{L}_S(\Theta(T)) \lesssim \frac{2\eta T \mathcal{L}_S(\Theta^{1/T}) + \Lambda_{\Theta^{1/T}}}{\eta T} \leq \frac{2\eta + g^2(\frac{1}{T})}{\eta T}.$$

This completes the proof of the Theorem. □

The proof of Theorem 4.9 is given as follows.

*Proof of Theorem 4.9.* We first bound the values of $f_{\Theta(0)}(\cdot)$ over the training dataset $S$. Recall that we defined $\mathbf{h}(v) = [b_1(v), \ldots, b_p(v)]^\top \in \mathbb{R}^p$ for any scalar $v \in \mathbb{R}$ and $\mathbf{h}(\mathbf{u}) = [\mathbf{h}(u_1)^\top, \ldots, \mathbf{h}(u_s)^\top]^\top \in \mathbb{R}^{sp}$ for vector $\mathbf{u} = [u_1, \ldots, u_s]^\top \in \mathbb{R}^s$. Note that

$$f_{\Theta(0)}(\mathbf{x}) = \frac{1}{\sqrt{m}}\mathbf{c}(0)^\top \mathbf{h}\left(\sigma\left(\frac{1}{\sqrt{d}}\mathbf{A}\mathbf{h}(\mathbf{x})\right)\right) = \frac{1}{\sqrt{m}}\sum_{i=1}^m \mathbf{c}_i(0)^\top \mathbf{h}\left(\sigma(\mathbf{a}_i^\top \mathbf{h}(\mathbf{x}))/\sqrt{d}\right).$$

Denote $\mathbf{h}_{i,j} = \mathbf{h}\left(\sigma(\mathbf{a}_i^\top \mathbf{h}(\mathbf{x}_j))/\sqrt{d}\right)$ for all $j \in [n]$. Then, from the fact $\mathbf{c}_i(0) \sim \mathcal{N}(0, \mathbf{I}_p)$ we know

$$f_{\Theta(0)}(\mathbf{x}_j) = \frac{1}{\sqrt{m}}\sum_{i=1}^m \mathbf{c}_i(0)^\top \mathbf{h}_{i,j} \sim \mathcal{N}\left(0, \frac{\sum_{i=1}^m \|\mathbf{h}_{i,j}\|_2^2}{m}\right).$$

Then, $f_{\Theta(0)}(\mathbf{x}_j)/(\sum_{i=1}^m \|\mathbf{h}_{i,j}\|_2^2/m)^{1/2} \sim \mathcal{N}(0,1)$ for all $j \in [n]$. According to (2.3) in (Vershynin, 2018), we know with probability at least $1 - \delta/2$ over $\mathbf{c}(0)$, it holds for all $j \in [n]$ that

$$|f_{\Theta(0)}(\mathbf{x}_j)| \leq \sqrt{2\log\left(\frac{2n}{\delta}\right)}\left(\frac{\sum_{i=1}^m \|\mathbf{h}_{i,j}\|_2^2}{m}\right)^{\frac{1}{2}} \leq B_b\sqrt{2p\log\left(\frac{2n}{\delta}\right)}, \tag{26}$$

where the last inequality used Assumption 4.1. Let $\Theta_0 \in \mathbb{R}^{m(d+1)p}$ be the parameter that satisfies the NTK separable assumption (see Assumption 4.8). Take $\tau \asymp \left(\log(T) + \sqrt{\log(n/\delta)}\right)/\gamma$, we denote $\Theta_\tau = \Theta(0) + \tau\Theta_0 = [\mathbf{c}_\tau^\top, \mathbf{a}_\tau^\top]^\top$. For any $j \in [n]$, from Taylor's theorem we know there is a $\Theta_\tau' \in [\Theta(0), \Theta_\tau]$, it holds that

$$y_j f_{\Theta_\tau}(\mathbf{x}_j) = y_j f_{\Theta(0)}(\mathbf{x}_j) + y_j\langle\nabla f_{\Theta(0)}(\mathbf{x}_j), \Theta_\tau - \Theta(0)\rangle_2 + \frac{1}{2}y_j(\Theta_\tau - \Theta(0))^\top \nabla^2 f_{\Theta_\tau'}(\mathbf{x}_j)(\Theta_\tau - \Theta(0))$$

$$\geq -|f_{\Theta(0)}(\mathbf{x}_j)| + \tau\, y_j\langle\nabla f_{\Theta(0)}(\mathbf{x}_j), \Theta_0\rangle_2 - \frac{\tau^2}{2}\|\nabla^2 f_{\Theta_\tau'}(\mathbf{x}_j)\|_2$$

$$\geq \tau\gamma - B_b\sqrt{2p\log\left(\frac{n}{\delta}\right)} - \frac{\tau^2}{2}\|\nabla^2 f_{\Theta_\tau'}(\mathbf{x}_j)\|_2$$

$$\geq \tau\gamma - B_b\sqrt{2p\log\left(\frac{n}{\delta}\right)} - \frac{\tau^2}{2}\frac{C_{\sigma,b}\,p^{\frac{3}{2}}\left(\sqrt{p} + \sqrt{\log(m/\delta)} + \tau\right)}{\sqrt{m}}$$

$$\geq \frac{\tau\gamma}{2} - \frac{\tau^2}{2}\frac{C_{\sigma,b}\,p^{\frac{3}{2}}\left(\sqrt{p} + \sqrt{\log(m/\delta)} + \tau\right)}{\sqrt{m}}$$

$$\geq \frac{\tau\gamma}{4} \gtrsim \log(T),$$

where the first inequality used Assumption 4.8 with $\|\Theta_0\|_2 = 1$, the second inequality again used Assumption 4.8 and the estimate of $|f_{\Theta(0)}(\mathbf{x}_j)|$ from above, the third inequality used (15) with $|\ell'(\cdot)| \leq 1$ and the fact $\max_{i\in[m]}\|(\mathbf{c}_\tau)_i - \mathbf{c}_i(0)\|_2 \leq \|\Theta_\tau - \Theta(0)\|_2 = \tau\|\Theta_0\|_2$, and the fourth inequality used the fact $\tau \asymp \left(\log(T) + \sqrt{\log(n/\delta)}\right)/\gamma$, and in the last second inequality we have used the condition $m \gtrsim \log(m/\delta)\left(\log^6(T) + \log^3(n/\delta)\right)/\gamma^6 \gtrsim \left(\sqrt{\log(m/\delta)} + \tau\right)^2\tau^2/\gamma^2$.

Then, for all $j \in [n]$, it holds that

$$\ell\left(y_j f_{\Theta_\tau}(\mathbf{x}_j)\right) = \log\left(1 + \exp\left(-y_j f_{\Theta_\tau}(\mathbf{x}_j)\right)\right) \leq \exp\left(-y_j f_{\Theta_\tau}(\mathbf{x}_j)\right) \leq \exp(-\log(T)) = \frac{1}{T}.$$

Then, we know $\mathcal{L}_S(\Theta_\tau) \leq \frac{1}{T}$ and the corresponding $g(1/T) = \|\Theta_\tau - \Theta(0)\|_2 = \tau \asymp \left(\log(T) + \sqrt{\log(n/\delta)}\right)/\gamma$. Then, we know Assumption 4.5 holds with $\epsilon = 1/T$. Plugging the estimate of $g(1/T)$ back into Theorem 4.7, and note that (6) with $\delta$ replaced by $\delta/2$ holds with probability at least $1 - \delta/2$ over initialization $\mathbf{c}(0)$, we can obtain the desired results. □

## D. Proofs for Generalization

To establish generalization results of GD, we introduce the concept of algorithmic stability. For a randomized algorithm $\mathcal{A}$, let $\mathcal{A}(S) \in \mathbb{R}^{mp(d+1)}$ be the output of $\mathcal{A}$ based on dataset $S$. The on-average argument stability measures the on-average sensitivity of the output up to the perturbation of the dataset.

**Definition D.1** (On-average argument stability (Lei & Ying, 2020)). Let $S = \{z_1, \ldots, z_n\}$ and $\widetilde{S} = \{z_1', \ldots, z_n'\}$ be drawn independently from $\mathcal{P}$. For any $i \in [n]$, define $S^i = \{z_1, \ldots, z_{i-1}, z_i', z_{i+1}, \ldots, z_n\}$. Let $\mathcal{A}(S)$ and $\mathcal{A}(S^i)$ be produced by an randomized algorithm $\mathcal{A}$ based on $S$ and $S^i$ respectively. We say $\mathcal{A}$ is on-average argument $\epsilon$-stable if

$$\mathbb{E}_{S, \widetilde{S}, \mathcal{A}} \left[ \frac{1}{n} \sum_{i=1}^{n} \|\mathcal{A}(S) - \mathcal{A}(S^i)\|_2 \right] \leq \epsilon.$$

We consider using the connection between the on-average argument stability and generalization error bounds (Lei & Ying, 2020).

**Lemma D.2** ((Lei & Ying, 2020)). *If $\mathcal{A}$ is on-average argument $\epsilon$-stable and the loss $\ell$ is L-Lipschitz with respect to $\mathcal{A}(S)$, then*

$$\mathbb{E}_{S, \mathcal{A}} \left[ \mathcal{L}(\mathcal{A}(S)) - \mathcal{L}_S(\mathcal{A}(S)) \right] \leq 2L\epsilon.$$

We will first estimate stability bound of GD and then using the above connection to obtain the generalization error bounds. Our stability analysis requires the following lemma.

**Lemma D.3** (Expansiveness of GD). *Suppose Assumptions 4.1 and 4.2 hold. Let $\alpha \in [0, 1]$. For all $\Theta, \Theta' \in \mathbb{R}^{mp(d+1)}$, we denote $\Theta_\alpha = \alpha\Theta + (1 - \alpha)\Theta'$. Then, it holds for any training dataset $S$ that*

$$\left\| \Theta - \eta\nabla\mathcal{L}_S(\Theta) - \left(\Theta' - \eta\nabla\mathcal{L}_S(\Theta')\right) \right\|_2 \leq \max_{\alpha \in [0,1]} \left\{ 1 - \eta\lambda_{\min}(\nabla^2\mathcal{L}_S(\Theta_\alpha)), \eta\lambda_{\max}(\nabla^2\mathcal{L}_S(\Theta_\alpha)) \right\} \left\| \Theta - \Theta' \right\|_2.$$

*Proof.* The proof can be directly obtained by Lemma B.1 in (Taheri & Thrampoulidis, 2024). Specifically, similar to (32) of their Lemma B.1, one can show that

$$\left\| \Theta - \eta\nabla\mathcal{L}_S(\Theta) - \left(\Theta' - \eta\nabla\mathcal{L}_S(\Theta')\right) \right\|_2 \leq \max_{\alpha \in [0,1]} \left\| \mathbf{I} - \eta\nabla^2\mathcal{L}_S(\Theta_\alpha) \right\|_{op} \left\| \Theta - \Theta' \right\|_2.$$

Note that

$$\max_{\alpha \in [0,1]} \left\| \mathbf{I} - \eta\nabla^2\mathcal{L}_S(\Theta_\alpha) \right\|_{op} \leq \max \left\{ \left| 1 - \eta\lambda_{\min}(\nabla^2\mathcal{L}_S(\Theta_\alpha)) \right|, \left| 1 - \eta\lambda_{\max}(\nabla^2\mathcal{L}_S(\Theta_\alpha)) \right| \right\}$$

$$\leq \max \left\{ 1 - \eta\lambda_{\min}(\nabla^2\mathcal{L}_S(\Theta_\alpha)), \eta\lambda_{\max}(\nabla^2\mathcal{L}_S(\Theta_\alpha)) \right\}.$$

Combining the above two observations and noting that (6) holds with probability at least $1 - \delta$ over initialization $\mathbf{c}(0)$ complete the proof of the theorem. $\square$

Recall that $c_{\max} = C_{\sigma,b} p^{\frac{3}{2}} \left( \sqrt{\log(\frac{m}{\delta})} + \sqrt{p} + 3\|\Theta(0) - \Theta^*\|_2 \right)$ and $\rho_\ell = C_{\sigma,b} p^2 (p + 3\|\Theta(0) - \Theta^*\|_2^2)$. The following lemma gives the on-average argument stability bound of GD. In this section, we assume the reference model $\Theta^*$ is independent of $S$.

Observe that the only source of randomness in Theorem 4.10 comes from the initialization, and the conclusion holds on the event (6). Therefore, we may equivalently restate Theorem 4.10 on the event (6), replacing the statement "with probability at least $1 - \delta$" by the condition (6).

**Lemma D.4** (Restatement of Theorem 4.10). *Let $\delta \in (0, 1)$. Suppose (6), 4.1 and 4.2 hold. For any reference point $\Theta^* \in \mathbb{R}^{mp(d+1)}$ satisfies $\|\Theta(0) - \Theta^*\|_2^2 \geq 8 \max \left\{ \eta T \left( \mathcal{L}_S(\Theta^*) + \mathcal{L}_{\widetilde{S}}(\Theta^*) \right), \eta \left( \mathcal{L}_S(\Theta(0)) + \mathcal{L}_{\widetilde{S}}(\Theta(0)) \right) \right\}$, suppose $\eta \leq \min\{1/2\rho_\ell, 1\}$, and*

$$m \gtrsim p^3 (\log(\frac{m}{\delta}) + p + \|\Theta(0) - \Theta^*\|_2^2) \max \left\{ \|\Theta(0) - \Theta^*\|_2^2, \|\Theta(0) - \Theta^*\|_2^4 \right\}.$$

*Then, it holds that*

$$\mathbb{E}_S \left[ \mathcal{L}(\Theta(T)) - \mathcal{L}_S(\Theta(T)) \right] \leq C_{\sigma,b} p^2 \left( p + p\|\Theta^* - \Theta(0)\|_2^2 \right) \frac{\eta}{n} \mathbb{E}_S \left[ \sum_{t=0}^{T} \mathcal{L}_S(\Theta(t)) \right].$$

*Proof.* For any $i \in [n]$, let $\{\Theta^i(k)\}$ and $\{\Theta^{-i}(k)\}$ be produced by GD based $S^i$ and $S^{-i}$, respectively. Observe that the condition $\|\Theta(0) - \Theta^*\|_2^2 \geq 8 \max\{\eta T(\mathcal{L}_S(\Theta^*) + \mathcal{L}_{\widetilde{S}}(\Theta^*)), \eta(\mathcal{L}_S(\Theta(0)) + \mathcal{L}_{\widetilde{S}}(\Theta(0)))\}$ implies $\|\Theta(0) - \Theta^*\|_2^2 \geq 4 \max\{\eta T\mathcal{L}_{S^i}(\Theta^*), \eta T\mathcal{L}_{S^{-i}}(\Theta^*), \eta\mathcal{L}_{S^i}(\Theta(0)), \eta\mathcal{L}_{S^{-i}}(\Theta(0))\}$ for all $i \in [n]$. Then, Theorem C.4 can be applied to any $S^i$ and $S^{-i}$ and the associated GD outputs $\Theta^i(k)$ and $\Theta^{-i}(k)$.

For any $i \in [n]$, define $S^{-i} = \{z_1, \ldots, z_{i-1}, z_{i+1}, \ldots, z_n\} = S \backslash \{z_i\}$. From the update rule of GD and the self-bounding property of $\ell$, for any $k = 1, \ldots, T-1$, it holds

$$\left\|\Theta(k+1) - \Theta^i(k+1)\right\|_2$$
$$= \left\|\Theta(k) - \eta\nabla\mathcal{L}_S(\Theta(k)) - \Theta^i(k) + \eta\nabla\mathcal{L}_{S^i}(\Theta^i(k))\right\|_2$$
$$\leq \left\|\Theta(k) - \frac{\eta(n-1)}{n}\nabla\mathcal{L}_{S^{-i}}(\Theta(k)) - \Theta^i(k) + \frac{\eta(n-1)}{n}\nabla\mathcal{L}_{S^{-i}}(\Theta^i(k))\right\|_2 + \frac{\eta}{n}\left\|\nabla\ell(y_i f_{\Theta(k)}(\mathbf{x}_i)) - \nabla\ell(y_i' f_{\Theta^i(k)}(\mathbf{x}_i'))\right\|_2$$
$$\leq G_\alpha^i(k)\left\|\Theta(k) - \Theta^i(k)\right\|_2 + \frac{\eta}{n}\left[\ell(y_i f_{\Theta(k)}(\mathbf{x}_i))\left\|\nabla f_{\Theta(k)}(\mathbf{x}_i)\right\|_2 + \ell(y_i' f_{\Theta^i(k)}(\mathbf{x}_i'))\left\|\nabla f_{\Theta^i(k)}(\mathbf{x}_i')\right\|_2\right], \tag{27}$$

where in the last inequality we have used Lemma D.3 with $S = S^{-i}$. Here, $G_\alpha^i(k) = \max_{\alpha \in [0,1]}\{1 - \eta\lambda_{\min}(\nabla^2\mathcal{L}_{S^{-i}}(\Theta_\alpha)), \eta\lambda_{\max}(\nabla^2\mathcal{L}_{S^{-i}}(\Theta_\alpha(k)))\}$ with $\Theta_\alpha(k) = \alpha\Theta(k) + (1-\alpha)\Theta^i(k)$. From Proposition C.2 with $S = S^{-i}$, we have the following estimates

$$\lambda_{\min}(\nabla^2\mathcal{L}_{S^{-i}}(\Theta_\alpha(k))) \geq -\frac{C_{\sigma,b}\, p^{\frac{3}{2}}\left(\sqrt{\log(\frac{m}{\delta})} + \sqrt{p} + \max_{i \in m}\|\mathbf{c}_i(0) - [\mathbf{c}_\alpha(k)]_i\|_2\right)}{\sqrt{m}}\mathcal{L}_{S^{-i}}(\Theta_\alpha(k)),$$

$$\lambda_{\max}(\nabla^2\mathcal{L}_{S^{-i}}(\Theta_\alpha(k))) \leq C_{\sigma,b}\, p^2\left(p + \max_{i \in m}\|\mathbf{c}_i(0) - [\mathbf{c}_\alpha(k)]_i\|_2^2\right).$$

To control $G_\alpha^i(k)$, we need to estimate $\lambda_{\min}(\nabla^2\mathcal{L}_{S^{-i}}(\Theta_\alpha(k)))$ and $\lambda_{\max}(\nabla^2\mathcal{L}_{S^{-i}}(\Theta_\alpha(k)))$, respectively. Since $\Theta(k)$ and $\Theta^i(k)$ are the outputs of GD, from Theorem C.4 we know

$$\max_{i \in m}\|\mathbf{c}_i(0) - [\mathbf{c}_\alpha(k)]_i\|_2 \leq \|\Theta(0) - \Theta_\alpha(k)\|_2 \leq \max\{\|\Theta(0) - \Theta(k)\|_2, \|\Theta(0) - \Theta^i(k)\|_2\} \leq 3\|\Theta(0) - \Theta^*\|_2.$$

Recall that we defined $c_{\max} = C_{\sigma,b}\, p^{\frac{3}{2}}\left(\sqrt{\log(\frac{m}{\delta})} + \sqrt{p} + 3\|\Theta(0) - \Theta^*\|_2\right)$. It then follows that

$$\lambda_{\min}(\nabla^2\mathcal{L}_{S^{-i}}(\Theta_\alpha(k))) \geq -\frac{c_{\max}}{\sqrt{m}}\mathcal{L}_{S^{-i}}(\Theta_\alpha(k)), \tag{28}$$

$$\lambda_{\max}(\nabla^2\mathcal{L}_{S^{-i}}(\Theta_\alpha(k))) \leq C_{\sigma,b}\, p^2\left(p + 3\|\Theta(0) - \Theta^*\|_2^2\right). \tag{29}$$

Applying Theorem C.4 again, we have

$$\left\|\Theta(k) - \Theta^i(k)\right\|_2 \leq \left\|\Theta(k) - \Theta(0)\right\|_2 + \left\|\Theta(0) - \Theta^i(k)\right\|_2 \leq 6\left\|\Theta(0) - \Theta^*\right\|_2,$$

Define $D = 6\|\Theta(0) - \Theta^*\|_2$ and $\kappa = \frac{c_{\max}}{\sqrt{m}}$. The condition $m \geq 36c_{\max}^2\|\Theta(0) - \Theta^*\|_2^4$ implies $D^2\kappa \leq 1$. Hence, Lemma B.4 with $\tau = 2$ yields

$$\max_{\alpha \in [0,1]}\mathcal{L}_{S^{-i}}(\Theta_\alpha(k)) \leq 2\max\left\{\mathcal{L}_{S^{-i}}(\Theta(k)), \mathcal{L}_{S^{-i}}(\Theta^i(k))\right\}$$
$$= \frac{2}{n-1}\max\left\{(n-1)\mathcal{L}_{S^{-i}}(\Theta(k)), (n-1)\mathcal{L}_{S^{-i}}(\Theta^i(k))\right\}$$
$$\leq \frac{2}{n-1}\max\left\{n\mathcal{L}_S(\Theta(k)), n\mathcal{L}_{S^i}(\Theta^i(k))\right\}$$
$$\leq 4\max\left\{\mathcal{L}_S(\Theta(k)), \mathcal{L}_{S^i}(\Theta^i(k))\right\}. \tag{30}$$

Plugging (30) back into (28) yields

$$\lambda_{\min}(\nabla^2\mathcal{L}_S(\Theta_\alpha(k))) \geq -\frac{4c_{\max}}{\sqrt{m}}\max\left\{\mathcal{L}_S(\Theta(k)), \mathcal{L}_{S^i}(\Theta^i(k))\right\}. \tag{31}$$

Note that $\eta \leq 1/\rho_\ell \leq \left(C_{\sigma,b}\, p^2(p+3\|\Theta(0)-\Theta^*\|_2^2)\right)^{-1}$ implies $\eta\lambda_{\max}(\nabla^2\mathcal{L}_{S^{-i}}(\Theta_\alpha(k))) \leq 1$. Then, we have

$$G_\alpha^i(k) \leq 1 + \frac{4\eta c_{\max}}{\sqrt{m}} \max\left\{\mathcal{L}_S(\Theta(k)), \mathcal{L}_{S^i}(\Theta^i(k))\right\} =: 1 + M^i(k).$$

Further, from Theorem C.4 and Lemma C.1 and the condition $m \gtrsim \log(1/\delta)$, it holds for any $k \in [T-1]$ and any $\mathbf{x} \in \mathcal{X}$ that

$$\left\|\nabla f_{\Theta(k)}(\mathbf{x})\right\|_2 \leq C_{\sigma,b}p\big(\sqrt{p} + \max_{i\in[m]}\|\mathbf{c}_i(k) - \mathbf{c}_i(0)\|_2\big) \leq C_{\sigma,b}p\big(\sqrt{p} + \|\Theta(k) - \Theta(0)\|_2\big)$$
$$\leq C_{\sigma,b}p\big(\sqrt{p} + 3\|\Theta^* - \Theta(0)\|_2\big). \tag{32}$$

Plugging the estimates of $G_\alpha^i(k)$ and $\|\nabla f_{\Theta(k)}(\mathbf{x})\|_2$ back into (27), we get

$$\left\|\Theta(k+1) - \Theta^i(k+1)\right\|_2$$
$$\leq \big(1 + M^i(k)\big)\left\|\Theta(k) - \Theta^i(k)\right\|_2 + C_{\sigma,b}p\big(\sqrt{p} + 3\|\Theta^* - \Theta(0)\|_2\big)\frac{\eta}{n}\left[\ell\big(y_i f_{\Theta(k)}(\mathbf{x}_i)\big) + \ell\big(y_i' f_{\Theta^i(k)}(\mathbf{x}_i')\big)\right]$$
$$\leq C_{\sigma,b}p\big(\sqrt{p} + 3\|\Theta^* - \Theta(0)\|_2\big)\frac{\eta}{n}\sum_{t=0}^{k}\bigg(\prod_{s=t+1}^{k}\big(1 + M^i(s)\big)\bigg)\left[\ell\big(y_i f_{\Theta(t)}(\mathbf{x}_i)\big) + \ell\big(y_i' f_{\Theta^i(t)}(\mathbf{x}_i')\big)\right]$$
$$\leq C_{\sigma,b}p\big(\sqrt{p} + 3\|\Theta^* - \Theta(0)\|_2\big)\frac{\eta}{n}\sum_{t=0}^{k}\bigg(\exp\big(\sum_{s=t+1}^{k} M^i(s)\big)\bigg)\left[\ell\big(y_i f_{\Theta(t)}(\mathbf{x}_i)\big) + \ell\big(y_i' f_{\Theta^i(t)}(\mathbf{x}_i')\big)\right]$$
$$\leq C_{\sigma,b}p\big(\sqrt{p} + 3\|\Theta^* - \Theta(0)\|_2\big)\frac{\eta}{n}\exp\big(\max_{i\in[n]}\sum_{s=1}^{k} M^i(s)\big)\sum_{t=0}^{k}\left[\ell\big(y_i f_{\Theta(t)}(\mathbf{x}_i)\big) + \ell\big(y_i' f_{\Theta^i(t)}(\mathbf{x}_i')\big)\right], \tag{33}$$

where the third inequality uses the inequality $(1 + x) \leq e^x$ for all $x \geq 0$.

Since $\sqrt{m} \geq 2\eta c_{\max}\max_{i\in[n]}\left\{\sum_{s=1}^{T}\mathcal{L}_S(\Theta(s)), \sum_{s=1}^{T}\mathcal{L}_{S^i}(\Theta^i(s))\right\}$, then According to Theorem C.4 with training dataset $S$ and $S^i$, respectively, it holds that

$$\max_{i\in[n]}\sum_{s=1}^{k} M^i(s) \leq \frac{2\eta c_{\max}}{\sqrt{m}} \max_{i\in[n]}\left\{\sum_{s=1}^{k}\mathcal{L}_S(\Theta(s)), \sum_{s=1}^{k}\mathcal{L}_{S^i}(\Theta^i(s))\right\}$$
$$\leq \frac{2\eta c_{\max}}{\sqrt{m}} \max_{i\in[n]}\left\{2T\mathcal{L}_S(\Theta^*) + 2T\mathcal{L}_{S^i}(\Theta^*) + \frac{1}{\eta}\|\Theta(0) - \Theta^*\|_2^2\right\}$$
$$\leq \frac{2\eta c_{\max}}{\sqrt{m}} \max_{i\in[n]}\left\{\frac{1}{4\eta}\|\Theta(0) - \Theta^*\|_2^2 + \frac{1}{\eta}\|\Theta(0) - \Theta^*\|_2^2\right\}$$
$$\leq 1,$$

where in the last second inequality used $\|\Theta(0)-\Theta^*\|_2^2 \geq 8\max\{\eta T\big(\mathcal{L}_S(\Theta^*)+\mathcal{L}_{\widetilde{S}}(\Theta^*)\big)\} \geq 8\eta T\max\{\mathcal{L}_S(\Theta^*), \mathcal{L}_{\widetilde{S}}(\Theta^*)\}$ due to $\mathcal{L}_S(\Theta^*) + \mathcal{L}_{\widetilde{S}}(\Theta^*) \geq \mathcal{L}_{S^i}(\Theta^*)$, and in the last inequality we have used $m \gtrsim c_{\max}^2 \asymp C_{\sigma,b}p^3(\log(\frac{m}{\delta}) + p + \|\Theta(0) - \Theta^*\|_2^2)$.

Putting the above observation into (33) yields

$$\left\|\Theta(k+1) - \Theta^i(k+1)\right\|_2 \leq C_{\sigma,b}p\big(\sqrt{p} + 3\|\Theta^* - \Theta(0)\|_2\big)\frac{\eta}{n}\sum_{t=0}^{k}\left[\ell\big(y_i f_{\Theta(k)}(\mathbf{x}_i)\big) + \ell\big(y_i' f_{\Theta^i(k)}(\mathbf{x}_i')\big)\right].$$

Summing over $i \in [n]$ and taking an average, it holds

$$\mathbb{E}_{S,\widetilde{S}}\left[\frac{1}{n}\sum_{i=1}^{n}\left\|\Theta(k+1) - \Theta^i(k+1)\right\|_2\right] \leq C_{\sigma,b}p\big(\sqrt{p} + 3\|\Theta^* - \Theta(0)\|_2\big)\frac{\eta}{n}\sum_{t=0}^{k}\mathbb{E}_S\left[\mathcal{L}_S\big(\Theta(t)\big)\right] \tag{34}$$

where we have used $\mathbb{E}_{S,\widetilde{S}}\left[\mathcal{L}_{S^i}\big(\Theta^i(t)\big)\right] = \mathbb{E}_S\left[\mathcal{L}_S\big(\Theta(t)\big)\right]$. The proof of the first inequality of the lemma is complete.

From (32) we know for all $k \in [T-1]$ and $(\mathbf{x}, y) \in \mathcal{X} \times \mathcal{Y}$, the Lipschitz constant of $\ell$ with respect to $\Theta(k)$ can be controlled as $\|\nabla\ell(yf_{\Theta(k)}(\mathbf{x}))\|_2 \leq \|\nabla f_{\Theta(k)}(\mathbf{x})\|_2 \leq C_{\sigma,b}p(\sqrt{p} + 3\|\Theta^* - \Theta(0)\|_2)$. Then, from Lemma D.2 and the on-average stability inequality (34), it holds that

$$\mathbb{E}_S\big[\mathcal{L}(\Theta(T)) - \mathcal{L}_S(\Theta(T))\big] \leq C_{\sigma,b}p^2\big(p + p\|\Theta^* - \Theta(0)\|_2^2\big)\frac{\eta}{n}\mathbb{E}_S\Big[\sum_{t=0}^{T}\mathcal{L}_S\big(\Theta(t)\big)\Big].$$

By further noting that (6) holds with probability at least $1 - \delta$ over initialization $\mathbf{c}(0)$, this completes the proof of the theorem. $\qquad\square$

Combining Theorems 4.3 and 4.10, the proof of Theorem 4.11 follows immediately.

*Proof of Theorem 4.11.* Eq. (25) together with the condition for $m$ and the condition $\Lambda_{\Theta^*}^2 = \|\Theta(0) - \Theta^*\|_2^2 \geq 8\max\{\eta T(\mathcal{L}_S(\Theta^*) + \mathcal{L}_{\widetilde{S}}(\Theta^*)), \eta(\mathcal{L}_S(\Theta(0)) + \mathcal{L}_{\widetilde{S}}(\Theta(0)))\}$ show that

$$\frac{1}{T}\sum_{k=0}^{T}\mathcal{L}_S(\Theta(k)) \leq \frac{1}{T}\mathcal{L}_S(\Theta(0)) + 2\mathcal{L}_S(\Theta^*) + \frac{1}{\eta T}\|\Theta(0) - \Theta^*\|_2^2$$

$$\leq \frac{2\eta T\mathcal{L}_S(\Theta^*) + \frac{9}{8}\|\Theta(0) - \Theta^*\|_2^2}{\eta T} \leq \frac{2\mathfrak{C}_S(\Theta^*)}{\eta T},$$

where $\mathfrak{C}_S(\Theta^*) = 2\eta T\mathcal{L}_S(\Theta^*) + \|\Theta(0) - \Theta^*\|_2^2$. Combining this with Theorems C.4 and D.4 yields

$$\mathbb{E}_S\big[\mathcal{L}(\Theta(T))\big] \lesssim \frac{\eta\Lambda_{\Theta^*}^2}{n}\mathbb{E}_S\Big[\sum_{t=0}^{T}\mathcal{L}_S\big(\Theta(t)\big)\Big] + \mathbb{E}_S\big[\mathcal{L}_S(\Theta(T))\big] \lesssim \big(\frac{\Lambda_{\Theta^*}^2}{n} + \frac{1}{\eta T}\big)\mathfrak{C}(\Theta^*),$$

where $\mathfrak{C}(\Theta^*) = \mathbb{E}_S\big[\mathfrak{C}_S(\Theta^*)\big]$. By further noting that (6) holds with probability at least $1 - \delta$ over initialization $\mathbf{c}(0)$, this completes the proof of the theorem. $\qquad\square$

**Theorem D.5** (Generalization under Realizability). *Suppose Assumption 4.5 holds. Let $\eta$ be a constant step size satisfying $\eta \lesssim \min\big\{g^2(1), g^2(1)(\mathcal{L}_S(\Theta(0)))^{-1}, (p^2(p + g^2(1/T)))^{-1}\big\}$. Assume $\eta T \gtrsim n$ and $m \gtrsim \big(\log(m/\delta) + g^2(1/T)\big)g^4(1/T)$. Then, with probability at least $1 - \delta$ over the random initialization, it holds that*

$$\mathbb{E}_S\big[\mathcal{L}(\Theta(T))\big] \lesssim \frac{\big(\eta + g^2(\frac{1}{T})\big)g^2(\frac{1}{T})}{n}.$$

*Proof.* The proof follows directly from Theorem 4.11. It suffices to verify that the width and step size conditions on $m$ and $\eta$ required by Theorem 4.11 are satisfied, which can be shown by arguments similar to those used in the proof of Theorem 4.7. $\qquad\square$

*Proof of Theorem 4.12.* Plugging the estimates of $\mathcal{L}_S(\Theta(0))$ and $g(1/T)$ (see the proof of Theorem 4.9) into Theorem D.5 completes the proof of the theorem. $\qquad\square$

# E. Proofs for DP-GD

## E.1. DP-GD Algorithm

The detailed differentially private GD algorithm given in Section 4.4 is described in Algorithm 1.

## E.2. Privacy Guarantee

In this subsection, we provide proofs for privacy guarantee (i.e., Theorem 4.14) of our DP-GD algorithm. The proofs are based on Rényi differential privacy (RDP), a relaxation of classical differential privacy that allows for a more refined analysis of privacy loss. We call two training

---

**Algorithm 1** Differentially Private GD for KANs

---

1: **Inputs:** Dataset $S = \{z_i \in \mathcal{Z}\}_{i=1}^n$, step size $\eta > 0$, iteration number $T > 0$,
   privacy parameters $\epsilon, \delta > 0$.
2: Generate $\mathbf{a}(0)$ and $\mathbf{c}(0)$ via (3)
3: Set $\widetilde{\Theta}(0) = (\mathbf{a}(0), \mathbf{c}(0))$.
4: **for** $k = 0$ to $T - 1$ **do**
5:     Update $\mathbf{a}(k + 1)$ via (4)
6:     Update $\mathbf{c}(k + 1)$ via (5)
7:     Set $\widetilde{\Theta}(k + 1) = (\mathbf{a}(k + 1), \mathbf{c}(k + 1))$
8: **end for**
9: **return:** $\Theta_{\text{priv}} := \widetilde{\Theta}(T) = (\mathbf{a}(T), \mathbf{c}(T))$

---

**Definition E.1** (RDP (Mironov, 2017)). For $\lambda > 1$, $\rho > 0$, a randomized algorithm $\mathcal{A}$ satisfies $(\lambda, \rho)$-RDP, if, for all neighboring datasets $S$ and $S'$, we have

$$D_\lambda\big(\mathcal{A}(S) \parallel \mathcal{A}(S')\big) := \frac{1}{\lambda - 1} \log \int \Big( \frac{P_{\mathcal{A}(S)}(\theta)}{P_{\mathcal{A}(S')}(\theta)} \Big)^\lambda P_{\mathcal{A}(S')}(\theta)\, d\theta \leq \rho,$$

where $P_{\mathcal{A}(S)}(\theta)$ and $P_{\mathcal{A}(S')}(\theta)$ are the density of $\mathcal{A}(S)$ and $\mathcal{A}(S')$, respectively.

To achieve DP, we need the concept of $\ell_2$-sensitivity defined as follows.

**Definition E.2** ($\ell_2$-sensitivity). The $\ell_2$-sensitivity of a function (mechanism) $\mathcal{M} : \mathcal{Z}^n \to \mathcal{W}$ is defined as $\Delta = \sup_{S,S'} \|\mathcal{M}(S) - \mathcal{M}(S')\|_2$, where $S$ and $S'$ are neighboring datasets.

A basic mechanism to obtain RDP is Gaussian mechanism.

**Lemma E.3** (Gaussian mechanism (Mironov, 2017)). *Consider a function* $\mathcal{M} : \mathcal{Z}^n \to \mathcal{R}^d$ *with the* $\ell_2$*-sensitivity parameter* $\Delta$*, and a dataset* $S \subset \mathcal{Z}^n$*. The Gaussian mechanism* $\mathcal{G}(S, \sigma) = \mathcal{M}(S) + \mathbf{b}$*, where* $\mathbf{b} \sim \mathcal{N}(0, \sigma^2 \mathbf{I}_d)$*, satisfies* $(\lambda, \frac{\lambda \Delta^2}{2\sigma^2})$*-RDP.*

A connection $(\epsilon, \delta)$-DP and RDP is established in the following lemma.

**Lemma E.4** (From RDP to $(\epsilon, \delta)$-DP (Mironov, 2017)). *If a randomized algorithm* $\mathcal{A}$ *satisfies* $(\lambda, \rho)$*-RDP with* $\lambda > 1$*, then* $\mathcal{A}$ *satisfies* $(\rho + \log(1/\delta)/(\lambda - 1), \delta)$*-DP for all* $\delta \in (0, 1)$*.*

The following post-processing property enables flexible use of private data outputs while preserving rigorous privacy guarantees.

**Lemma E.5** (Post-processing (Mironov, 2017)). *Let* $\mathcal{A} : \mathcal{Z}^n \to \mathcal{W}_1$ *satisfy* $(\lambda, \rho)$*-RDP and* $f : \mathcal{W}_1 \to \mathcal{W}_2$ *be an arbitrary function. Then* $f \circ \mathcal{A} : \mathcal{Z}^n \to \mathcal{W}_2$ *satisfies* $(\lambda, \rho)$*-RDP.*

The following RDP composition theorem characterizes the privacy of a composition of parallel or adaptive mechanisms in terms of the privacy guarantees of the individual mechanisms.

**Lemma E.6** (Composition of RDP (Mironov, 2017)). *Fix an order* $\lambda > 1$*. For each* $i \in [k]$*, let* $\mathcal{A}_i : \mathcal{Z}^n \to \mathcal{W}_i$ *be a randomized mechanism satisfying* $(\lambda, \rho_i)$*-RDP. Then the following statements hold.*

(a) *Joint (simultaneous) release. Let* $\mathcal{A}(S) = (\mathcal{A}_1(S), \dots, \mathcal{A}_k(S))$*. Suppose* $\{\mathcal{A}_i\}_{i=1}^k$ *are independent. Then* $\mathcal{A}$ *satisfies* $(\lambda, \sum_{i=1}^k \rho_i)$*-RDP.*

(b) *Adaptive composition. Suppose* $\mathcal{A}_1, \dots, \mathcal{A}_k$ *are applied sequentially, and for each* $i \in [k]$*,* $\mathcal{A}_i$ *may depend on the previous outputs* $\mathcal{A}_1(S), \dots, \mathcal{A}_{i-1}(S)$*. If for every fixed realization* $w_{<i} := (\mathcal{A}_1(S), \dots, \mathcal{A}_{i-1}(S))$ *of previous outputs, the conditional mechanism* $\mathcal{A}_i(\cdot\,; w_{<i})$ *satisfies* $(\lambda, \rho_i)$*-RDP, then the overall mechanism*

$$\mathcal{A}(S) = \big(\mathcal{A}_1(S), \mathcal{A}_2(S; \mathcal{A}_1(S)), \dots, \mathcal{A}_k(S; \mathcal{A}_1(S), \dots, \mathcal{A}_{k-1}(S))\big)$$

*satisfies* $(\lambda, \sum_{i=1}^k \rho_i)$*-RDP.*

Based on the above lemmas, we now prove Theorem 4.14.

*Proof of Theorem 4.14.* Without loss of generality, we assume that the neighboring datasets $S$ and $S'$ differ in the first data point, i.e., $z_1$ and $z_1'$. We begin by estimating the $\ell_2$-sensitivity of $\partial_{\mathbf{a}}\mathcal{L}_S(\widetilde{\Theta}(k))$ and $\partial_{\mathbf{c}}\mathcal{L}_S(\widetilde{\Theta}(k))$ at each iteration $k$. Note Lemma B.2 implies $\|\mathbf{c}(0)\|_2 \le 4\sqrt{pm} + 2\sqrt{\log(2/\delta)}$ with probability at least $1 - \delta/2$. In the following proof, we assume the event $\{\mathbf{c}(0) : \|\mathbf{c}(0)\|_2 \le 4\sqrt{pm} + 2\sqrt{\log(2/\delta)}\}$ holds and aim to show Algorithm 1 satisfies $(\epsilon, \delta/2)$-DP.

Noting that $S$ and $S'$ differ in the first data point. From the update rule of $\mathbf{a}(k)$ we know

$$
\begin{aligned}
&\left\|\partial_{\mathbf{a}}\mathcal{L}_S(\widetilde{\Theta}(k)) - \partial_{\mathbf{a}}\mathcal{L}_{S'}(\widetilde{\Theta}(k))\right\|_2 \\
&= \frac{1}{n}\left\|\partial_{\mathbf{a}}\ell(\widetilde{\Theta}(k), z_1) - \partial_{\mathbf{a}}\ell(\widetilde{\Theta}(k), z_1')\right\|_2 \le \frac{1}{n}\left(\left\|\partial_{\mathbf{a}}\ell(\widetilde{\Theta}(k), z_1)\right\|_2 + \left\|\partial_{\mathbf{a}}\ell(\widetilde{\Theta}(k), z_1')\right\|_2\right) \\
&\le \frac{2B_\ell' B_\sigma' B_b' B_b p}{n\sqrt{m}}\left(\|\mathbf{c}(0)\|_2 + \|\mathbf{c}(0) - \mathbf{c}\|_2\right) \le \frac{2B_\ell' B_\sigma' B_b' B_b p}{n}\left(4\sqrt{p} + \frac{\log(2/\delta) + R_2}{\sqrt{m}}\right),
\end{aligned}
$$

where in the last second inequality we have used (9) together with Assumption 4.2, and the last inequality follows from $\mathbf{c}(k) \in \Omega_{\mathbf{c}} = \mathcal{B}(\mathbf{c}(0), R_2)$ and the condition for $m$.

Hence, the $\ell_2$-sensitivity of the gradient $\partial_{\mathbf{a}}\mathcal{L}_S(\widetilde{\Theta}(k))$ is $\Delta_{\mathbf{a}} = \frac{2B_\ell' B_\sigma' B_b' B_b}{n}p\left(4\sqrt{p} + \frac{\log(2/\delta) + R_2}{\sqrt{m}}\right)$. Let $C_1 = 8(B_\ell' B_\sigma' B_b' B_b)^2 p^2\left(4\sqrt{p} + \frac{\log(2/\delta) + R_2}{\sqrt{m}}\right)^2$. We set

$$
\sigma_1^2 = \frac{2T(1 + \frac{\log(2T/\delta)}{\epsilon})\Delta_{\mathbf{a}}^2}{\epsilon} = \frac{C_1 T(1 + \frac{\log(2T/\delta)}{\epsilon})}{n^2 \epsilon}.
$$

Lemma E.3 implies that the mechanism $\partial_{\mathbf{a}}\mathcal{L}_S(\widetilde{\Theta}(k)) + \mathbf{b}_1(k)$ satisfies $(\lambda, \frac{\epsilon}{4T})$-RDP with $\lambda = 1 + \frac{2\log(2/\delta)}{\epsilon}$. Applying Lemma E.5 (post-processing property) we know $\mathbf{a}(k+1) = \mathbf{a}(k) - \eta\left(\partial_{\mathbf{a}}\mathcal{L}_S(\widetilde{\Theta}(k)) + \mathbf{b}_1(k)\right)$ also satisfies $(\lambda, \frac{\epsilon}{4T})$-RDP.

Similarly, combining Assumption 4.2 with (7), we know

$$
\left\|\partial_{\mathbf{c}}\mathcal{L}_S(\widetilde{\Theta}(k)) - \partial_{\mathbf{c}}\mathcal{L}_{S'}(\widetilde{\Theta}(k))\right\|_2 \le \frac{1}{n}\left(\left\|\partial_{\mathbf{c}}\ell(\widetilde{\Theta}(k), z_1)\right\|_2 + \left\|\partial_{\mathbf{c}}\ell(\widetilde{\Theta}(k), z_1')\right\|_2\right) \le \frac{2B_\ell' B_b \sqrt{p}}{n}.
$$

Then, the $\ell_2$-sensitivity of $\partial_{\mathbf{c}}\mathcal{L}_S(\widetilde{\Theta}(k))$ is $\Delta_{\mathbf{c}} = \frac{2B_\ell' B_b \sqrt{p}}{n}$. Let $C_2 = 8(B_\ell' B_b)^2 p$. Set

$$
\sigma_2^2 = \frac{2T(1 + \frac{\log(2T/\delta)}{\epsilon})\Delta_{\mathbf{c}}^2}{\epsilon} = \frac{C_2 T(1 + \frac{\log(2T/\delta)}{\epsilon})}{n^2 \epsilon}.
$$

Hence, by applying Lemmas E.3 and E.5 again, we know both the mechanism $\partial_{\mathbf{c}}\mathcal{L}_S(\Theta(k)) + \mathbf{b}_2(k)$ and $\mathbf{c}(k+1)$ satisfy $(\lambda, \frac{\epsilon}{4T})$-RDP.

According to part (a) in Lemma E.6, for each $k = 0, \ldots, T-1$ and condition on the fixed $(\widetilde{\Theta}(1), \ldots, \widetilde{\Theta}(k))$, we know $\widetilde{\Theta}(k+1) = [\mathbf{a}(k+1), \mathbf{c}(k+1)]$ satisfies $(\lambda, \frac{\epsilon}{2T})$-RDP. Combining all iterations, from part (b) in Lemma E.6, we know $\widetilde{\Theta}(T)$ is $(\lambda, \frac{\epsilon}{2})$-RDP where $\lambda = 1 + \frac{2\log(2/\delta)}{\epsilon}$. Finally, Lemma E.4 implies $\widetilde{\Theta}(T)$ satisfies $(\epsilon, \frac{\delta}{2})$-DP. Combining this observation with the event $\{\mathbf{c}(0) : \|\mathbf{c}(0)\|_2 \le 4\sqrt{pm} + 2\sqrt{\log(2/\delta)}\}$ holds with probability at least $1 - \delta/2$ completes the proof. □

### E.3. Utility Guarantee

Now, we present the proofs for Theorems 4.15. Throughout this subsection, we condition on a fixed initialization satisfying (6). Thus all estimates below are deterministic with respect to the initialization. The expectation $\mathbb{E}_{\mathcal{A}}$ is taken only over the Gaussian perturbations generated after initialization, and $\mathbb{E}_{S,\mathcal{A}}$ is taken over the sample and these Gaussian perturbations, conditional on the initialization. At the end, since (6) holds with probability at least $1 - \delta$ over the initialization, the resulting utility bounds hold with the same probability over the initialization.

We require the following two properties of the training loss to estimate the optimization error of DP-GD.

**Lemma E.7.** *Let $\delta \in (0,1)$. Suppose* (6)*, $m \gtrsim \log(m/\delta)$ and Assumptions 4.1 and 4.2 hold. Let $\bar{\rho} = C_{\sigma,b}\, p^2 (p + R_2^2)$ and $\bar{\kappa} = C_{\sigma,b}\, p\big(1 + \frac{\sqrt{p}(\sqrt{p}+R_2)}{\sqrt{m}}\big)$. For any $S$ and any $\widetilde{\Theta} = (\mathbf{a}, \mathbf{c})$ with $\mathbf{c} \in \Omega_{\mathbf{c}}$, it holds that*

$$\lambda_{\max}\big(\nabla^2 \mathcal{L}_S(\widetilde{\Theta})\big) \leq \bar{\rho}, \qquad \lambda_{\min}\big(\nabla^2 \mathcal{L}_S(\widetilde{\Theta})\big) \geq -\frac{\bar{\kappa}}{\sqrt{m}}\, \mathcal{L}_S(\widetilde{\Theta}),$$

*and*

$$\big\|\nabla \mathcal{L}_S(\widetilde{\Theta})\big\|_2^2 \leq 2\bar{\rho}\mathcal{L}_S(\widetilde{\Theta}).$$

*Proof.* Since $\mathbf{c} \in \Omega_{\mathbf{c}}$, we know $\max_{i\in[m]} \|\mathbf{c}_i - \mathbf{c}_i(0)\|_2 \leq R_2$. Applying Proposition C.2 with $\Theta = \widetilde{\Theta}$, the fact $\max_{i\in[m]} \|\mathbf{c}_i - \mathbf{c}_i(0)\|_2 \leq R_2$ and the condition $m \gtrsim \log(m/\delta)$, we obtain the estimates $\lambda_{\max}(\nabla^2 \mathcal{L}_S(\widetilde{\Theta})) \leq \bar{\rho}$ and $\lambda_{\min}(\nabla^2 \mathcal{L}_S(\widetilde{\Theta})) \geq -\bar{\kappa}\ell(y f_{\widetilde{\Theta}}(\mathbf{x}))/\sqrt{m}$ immediately.

Note that in the proof of Proposition C.2, $\bar{\rho}$ is an upper bound for the term $\sup_{\mathbf{x}\in\mathcal{X}} \|\nabla f_{\Theta}(\mathbf{x})\|_2^2 + \|\nabla^2 f_{\Theta}(\mathbf{x})\|_2$. Then, it holds that

$$\big\|\nabla \mathcal{L}_S(\widetilde{\Theta})\big\|_2^2 = \Big\|\frac{1}{n}\sum_{i=1}^n \ell'(y_i f_{\widetilde{\Theta}}(\mathbf{x}_i))y_i \nabla f_{\widetilde{\Theta}}(\mathbf{x}_i)\Big\|_2^2 \leq \Big|\frac{1}{n}\sum_{i=1}^n \ell'(y_i f_{\widetilde{\Theta}}(\mathbf{x}_i))\Big|^2 \sup_{i\in[n]} \big\|\nabla f_{\Theta}(\mathbf{x}_i)\big\|_2^2$$

$$\leq \bar{\rho} B'_\ell \Big|\frac{1}{n}\sum_{i=1}^n \ell(y_i f_{\widetilde{\Theta}}(\mathbf{x}_i))\Big| = \bar{\rho} B'_\ell \mathcal{L}_S(\widetilde{\Theta}) \leq 2\bar{\rho}\mathcal{L}_S(\widetilde{\Theta}),$$

where the second inequality used the self-boundedness property and $|\ell'(\cdot)| \leq B'_\ell$ (see Assumption 4.2). This completes the proof of the lemma. $\qquad\square$

**Lemma E.8.** *Let $\delta \in (0,1)$. Suppose* (6)*, Assumptions 4.1 and 4.2 hold. Let $\{\widetilde{\Theta}(k)\}$ be produced by Algorithm 1 based on $S$ with $T$ iterations. Let $\mathbf{B}(k) = (\mathbf{b}_1(k), \mathbf{b}_2(k))$, where $\mathbf{b}_1(k), \mathbf{b}_2(k)$ are independent Gaussian noise vectors added at iteration $k$. For $k = 0, \ldots, T-1$, it holds that*

$$\mathcal{L}_S(\widetilde{\Theta}(k+1)) \leq \mathcal{L}_S(\widetilde{\Theta}(k)) - \Big(\frac{1}{2\eta} - \frac{\bar{\rho}}{2}\Big)\|\widetilde{\Theta}(k) - \widetilde{\Theta}(k+1)\|_2^2 + \frac{\eta}{2}\|\mathbf{B}(k)\|_2^2.$$

*Furthermore, if we assume $\eta \leq 1/\bar{\rho}$, then*

$$\mathcal{L}_S(\widetilde{\Theta}(k+1)) \leq \mathcal{L}_S(\widetilde{\Theta}(k)) + \frac{\eta}{2}\|\mathbf{B}(k)\|_2^2.$$

*Proof.* Denote $\Omega = \Omega_{\mathbf{a}} \times \Omega_{\mathbf{c}} = \mathcal{B}(\mathbf{a}(0), R_1) \times \mathcal{B}(\mathbf{c}(0), R_2)$. From the update rule of DP-GD, for any $k = 0, \ldots, T-1$, we know

$$\widetilde{\Theta}(k+1) = \mathrm{Proj}_\Omega\big(\widetilde{\Theta}(k) - \eta\big(\nabla \mathcal{L}_S(\widetilde{\Theta}(k)) + \mathbf{B}(k)\big)\big).$$

Let $W(k) = \widetilde{\Theta}(k) - \eta\big(\nabla \mathcal{L}_S(\widetilde{\Theta}(k)) + \mathbf{B}(k)\big)$. Note that $\Omega$ is a closed convex set. By the projection theorem, we know

$$\big\langle W(k) - \widetilde{\Theta}(k+1), \widetilde{\Theta}(k) - \widetilde{\Theta}(k+1)\big\rangle \leq 0.$$

Substituting the definition of $W(k)$ and dividing $\eta$, we have

$$\frac{1}{\eta}\big\|\widetilde{\Theta}(k) - \widetilde{\Theta}(k+1)\big\|_2^2 \leq \big\langle \nabla \mathcal{L}_S(\widetilde{\Theta}(k)) + \mathbf{B}(k), \widetilde{\Theta}(k) - \widetilde{\Theta}(k+1)\big\rangle. \tag{35}$$

On the other hand, according to Lemma E.7, we know that $\lambda_{\max}\big(\nabla^2 \mathcal{L}_S(\widetilde{\Theta}(k))\big) \leq \bar{\rho}$ for all $k \in [T]$ with probability at least $1 - \delta$. It then follows

$$\mathcal{L}_S(\widetilde{\Theta}(k+1)) \leq \mathcal{L}_S(\widetilde{\Theta}(k)) + \big\langle \nabla \mathcal{L}_S(\widetilde{\Theta}(k)), \widetilde{\Theta}(k+1) - \widetilde{\Theta}(k)\big\rangle + \frac{\bar{\rho}}{2}\big\|\widetilde{\Theta}(k+1) - \widetilde{\Theta}(k)\big\|_2^2$$

$$\leq \mathcal{L}_S(\widetilde{\Theta}(k)) - \big(\frac{1}{\eta} - \frac{\bar{\rho}}{2}\big)\big\|\widetilde{\Theta}(k) - \widetilde{\Theta}(k+1)\big\|_2^2 + \big\langle \mathbf{B}(k), \widetilde{\Theta}(k) - \widetilde{\Theta}(k+1)\big\rangle$$

$$\leq \mathcal{L}_S(\widetilde{\Theta}(k)) - \big(\frac{1}{2\eta} - \frac{\bar{\rho}}{2}\big)\big\|\widetilde{\Theta}(k) - \widetilde{\Theta}(k+1)\big\|_2^2 + \frac{\eta}{2}\big\|\mathbf{B}(k)\big\|_2^2,$$

where in the second inequality we have used (35) and the last inequality is due to $ab \leq \frac{\eta a^2}{2} + \frac{b^2}{2\eta}$. The proof is complete. $\quad\square$

Our proofs also need the following mirror descent inequality.

**Lemma E.9.** *Let $\Omega \subseteq \mathbb{R}^{(m+1)dp}$ be a nonempty closed convex set, and let $Proj_\Omega(\cdot)$ denote the Euclidean projection onto $\Omega$. Fix any $\eta > 0$ and any vector $g \in \mathbb{R}^{(m+1)dp}$. Define*

$$\Theta^+ = Proj_\Omega(\Theta - \eta g).$$

*Then, for any comparator $\Theta^* \in \Omega$, the following inequality holds*

$$\langle g, \, \Theta - \Theta^* \rangle \leq \frac{1}{2\eta}\Big(\|\Theta - \Theta^*\|_2^2 - \|\Theta^+ - \Theta^*\|_2^2\Big) + \frac{\eta}{2}\|g\|_2^2.$$

*Proof.* Let $z := \Theta - \eta g$ such that $\Theta^+ = Proj_\Omega(z)$. From the projection theorem we know for all $u \in \Omega$, $\langle z - \Theta^+, \, u - \Theta^+ \rangle \leq 0$. Choosing $u = \Theta^* \in \Omega$ and substituting $z = \Theta - \eta g$ yield

$$\eta\langle g, \, \Theta^* - \Theta^+ \rangle \geq \langle \Theta - \Theta^+, \, \Theta^* - \Theta^+ \rangle. \tag{36}$$

Note that for any vectors $\mathbf{a}, \mathbf{b}, \mathbf{c} \in \mathbb{R}^{(m+1)dp}$, $2\langle \mathbf{a} - \mathbf{b}, \, \mathbf{c} - \mathbf{b} \rangle = \|\mathbf{b} - \mathbf{c}\|_2^2 + \|\mathbf{a} - \mathbf{b}\|_2^2 - \|\mathbf{a} - \mathbf{c}\|_2^2$. Applying this with $(\mathbf{a}, \mathbf{b}, \mathbf{c}) = (\Theta, \Theta^+, \Theta^*)$ gives

$$\langle \Theta - \Theta^+, \, \Theta^* - \Theta^+ \rangle = \frac{1}{2}\Big(\|\Theta^+ - \Theta^*\|_2^2 + \|\Theta - \Theta^+\|_2^2 - \|\Theta - \Theta^*\|_2^2\Big).$$

Combining (36) and the above equality and dividing by $\eta$ yield

$$\langle g, \, \Theta^* - \Theta^+ \rangle \geq \frac{1}{2\eta}\Big(\|\Theta^+ - \Theta^*\|_2^2 + \|\Theta - \Theta^+\|_2^2 - \|\Theta - \Theta^*\|_2^2\Big).$$

Therefore,

$$\langle g, \, \Theta - \Theta^* \rangle = \langle g, \, \Theta - \Theta^+ \rangle - \langle g, \, \Theta^* - \Theta^+ \rangle \leq \langle g, \, \Theta - \Theta^+ \rangle + \frac{1}{2\eta}\Big(\|\Theta - \Theta^*\|_2^2 - \|\Theta^+ - \Theta^*\|_2^2 - \|\Theta - \Theta^+\|_2^2\Big)$$

$$\leq \frac{\eta}{2}\|g\|_2^2 + \frac{1}{2\eta}\|\Theta - \Theta^+\|_2^2 + \frac{1}{2\eta}\Big(\|\Theta - \Theta^*\|_2^2 - \|\Theta^+ - \Theta^*\|_2^2 - \|\Theta - \Theta^+\|_2^2\Big)$$

$$\leq \frac{1}{2\eta}\Big(\|\Theta - \Theta^*\|_2^2 - \|\Theta^+ - \Theta^*\|_2^2\Big) + \frac{\eta}{2}\|g\|_2^2,$$

where we have used Cauchy-Schwarz inequality and the basic inequality $2ab \leq a^2 + b^2$ for all $a, b \in \mathbb{R}$. This completes the proof of the lemma. $\square$

Now we give the optimization risk bounds. For convenience, we restate it below. Let $R = R_1 + R_2$.

**Theorem E.10.** *Suppose (6), Assumptions 4.1 and 4.2 hold. Let $\{\widetilde{\Theta}(k)\}$ be produced by Algorithm 1 with $\eta \leq \min\{1/3\bar{\rho}, 1\}$ and $T$ iterations. For any reference point $\Theta^* = (\mathbf{a}^*, \mathbf{c}^*)$ with $\mathbf{a}^* \in \Omega_\mathbf{a}$ and $\mathbf{c}^* \in \Omega_\mathbf{c}$, assume $m \gtrsim p^2(\log(m/\delta) + R^2)(R^4 + \|\Theta^* - \widetilde{\Theta}(0)\|_2^4)$. Then, it holds that*

$$\frac{1}{T}\sum_{k=1}^{T} \mathbb{E}_\mathcal{A}\big[\mathcal{L}_S(\widetilde{\Theta}(k))\big] \leq 4\mathcal{L}_S(\Theta^*) + \frac{3}{2\eta T}\|\widetilde{\Theta}(0) - \Theta^*\|_2^2 + \frac{mp^4\eta Td\log(2T/\delta)}{n^2\epsilon^2} + 2\mathcal{L}_S(\widetilde{\Theta}(0)).$$

*Here, the expectation is taken only over the Gaussian perturbations generated after the fixed initialization. If we further assume $\widetilde{\Lambda}_{\Theta^*}^2 \geq \eta\mathcal{L}_S(\widetilde{\Theta}(0))$, then*

$$\frac{1}{T}\sum_{k=1}^{T} \mathbb{E}_\mathcal{A}\big[\mathcal{L}_S(\widetilde{\Theta}(k))\big] \leq 4\mathcal{L}_S(\Theta^*) + \frac{4}{\eta T}\|\widetilde{\Theta}(0) - \Theta^*\|_2^2 + \frac{mp^4\eta Td\log(2T/\delta)}{n^2\epsilon^2}.$$

*Proof.* Note that $\|\mathbf{c}(k)\|_2 \in \Omega_{\mathbf{c}}$ for all iterations $k$. Then, Lemma E.7 shows that $\lambda_{\min}\big(\nabla^2\ell(yf_{\widetilde{\Theta}(k)}(\mathbf{x}))\big) \geq -\frac{\bar{\kappa}}{\sqrt{m}}\ell(yf_{\widetilde{\Theta}(k)}(\mathbf{x}))$. Similar to (19), for any fixed $\Theta^*$, according to Taylor expansion and the estimate for $\lambda_{\min}\big(\nabla^2\ell(yf_{\widetilde{\Theta}(k)}(\mathbf{x}))\big)$, we have

$$\mathcal{L}_S(\widetilde{\Theta}(k)) \leq \mathcal{L}_S(\Theta^*) - \big\langle \nabla\mathcal{L}_S(\widetilde{\Theta}(k)), \Theta^* - \widetilde{\Theta}(k)\big\rangle + \frac{\bar{\kappa}}{2\sqrt{m}}\max_{\alpha\in[0,1]}\mathcal{L}_S(\widetilde{\Theta}_{\alpha,k})\big\|\Theta^* - \widetilde{\Theta}(k)\big\|_2^2. \tag{37}$$

where $\widetilde{\Theta}_{\alpha,k} = \alpha\widetilde{\Theta}(k) + (1-\alpha)\Theta^*$. Note that $\rho \leq 1/(2\bar{\rho})$. From Lemma E.8 we know that

$$\mathcal{L}_S(\widetilde{\Theta}(k+1)) \leq \mathcal{L}_S(\widetilde{\Theta}(k)) + \frac{\eta}{2}\big\|\mathbf{B}(k)\big\|_2^2,$$

where $\mathbf{B}(k) = (\mathbf{b}_1(k), \mathbf{b}_2(k))$ with $\mathbf{b}_1(k), \mathbf{b}_2(k)$ being the independent Gaussian noise vectors added at iteration $k$. Plugging (37) into the above inequality yields

$$\mathcal{L}_S(\widetilde{\Theta}(k+1)) \leq \mathcal{L}_S(\Theta^*) - \big\langle \nabla\mathcal{L}_S(\widetilde{\Theta}(k)), \Theta^* - \widetilde{\Theta}(k)\big\rangle + \frac{\eta}{2}\big\|\mathbf{B}(k)\big\|_2^2 + \frac{\bar{\kappa}}{2\sqrt{m}}\max_{\alpha\in[0,1]}\mathcal{L}_S(\widetilde{\Theta}_{\alpha,k})\big\|\Theta^* - \widetilde{\Theta}(k)\big\|_2^2. \tag{38}$$

On the other hand, Lemma E.9 with $g = \nabla\mathcal{L}_S(\widetilde{\Theta}(k)) + \mathbf{B}(k)$, $\Theta = \widetilde{\Theta}(k)$ and $\Theta^+ = \widetilde{\Theta}(k+1)$ implies

$$\big\langle \nabla\mathcal{L}_S(\widetilde{\Theta}(k)) + \mathbf{B}(k), \widetilde{\Theta}(k) - \Theta^*\big\rangle \leq \frac{1}{2\eta}\Big(\big\|\widetilde{\Theta}(k) - \Theta^*\big\|_2^2 - \big\|\widetilde{\Theta}(k+1) - \Theta^*\big\|_2^2\Big) + \frac{\eta}{2}\big\|\nabla\mathcal{L}_S(\widetilde{\Theta}(k)) + \mathbf{B}(k)\big\|_2^2.$$

It then follows that

$$-\big\langle \nabla\mathcal{L}_S(\widetilde{\Theta}(k)), \Theta^* - \widetilde{\Theta}(k)\big\rangle \leq \frac{1}{2\eta}\Big(\big\|\widetilde{\Theta}(k) - \Theta^*\big\|_2^2 - \big\|\widetilde{\Theta}(k+1) - \Theta^*\big\|_2^2\Big) + \frac{\eta}{2}\big\|\nabla\mathcal{L}_S(\widetilde{\Theta}(k)) + \mathbf{B}(k)\big\|_2^2 + \big\langle \mathbf{B}(k), \Theta^* - \widetilde{\Theta}(k)\big\rangle.$$

Putting the above inequality into (38), we know

$$\mathcal{L}_S(\widetilde{\Theta}(k+1)) \leq \mathcal{L}_S(\Theta^*) + \frac{1}{2\eta}\Big(\big\|\widetilde{\Theta}(k) - \Theta^*\big\|_2^2 - \big\|\widetilde{\Theta}(k+1) - \Theta^*\big\|_2^2\Big) + \frac{\eta}{2}\big\|\nabla\mathcal{L}_S(\widetilde{\Theta}(k)) + \mathbf{B}(k)\big\|_2^2$$

$$+ \big\langle \mathbf{B}(k), \Theta^* - \widetilde{\Theta}(k)\big\rangle + \frac{\eta}{2}\big\|\mathbf{B}(k)\big\|_2^2 + \frac{1}{4}\max_{\alpha\in[0,1]}\mathcal{L}_S(\widetilde{\Theta}_{\alpha,k}), \tag{39}$$

where the inequality follows from $\frac{\bar{\kappa}}{2\sqrt{m}}\big\|\Theta^* - \widetilde{\Theta}(k)\big\|_2^2 \leq \frac{1}{4}$, which is ensured by the fact $\|\Theta^* - \widetilde{\Theta}(k)\|_2 \leq \|\Theta^* - \widetilde{\Theta}(0)\|_2 + \|\widetilde{\Theta}(0) - \widetilde{\Theta}(k)\|_2 \leq \|\Theta^* - \Theta(0)\|_2 + R$ and the condition $m \gtrsim p^2(\log(m/\delta) + R^2)(R^4 + \|\Theta^* - \widetilde{\Theta}(0)\|_2^4)$.

Applying Lemma B.4 with $D = \big\|\Theta^* - \widetilde{\Theta}(k)\big\|_2$ and $\kappa = \frac{\bar{\kappa}}{\sqrt{m}}$, and $\tau = (1 - \kappa D/2)^{-1}$, we know

$$\max_{\alpha\in[0,1]}\mathcal{L}_S(\widetilde{\Theta}_{\alpha,k}) \leq \frac{4}{3}\max\big\{\mathcal{L}_S(\widetilde{\Theta}(k)), \mathcal{L}_S(\Theta^*)\big\} \leq \frac{4}{3}\big(\mathcal{L}_S(\widetilde{\Theta}(k)) + \mathcal{L}_S(\Theta^*)\big).$$

Plugging the above inequality back into (39) and taking expectation over the randomness of $\mathbf{B}(k)$, it holds that

$$\mathbb{E}_{\mathbf{B}(k)}\big[\mathcal{L}_S(\widetilde{\Theta}(k+1))\big] \leq \mathcal{L}_S(\Theta^*) + \frac{1}{2\eta}\big(\big\|\widetilde{\Theta}(k) - \Theta^*\big\|_2^2 - \mathbb{E}_{\mathbf{B}(k)}\big[\big\|\widetilde{\Theta}(k+1) - \Theta^*\big\|_2^2\big]\big) + \frac{\eta}{2}\big\|\nabla\mathcal{L}_S(\widetilde{\Theta}(k))\big\|_2^2$$

$$+ \frac{3mp\eta T(1 + \frac{\log(2T/\delta)}{\epsilon})}{2n^2\epsilon}(C_1 d + C_2) + \frac{1}{3}\big(\mathcal{L}_S(\widetilde{\Theta}(k)) + \mathcal{L}_S(\Theta^*)\big)$$

$$\leq \mathcal{L}_S(\Theta^*) + \frac{1}{2\eta}\big(\big\|\widetilde{\Theta}(k) - \Theta^*\big\|_2^2 - \mathbb{E}_{\mathbf{B}(k)}\big[\big\|\widetilde{\Theta}(k+1) - \Theta^*\big\|_2^2\big]\big) + \eta\bar{\rho}\mathcal{L}_S(\widetilde{\Theta}(k))$$

$$+ \frac{3mp\eta T(1 + \frac{\log(2T/\delta)}{\epsilon})}{2n^2\epsilon}(C_1 d + C_2) + \frac{1}{3}\big(\mathcal{L}_S(\widetilde{\Theta}(k)) + \mathcal{L}_S(\Theta^*)\big), \tag{40}$$

where in the first inequality we have used $\mathbb{E}_{\mathbf{B}(k)}\big[\big\langle \mathbf{B}(k), \Theta^* - \widetilde{\Theta}(k)\big\rangle\big] = \mathbb{E}_{\mathbf{B}(k)}\big[\big\langle \mathbf{B}(k), \nabla\mathcal{L}_S(\widetilde{\Theta}(k))\big\rangle\big] = 0$ by noting that $\mathbb{E}[\mathbf{B}(k)] = 0$ and $\Theta^*, \widetilde{\Theta}(k)$ and $S$ are independent of $\mathbf{B}(k)$, and $\mathbb{E}[\|\mathbf{b}\|_2^2] = \sigma^2 d$ if $\mathbf{b} \sim \mathcal{N}(0, \sigma^2\mathbf{I}_d)$, and the second inequality is due to $\|\nabla\mathcal{L}_S(\widetilde{\Theta}(k))\|_2^2 \leq 2\bar{\rho}\mathcal{L}_S(\widetilde{\Theta}(k))$ implied by Lemma E.7.

Taking the expectation over all $\{\mathbf{B}(t)\}_{t=1}^{T-1}$ on both sides of the above inequality and applying it recursively, we have

$$\sum_{k=1}^{T}\mathbb{E}_{\mathcal{A}}\big[\mathcal{L}_S(\widetilde{\Theta}(k))\big] \leq \frac{4T}{3}\mathcal{L}_S(\Theta^*) + \frac{1}{2\eta}\|\widetilde{\Theta}(0) - \Theta^*\|_2^2 + \Big(\eta\bar{\rho} + \frac{1}{3}\Big)\sum_{k=0}^{T-1}\mathbb{E}_{\mathcal{A}}\big[\mathcal{L}_S(\widetilde{\Theta}(k))\big] + \frac{3mp\eta T^2(1 + \frac{\log(2T/\delta)}{\epsilon})}{2n^2\epsilon}\big(C_1 d + C_2\big).$$

Noting that $\eta \leq \frac{1}{3\bar{\rho}}$ implies $\eta\bar{\rho} + \frac{1}{3} \leq \frac{2}{3}$. It then follows that

$$\sum_{k=1}^{T}\mathbb{E}_{\mathcal{A}}\big[\mathcal{L}_S(\widetilde{\Theta}(k))\big] \leq 4T\mathcal{L}_S(\Theta^*) + \frac{3}{2\eta}\|\widetilde{\Theta}(0) - \Theta^*\|_2^2 + \frac{9mp\eta T^2(1 + \frac{\log(2T/\delta)}{\epsilon})}{2n^2\epsilon}\big(C_1 d + C_2\big) + 2\mathcal{L}_S(\widetilde{\Theta}(0)). \quad (41)$$

Therefore,

$$\frac{1}{T}\sum_{k=1}^{T}\mathbb{E}_{\mathcal{A}}\big[\mathcal{L}_S(\widetilde{\Theta}(k))\big] \leq 4\mathcal{L}_S(\Theta^*) + \frac{3}{2\eta T}\|\widetilde{\Theta}(0) - \Theta^*\|_2^2 + \frac{mp^4\eta T d\log(2T/\delta)}{n^2\epsilon^2} + \frac{2}{T}\mathcal{L}_S(\widetilde{\Theta}(0)).$$

By further using $\widetilde{\Lambda}_{\Theta^*}^2 \geq \eta\mathcal{L}_S(\widetilde{\Theta}(0))$, it holds

$$\frac{1}{T}\sum_{k=1}^{T}\mathbb{E}_{\mathcal{A}}\big[\mathcal{L}_S(\widetilde{\Theta}(k))\big] \leq 4\mathcal{L}_S(\Theta^*) + \frac{4}{\eta T}\|\widetilde{\Theta}(0) - \Theta^*\|_2^2 + \frac{mp^4\eta T d\log(2T/\delta)}{n^2\epsilon^2}.$$

The proof is completed. $\qquad\square$

Now, we turn to prove our generalization bounds for DP-GD. Recall that we define $S = \{z_1, \ldots, z_n\}$ and $\widetilde{S} = \{z_1', \ldots, z_n'\}$ be drawn independently from $\mathcal{P}$. For any $i \in [n]$, define $S^i = \{z_1, \ldots, z_{i-1}, z_i', z_{i+1}, \ldots, z_n\}$. We need high probability version of optimization risk bound.

**Lemma E.11** (High-probability optimization bound for DP-GD). *Suppose* (6) *and Assumptions 4.1 and 4.2 hold. Let* $\{\widetilde{\Theta}(k)\}_{k=0}^{T}$ *be produced by Algorithm 1. Assume the conditions of Theorem E.10 hold and* $\|\widetilde{\Theta}(0) - \Theta^*\|_2^2 \geq C\eta\mathcal{L}_S(\widetilde{\Theta}(0))$ *with* $C > 0$ *a constant. Then, conditional on* (6)*, with probability at least* $1 - \delta$ *over the Gaussian perturbations,*

$$\frac{1}{T}\sum_{k=1}^{T}\mathcal{L}_S(\widetilde{\Theta}(k)) \lesssim \mathcal{L}_S(\Theta^*) + \frac{4\|\widetilde{\Theta}(0) - \Theta^*\|_2^2}{\eta T} + \frac{mp^4\eta T d\log(T/\delta)}{n^2\epsilon^2} + \frac{Rp^{3/2}\log(T/\delta)}{n\epsilon}.$$

*Proof.* The proof follows the proof of Theorem E.10, except that we do not take expectation over the Gaussian perturbations.

After summing the one-step inequality, the only random terms are

$$\sum_{k=0}^{T-1}\|\mathbf{B}(k)\|_2^2 \quad \text{and} \quad \sum_{k=0}^{T-1}\langle\mathbf{B}(k), \Theta^* - \widetilde{\Theta}(k)\rangle.$$

The first term is controlled by a standard chi-square concentration bound for Gaussian vectors:

$$\sum_{k=0}^{T-1}\|\mathbf{B}(k)\|_2^2 \lesssim \frac{mp^4 dT^2\log(T/\delta)}{n^2\epsilon^2}$$

with probability at least $1 - \delta/2$.

For the second term, since $\widetilde{\Theta}(k)$ is measurable with respect to the past randomness and $\mathbf{B}(k)$ is an independent centered Gaussian vector, the sequence $\langle\mathbf{B}(k), \Theta^* - \widetilde{\Theta}(k)\rangle$ is a martingale difference sequence. Moreover, on the projected set, $\|\Theta^* - \widetilde{\Theta}(k)\|_2 \leq 2R$. Therefore, by conditional Gaussian concentration, with probability at least $1 - \delta/2$,

$$\sum_{k=0}^{T-1}\langle\mathbf{B}(k), \Theta^* - \widetilde{\Theta}(k)\rangle \lesssim \frac{Rp^{3/2}T\log(T/\delta)}{n\epsilon}.$$

Substituting these two bounds into the summed optimization recursion gives the claim. $\qquad\square$

The following lemma provides on-average argument stability bounds for DP-GD algorithm.

**Lemma E.12** (DP-GD stability). *Let $\delta \in (0, 1)$. Suppose (6), Assumptions 4.1 and 4.2 hold. For any $i \in [n]$, let $\{\widetilde{\Theta}(k)\}_{k=0}^{T}$ and $\{\widetilde{\Theta}^i(k)\}_{k=0}^{T}$ be produced by Algorithm 1 with $\eta \leq \min\{1/3\bar{\rho}, 1\}$ and $T$ iterations based on $S$ and $S^i$, respectively. For any reference point $\Theta^* = (\mathbf{a}^*, \mathbf{c}^*)$ with $\mathbf{a}^* \in \Omega_{\mathbf{a}}$ and $\mathbf{c}^* \in \Omega_{\mathbf{c}}$, assume*

$$m \gtrsim p^2 \big(\log(\tfrac{m}{\delta}) + R^2\big) \cdot \max \Big\{ R^4 + \|\Theta^* - \widetilde{\Theta}(0)\|_2^4, \; \big(\eta T(\mathcal{L}_S(\Theta^*) + \mathcal{L}_{\widetilde{S}}(\Theta^*)) + \eta(\mathcal{L}_S(\widetilde{\Theta}(0)) + \mathcal{L}_{\widetilde{S}}(\widetilde{\Theta}(0)))\big)^2,$$

$$\big(\frac{Rp^{3/2}\eta T \log(T/\delta)}{n\epsilon}\big)^2\Big\},$$

*and*

$$m \lesssim \frac{(n\epsilon)^4}{p^{10}d^2(\log(m/\delta) + R^2)(\eta T)^4 \log^2(T/\delta)}.$$

*Then, conditional on (6), for any $k = 0, \ldots, T - 1$, it holds that*

$$\mathbb{E}_{S,\widetilde{S},\mathcal{A}}\Big[\frac{1}{n}\sum_{i=1}^{n}\big\|\widetilde{\Theta}(k+1) - \widetilde{\Theta}^i(k+1)\big\|_2\Big] \leq \frac{2e\eta C_{\sigma,b}p(\sqrt{p}+R)}{n}\sum_{t=0}^{k}\mathbb{E}_{S,\mathcal{A}}\big[\mathcal{L}_S(\widetilde{\Theta}(t))\big] + 2R\delta.$$

*Proof.* All expectations below are taken conditional on the fixed initialization satisfying (6). For each $i \in [n]$, the two trajectories $\{\widetilde{\Theta}(k)\}_{k=0}^{T}$ and $\{\widetilde{\Theta}^i(k)\}_{k=0}^{T}$ are coupled with the same Gaussian perturbations. Thus the noise terms cancel in the stability recursion.

Recall that $S^{-i} = S \setminus \{z_i\}$. Since the projection is non-expansive, using Lemma D.3, the self-bounding property of $\ell$, and Lemma C.1, we obtain

$$\big\|\widetilde{\Theta}(k+1) - \widetilde{\Theta}^i(k+1)\big\|_2 \leq \big\|\widetilde{\Theta}(k) - \eta\nabla\mathcal{L}_S(\widetilde{\Theta}(k)) - \eta\mathbf{B}(k) - \widetilde{\Theta}^i(k) + \eta\nabla\mathcal{L}_{S^i}(\widetilde{\Theta}^i(k)) + \eta\mathbf{B}(k)\big\|_2$$

$$\leq \Big\|\widetilde{\Theta}(k) - \frac{\eta(n-1)}{n}\nabla\mathcal{L}_{S^{-i}}(\widetilde{\Theta}(k)) - \widetilde{\Theta}^i(k) + \frac{\eta(n-1)}{n}\nabla\mathcal{L}_{S^{-i}}(\widetilde{\Theta}^i(k))\Big\|_2 + \frac{\eta}{n}\big\|\nabla\ell\big(y_i f_{\widetilde{\Theta}(k)}(\mathbf{x}_i)\big) - \nabla\ell\big(y_i' f_{\widetilde{\Theta}^i(k)}(\mathbf{x}_i')\big)\big\|_2$$

$$\leq G_\alpha^i(k)\big\|\widetilde{\Theta}(k) - \widetilde{\Theta}^i(k)\big\|_2 + \frac{\eta}{n}\big[\ell\big(y_i f_{\widetilde{\Theta}(k)}(\mathbf{x}_i)\big)\|\nabla f_{\widetilde{\Theta}(k)}(\mathbf{x}_i)\|_2 + \ell\big(y_i' f_{\widetilde{\Theta}^i(k)}(\mathbf{x}_i')\big)\|\nabla f_{\widetilde{\Theta}^i(k)}(\mathbf{x}_i)\|_2\big]$$

$$\leq G_\alpha^i(k)\big\|\widetilde{\Theta}(k) - \widetilde{\Theta}^i(k)\big\|_2 + C_{\sigma,b}p(\sqrt{p}+R)\frac{\eta}{n}\big[\ell\big(y_i f_{\widetilde{\Theta}(k)}(\mathbf{x}_i)\big) + \ell\big(y_i' f_{\widetilde{\Theta}^i(k)}(\mathbf{x}_i')\big)\big], \tag{42}$$

where $G_\alpha^i(k) \leq 1 + M^i(k)$ with $M^i(k) = \frac{2\eta\bar{\kappa}}{\sqrt{m}}\max\big\{\mathcal{L}_{S^{-i}}(\widetilde{\Theta}(k)), \mathcal{L}_{S^{-i}}(\widetilde{\Theta}^i(k))\big\}$. Here $\bar{\kappa}$ is the curvature constant in Lemma E.7.

By Lemma E.11 applied to the two datasets $S$ and $S^i$, with failure probability $\delta/2$ for each, we obtain that with probability at least $1 - \delta$,

$$\eta\sum_{t=0}^{T-1}\mathcal{L}_S(\widetilde{\Theta}(t)) \lesssim \|\widetilde{\Theta}(0) - \Theta^*\|_2^2 + \eta T\mathcal{L}_S(\Theta^*) + \eta\mathcal{L}_S(\widetilde{\Theta}(0)) + \frac{mp^4 d\,\eta^2 T^2 \log(T/\delta)}{n^2\epsilon^2} + \frac{Rp^{3/2}\eta T \log(T/\delta)}{n\epsilon},$$

$$\eta\sum_{t=0}^{T-1}\mathcal{L}_{S^i}(\widetilde{\Theta}^i(t)) \lesssim \|\widetilde{\Theta}(0) - \Theta^*\|_2^2 + \eta T\mathcal{L}_{S^i}(\Theta^*) + \eta\mathcal{L}_{S^i}(\widetilde{\Theta}(0)) + \frac{mp^4 d\,\eta^2 T^2 \log(T/\delta)}{n^2\epsilon^2} + \frac{Rp^{3/2}\eta T \log(T/\delta)}{n\epsilon}.$$

Denote this high-probability event by $\mathcal{E}_{\text{opt}}^{(i)}$. On $\mathcal{E}_{\text{opt}}^{(i)}$, we use $\mathcal{L}_{S^{-i}}(\widetilde{\Theta}) \leq \frac{n}{n-1}\mathcal{L}_S(\widetilde{\Theta}) \leq 2\mathcal{L}_S(\widetilde{\Theta})$, and similarly $\mathcal{L}_{S^{-i}}(\widetilde{\Theta}^i) \leq 2\mathcal{L}_{S^i}(\widetilde{\Theta}^i)$. Therefore,

$$\sum_{s=0}^{T-1}M^i(s) \leq \frac{4\eta\bar{\kappa}}{\sqrt{m}}\max\Big\{\sum_{s=0}^{T-1}\mathcal{L}_S(\widetilde{\Theta}(s)), \sum_{s=0}^{T-1}\mathcal{L}_{S^i}(\widetilde{\Theta}^i(s))\Big\}$$

$$\lesssim \frac{p}{\sqrt{m}}\Big(\|\widetilde{\Theta}(0) - \Theta^*\|_2^2 + \frac{mp^4 d\,\eta^2 T^2 \log(T/\delta)}{n^2\epsilon^2} + \frac{Rp^{3/2}\eta T \log(T/\delta)}{n\epsilon}\Big)$$

$$\leq 1,$$

where the last inequality follows from the condition of $m$.

Hence, on $\mathcal{E}_{\mathrm{opt}}^{(i)}$, it holds

$$\prod_{s=t+1}^{k}(1+M^i(s)) \le \exp\Big(\sum_{s=t+1}^{k}M^i(s)\Big) \le e.$$

Unrolling the recursion (42) gives, on $\mathcal{E}_{\mathrm{opt}}^{(i)}$,

$$\big\|\widetilde{\Theta}(k+1)-\widetilde{\Theta}^i(k+1)\big\|_2 \le \frac{e\eta C_{\sigma,b}p(\sqrt{p}+R)}{n}\sum_{t=0}^{k}\Big[\ell\big(y_i f_{\widetilde{\Theta}(t)}(\mathbf{x}_i)\big)+\ell\big(y_i' f_{\widetilde{\Theta}^i(t)}(\mathbf{x}_i')\big)\Big].$$

On the complement $(\mathcal{E}_{\mathrm{opt}}^{(i)})^c$, the projection gives the crude bound $\big\|\widetilde{\Theta}(k+1)-\widetilde{\Theta}^i(k+1)\big\|_2 \le 2R$. Since $\mathbb{P}_{\mathcal{A}}((\mathcal{E}_{\mathrm{opt}}^{(i)})^c) \le \delta$, taking expectation over the Gaussian perturbations yields

$$\mathbb{E}_{\mathcal{A}}\Big[\big\|\widetilde{\Theta}(k+1)-\widetilde{\Theta}^i(k+1)\big\|_2\Big] \le \frac{e\eta C_{\sigma,b}p(\sqrt{p}+R)}{n}\sum_{t=0}^{k}\mathbb{E}_{\mathcal{A}}\Big[\ell\big(y_i f_{\widetilde{\Theta}(t)}(\mathbf{x}_i)\big)+\ell\big(y_i' f_{\widetilde{\Theta}^i(t)}(\mathbf{x}_i')\big)\Big] + 2R\delta.$$

Finally, averaging over $i \in [n]$ and taking expectation over $S, \widetilde{S}$, we get

$$\mathbb{E}_{S,\widetilde{S},\mathcal{A}}\Big[\frac{1}{n}\sum_{i=1}^{n}\big\|\widetilde{\Theta}(k+1)-\widetilde{\Theta}^i(k+1)\big\|_2\Big] \le \frac{2e\eta C_{\sigma,b}p(\sqrt{p}+R)}{n}\sum_{t=0}^{k}\mathbb{E}_{S,\mathcal{A}}\big[\mathcal{L}_S(\widetilde{\Theta}(t))\big] + 2R\delta,$$

where we used $\mathbb{E}_{S,\widetilde{S},\mathcal{A}}\big[\mathcal{L}_{S^i}(\widetilde{\Theta}^i(t))\big] = \mathbb{E}_{S,\mathcal{A}}\big[\mathcal{L}_S(\widetilde{\Theta}(t))\big]$. This completes the proof. $\qquad\square$

**Theorem E.13.** *Suppose* (6) *and Assumptions 4.1 and 4.2 hold. For any reference point* $\Theta^* = (\mathbf{a}^*,\mathbf{c}^*)$ *with* $\mathbf{a}^* \in \Omega_{\mathbf{a}}$ *and* $\mathbf{c}^* \in \Omega_{\mathbf{c}}$, *assume* $\|\widetilde{\Theta}(0)-\Theta^*\|_2^2 \ge C\max\{\eta T(\mathcal{L}_S(\Theta^*)+\mathcal{L}_{\widetilde{S}}(\Theta^*)), \eta(\mathcal{L}_S(\widetilde{\Theta}(0))+\mathcal{L}_{\widetilde{S}}(\widetilde{\Theta}(0)))\}$. *Suppose* $\eta \le \min\{1/3\bar\rho, 1\}$, $m \gtrsim p^2(\log(\frac{m}{\delta})+R^2)\max\big\{R^4+\|\widetilde{\Theta}(0)-\Theta^*\|_2^4, (\frac{Rp^{3/2}\eta T\log(T/\delta)}{n\epsilon})^2\big\}$ *and* $m \lesssim \frac{(n\epsilon)^4}{p^{10}(\log(\frac{m}{\delta})+R^2)d^2(\eta T)^4\log^2(T/\delta)}$. *Then, for any* $k \in [T]$, *it holds that*

$$\mathbb{E}_{S,\mathcal{A}}\big[\mathcal{L}(\widetilde{\Theta}(k))-\mathcal{L}_S(\widetilde{\Theta}(k))\big] \lesssim \frac{p^2 m^{1/4}\|\widetilde{\Theta}(0)-\Theta^*\|_2^2}{n} + \frac{mp^6 d\eta^2 T^2\log(T/\delta)}{n^3\epsilon^2} + p^{\frac{3}{2}}\delta.$$

*Proof.* We will use Lemma E.12 to prove the theorem. The lower-width condition controls the deterministic localization terms and the martingale fluctuation term in the high-probability optimization bound, while the upper-width condition controls the usual DP noise-square term.

Note that the condition for $m$ implies $C_1 = 8(B_\ell' B_\sigma' B_b' B_b)^2 p^2\big(4\sqrt{p}+\frac{2\sqrt{\log(2/\delta)}+R_2}{\sqrt{m}}\big)^2 \lesssim p^3$ and $C_2 = 8(B_\ell' B_b)^2 p \lesssim p$. For any $S$ and the trajectory produced by Algorithm 1, (41) and the comparator condition imply

$$\eta\sum_{k=1}^{T}\mathbb{E}_{\mathcal{A}}\big[\mathcal{L}_S(\widetilde{\Theta}(k))\big] \lesssim \|\widetilde{\Theta}(0)-\Theta^*\|_2^2 + \frac{mp^4 d\eta^2 T^2\log(T/\delta)}{n^2\epsilon^2}. \tag{43}$$

The same bound holds with $S$ replaced by $S^i$, using $\mathcal{L}_{S^i}(\Theta^*) \le \mathcal{L}_S(\Theta^*)+\mathcal{L}_{\widetilde{S}}(\Theta^*)$ and similarly for $\widetilde{\Theta}(0)$.

The assumptions on $m$ ensure that the width condition required in Lemma E.12 is satisfied. Therefore Lemma E.12, together with (43), gives the following on-average argument stability bound

$$\epsilon_{\mathrm{stab}} \lesssim \frac{\|\widetilde{\Theta}(0)-\Theta^*\|_2^2}{n} + \frac{mp^4 d\eta^2 T^2\log(T/\delta)}{n^3\epsilon^2} + \delta.$$

From Lemma C.1, the condition on $m$, and the fact that the iterates stay in the projected set, the loss is Lipschitz with respect to the parameter, it holds

$$\sup_{(\mathbf{x},y)\in\mathcal{X}\times\mathcal{Y}}\big|\ell'(y f_{\widetilde{\Theta}(k)}(\mathbf{x}))\big|\big\|\nabla f_{\widetilde{\Theta}(k)}(\mathbf{x})\big\|_2 \le C_{\sigma,b}p(\sqrt{p}+R).$$

By Lemma D.2, we have

$$\mathbb{E}_{S,\mathcal{A}}\big[\mathcal{L}(\widetilde{\Theta}(k)) - \mathcal{L}_S(\widetilde{\Theta}(k))\big] \lesssim p(\sqrt{p} + R)\Big(\frac{\|\widetilde{\Theta}(0) - \Theta^*\|_2^2}{n} + \frac{mp^4 d\eta^2 T^2 \log(T/\delta)}{n^3\epsilon^2} + R\delta\Big)$$

$$\lesssim \frac{p^2 m^{1/4}\|\widetilde{\Theta}(0) - \Theta^*\|_2^2}{n} + \frac{mp^6 d\eta^2 T^2 \log(T/\delta)}{n^3\epsilon^2} + p^{\frac{3}{2}}\delta,$$

where the last inequality uses $m \gtrsim p^2\big(\log(\frac{m}{\delta}) + R^2\big)\big(R^2 + \|\widetilde{\Theta}(0) - \Theta^*\|_2^4\big)$, which implies $R \lesssim m^{1/4}$ and $\sqrt{p} \lesssim m^{1/4}$. This completes the proof. $\qquad\square$

**Theorem E.14** (Restatement of Theorem 4.15)**.** *Let the sequence $\{\widetilde{\Theta}(k)\}_{k=1}^T$ be produced by Algorithm 1 with step size $\eta > 0$. Let $\Theta^*$ be a reference point satisfying $\|\widetilde{\Theta}(0) - \Theta^*\|_2^2 \geq C \max\{\eta T\big(\mathcal{L}_S(\Theta^*) + \mathcal{L}_{\widetilde{S}}(\Theta^*)\big), \eta\big(\mathcal{L}_S(\widetilde{\Theta}(0)) + \mathcal{L}_{\widetilde{S}}(\widetilde{\Theta}(0))\big)\}$. If $\eta \leq \min\{1/3\bar{\rho}, 1\}$, $m \gtrsim p^2\big(\log(\frac{m}{\delta}) + R^2\big)\max\big\{R^4 + \widetilde{\Lambda}_{\Theta^*}^4, \big(\frac{Rp^{3/2}\eta T \log(T/\delta)}{n\epsilon}\big)^2\big\}$ and $m \lesssim \frac{(n\epsilon)^4}{p^{10}d^2(\log(m/\delta) + R^2)(\eta T)^4 \log^2(T/\delta)}$. Then with probability at least $1 - \delta$ over the randomness of the initialization, it holds that*

$$\frac{1}{T}\sum_{k=1}^T \mathbb{E}_{S,\mathcal{A}}\big[\mathcal{L}(\widetilde{\Theta}(k))\big] \lesssim \big(\frac{1}{\eta T} + \frac{m^{1/4}}{n}\big)\widetilde{\Lambda}_{\Theta^*}^2 + \big(1 + \frac{\eta T}{n}\big)\frac{m\eta T d \log(T/\delta)}{n^2\epsilon^2} + p^{\frac{3}{2}}\delta.$$

*Furthermore, by setting $\eta T \asymp \frac{c_0 n\epsilon}{\sqrt{d} \log^\alpha(n/\delta)}$ for $\alpha > 1$ and $c_0 \in (0, 1]$, and assuming $\delta \lesssim \frac{\sqrt{d}}{n\epsilon p^{3/2}}$, it holds that*

$$\frac{1}{T}\sum_{k=1}^T \mathbb{E}_{S,\mathcal{A}}\big[\mathcal{L}(\widetilde{\Theta}(k))\big] \lesssim \Big(\frac{\log^\alpha(\frac{n}{\delta})}{c_0}\widetilde{\Lambda}_{\Theta^*}^2 + \frac{\log^{3\alpha-1}(\frac{n}{\delta})}{c_0^3 R^2}\Big)\frac{\sqrt{d}}{n\epsilon}.$$

*Proof.* Let $\mathcal{E}_{\mathrm{init}}$ denote the initialization event in (6). By Corollary B.2, we have

$$\mathbb{P}(\mathcal{E}_{\mathrm{init}}) \geq 1 - \delta.$$

We prove the desired bound on $\mathcal{E}_{\mathrm{init}}$. Conditional on this event, all expectations $\mathbb{E}_{\mathcal{A}}$ are taken only over the Gaussian perturbations generated after initialization.

On $\mathcal{E}_{\mathrm{init}}$, combining Theorems E.10 and E.13 gives

$$\frac{1}{T}\sum_{k=1}^T \mathbb{E}_{S,\mathcal{A}}\big[\mathcal{L}(\widetilde{\Theta}(k))\big] = \frac{1}{T}\sum_{k=1}^T \mathbb{E}_{S,\mathcal{A}}\big[\mathcal{L}(\widetilde{\Theta}(k)) - \mathcal{L}_S(\widetilde{\Theta}(k))\big] + \frac{1}{T}\sum_{k=1}^T \mathbb{E}_{S,\mathcal{A}}\big[\mathcal{L}_S(\widetilde{\Theta}(k))\big]$$

$$\lesssim \big(\frac{1}{\eta T} + \frac{m^{1/4}}{n}\big)\widetilde{\Lambda}_{\Theta^*}^2 + \big(1 + \frac{\eta T}{n}\big)\frac{m\eta T d \log(T/\delta)}{n^2\epsilon^2} + p^{\frac{3}{2}}\delta.$$

This proves the first claim on $\mathcal{E}_{\mathrm{init}}$. Since $\mathbb{P}(\mathcal{E}_{\mathrm{init}}) \geq 1 - \delta$, the first claim holds with probability at least $1 - \delta$ over the initialization.

Now set $\eta T \asymp \frac{c_0 n\epsilon}{\sqrt{d} \log^\alpha(n/\delta)}$. Then $\frac{1}{\eta T} \asymp \frac{\sqrt{d} \log^\alpha(n/\delta)}{c_0 n\epsilon}$. Moreover, the upper-width condition gives

$$m \lesssim \frac{(n\epsilon)^4}{p^{10}d^2(\log(m/\delta) + R^2)(\eta T)^4 \log^2(T/\delta)} \lesssim \frac{\log^{4\alpha-2}(n/\delta)}{c_0^4 R^2}.$$

Hence

$$\frac{m^{1/4}}{n} \lesssim \frac{\log^{\alpha-\frac{1}{2}}(n/\delta)}{c_0 n R^{1/2}}.$$

Therefore,

$$\frac{1}{T}\sum_{k=1}^T \mathbb{E}_{S,\mathcal{A}}\big[\mathcal{L}(\widetilde{\Theta}(k))\big] \lesssim \Big(\frac{\sqrt{d} \log^\alpha(n/\delta)}{c_0 n\epsilon} + \frac{\log^{\alpha-\frac{1}{2}}(n/\delta)}{c_0 n}\Big)\widetilde{\Lambda}_{\Theta^*}^2 + \frac{\sqrt{d} \log^{3\alpha-1}(n/\delta)}{c_0^3 R^2 n\epsilon} + p^{\frac{3}{2}}\delta.$$

Under the assumption $\delta \lesssim \frac{\sqrt{d}}{n\epsilon p^{3/2}}$ and using $\alpha > 1$ and $\widetilde{\Lambda}_{\Theta^*} \geq 1$, the right-hand side is bounded by

$$\left(\frac{\log^\alpha(\frac{n}{\delta})}{c_0}\widetilde{\Lambda}_{\Theta^*}^2 + \frac{\log^{3\alpha-1}(\frac{n}{\delta})}{c_0^3 R^2}\right)\frac{\sqrt{d}}{n\epsilon}.$$

This completes the proof. $\qquad\square$

### E.4. Proofs under NTK Separability

**Theorem E.15** (Restatement of Theorem 4.16)**.** *Let Assumptions 4.1, 4.2 and 4.8 hold. Assume $\eta \lesssim \left(\log(n/\delta)\right)^{-1/2}$ be a constant, $\eta T \asymp \frac{\gamma^2 n\epsilon}{\sqrt{d}\log^{5/2}(n/\delta)}$ and $m \asymp \frac{\log^6(n/\delta)}{\gamma^6}$. Let $R \asymp \frac{\log^{1/2}(n/\delta)}{\gamma}$ and $\delta \lesssim \frac{\sqrt{d}}{n\epsilon p^{3/2}}$. Then, with probability at least $1 - \delta$ over the initialization $\mathbf{c}(0)$, it holds that*

$$\frac{1}{T}\sum_{k=1}^T \mathbb{E}_{S,\mathcal{A}}\left[\mathcal{L}(\widetilde{\Theta}(k))\right] \lesssim \log^6\left(\frac{n}{\delta}\right)\frac{\sqrt{d}}{\gamma^4 n\epsilon}.$$

*Proof.* Note that in the proof of Theorem 4.9 we showed that with probability at least $1 - \delta$ over $\mathbf{c}(0)$, it holds that $\mathcal{L}_S(\Theta_\tau) \leq \frac{1}{T}$ when we set $\Theta_\tau = \Theta(0) + \tau\Theta_0$ and $\tau \asymp \left(\log(T) + \sqrt{\log(n/\delta)}\right)/\gamma$. Here, $\Theta_0$ and $\gamma$ are the parameter and the margin in Assumption 4.8.

Setting $R_1 + R_2 = R \asymp \tau \asymp \left(\log(T) + \sqrt{\log(n/\delta)}\right)/\gamma$ and the reference point $\Theta^* = \Theta_\tau \in \mathcal{B}(\Theta(0), R)$. Then, $\widetilde{\Lambda}_{\Theta^*} = \|\Theta^* - \widetilde{\Theta}(0)\|_2 = \tau$.

Now, we show that the conditions in Theorem 4.15 are satisfied. Let $c_0 \asymp \gamma^2 \in (0, 1]$ and $\alpha = 5/2$.

Then, we know $m \gtrsim \left(\log(\frac{m}{\delta}) + R^2\right)(R^2 + \widetilde{\Lambda}_{\Theta^*}^6) \asymp \log^6(n/\delta)/\gamma^6$ and $m \lesssim \frac{(n\epsilon)^4}{p^{10}(\log(\frac{m}{\delta})+R^2)d^2(\eta T)^4 \log^2(T/\delta)} \asymp \log^6(n/\delta)/\gamma^6$. These matches the width condition stated in the theorem.

Note $\widetilde{S}$ is an independent copy of $S$, one can also show that $\mathcal{L}_{\widetilde{S}}(\Theta^*) \leq \frac{1}{T}$. Since $\mathcal{L}_S(\Theta^*) + \mathcal{L}_{\widetilde{S}}(\Theta^*) \leq \frac{2}{T}$, we have $\eta T\left(\mathcal{L}_S(\Theta^*)+\mathcal{L}_{\widetilde{S}}(\Theta^*)\right) \lesssim \eta$. Moreover, by the initialization bound used in the proof of Theorem 4.9, $\mathcal{L}_S(\widetilde{\Theta}(0))+\mathcal{L}_{\widetilde{S}}(\widetilde{\Theta}(0))$ is at most logarithmic in $n/\delta$. Together with $\eta \lesssim \log^{-1/2}(n/\delta)$, this term is dominated by $\widetilde{\Lambda}_{\Theta^*}^2 = \|\Theta^* - \widetilde{\Theta}(0)\|_2^2 \asymp \frac{\log(n/\delta)}{\gamma^2}$. Therefore the comparator condition in Theorem 4.15 is satisfied.

Noting that $\delta \lesssim \frac{\sqrt{d}}{n\epsilon p^{3/2}}$. Applying Theorem 4.15 with $\alpha = 5/2$, $c_0 = \gamma^2$ and $R^2 \asymp \tau^2 = \widetilde{\Lambda}_{\Theta^*}^2 \gtrsim \log(n/\delta)/\gamma^2$, we know

$$\frac{1}{T}\sum_{k=1}^T \mathbb{E}_{S,\mathcal{A}}\left[\mathcal{L}(\widetilde{\Theta}(k))\right] \lesssim \log^6\left(\frac{n}{\delta}\right)\frac{\sqrt{d}}{\gamma^4 n\epsilon}.$$

This completes the proof of the theorem. $\qquad\square$

## F. Detailed Experiments

This section provides the detailed DP-GD algorithm and describes the experimental details and hyperparameters.

### F.1. Datasets

We consider a synthetic binary classification dataset and MNIST (Deng, 2012) for our experiments.

**Synthetic Logistic Dataset.** We generate a challenging dataset $\{(x_i, y_i)\}_{i=1}^n$ with $x_i \in [-1, 1]^d$ uniformly sampled. The label distribution is defined through a logistic model

$$y_i \sim \text{Bernoulli}\left(\sigma(s \cdot h(x_i) + \xi_i)\right),$$

where $\sigma(\cdot)$ denotes the sigmoid, $s > 0$ is a signal strength parameter, and $\xi_i \sim \mathcal{N}(0, \sigma_\xi^2)$ is optional label noise. The latent score $h(x)$ is created in a spline-like manner with random coefficients:

$$h(x_i) = \sum_{j=1}^{d} u_j(x_{i,j}), \qquad u_j(x_{i,j}) = \sum_{\ell=1}^{k} \theta_{j\ell}\, b_\ell(x_{i,j}).$$

The basis functions $b_\ell(\cdot)$ are triangular functions centered at uniformly spaced knots $\{t_\ell\}_{\ell=1}^{k} \subset [-1, 1]$,

$$b_\ell(x) = \max\left(1 - \frac{|x - t_\ell|}{\Delta},\, 0\right),$$

with $\Delta = \frac{2}{k-1}$ denoting the knot spacing and $\theta_{j\ell} \sim \mathcal{N}(0, 1)$ being random coefficients. For our experiments, we set $s = 4$, $d = 10$, $\sigma_\xi^2 = 0.1$, and $k = 40$.

**MNIST.** We transform MNIST into a binary classification dataset by restricting it to the first two classes (digit zero and digit one).

## F.2. Model Hyperparameters

We study the two-layer KAN described in the main paper. Unless otherwise specified, we set the model width to $m = 32$, the number of splines to $p = 8$, and the number of full-pass iterations to $T = 100$. We use a learning rate of $\eta = 0.5$ for MNIST and $\eta = 1$ for the synthetic logistic data. The larger step size for the logistic data is chosen to speed up the experiments. We observe similar qualitative trends when using smaller step sizes.

For the synthetic logistic data, we use $n = 20,000$ training samples and $8,000$ test samples. For MNIST, we use the full training and test sets restricted to the two classes under consideration. Unless otherwise specified, we set the gradient ball bounds to $R_1 = R_2 = 1$, the privacy budget to $\epsilon = 2.0$, and $\delta = 1/n$.

## F.3. Experiments

The paper shows results for two types of experiments.

**Loss over $m$.** Here, we vary the model width $m$ and report the resulting training and test metrics of the model for each 50 random seeds on the synthetic data and 20 random seeds on MNIST.

**Loss over $T$.** Here, we vary the full training dataset iterations $T$ and report the resulting training and test metrics of the model for each 50 random seeds on the synthetic data and 20 random seeds on MNIST. For MNIST, we use full-batch GD, which incurs a higher computational cost as the width $m$ increases, hence MNIST experiments are restricted to widths up to $m = 32$.

## F.4. Loss curves

Figure 3 presents loss curves, analogous to the accuracy curves in the main paper.

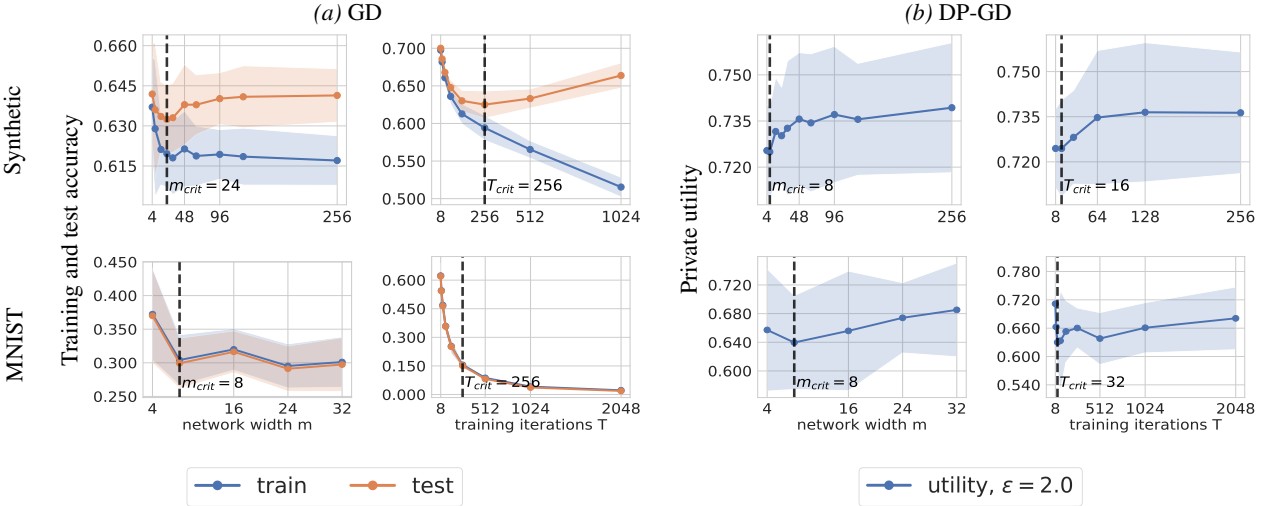

*Figure 3.* Training and test losses versus $m$ and $T$.

