# OpenReview forum: "Optimization,  Generalization and Differential Privacy Bounds for Gradient Descent on Kolmogorov–Arnold Networks"
_ICML.cc/2026/Conference — ICML 2026 regular_

### Official Review · Reviewer_y8GA · 2026-03-12

**Soundness:** 4
**Presentation:** 4
**Significance:** 3
**Originality:** 4
**Overall Recommendation:** 5
**Confidence:** 3

**Summary:**

KANs form an alternative to MLPs and show a promising utility improvement.

The paper analyses the utility of GD and DP-GD for training two-layer Kolmogorov-Arnold Networks (KANs) by providing a unified (for both GD and DP-GD) reference-point analysis given NTK separability. The two-layer KAN is given by (2) where the univariate edge functions are a B-spline basis.

The results show that for polylog width (sufficient condition) GD achieve optimization risk Otilde(1/T) and population risk Otilde(1/n) for T iterations and a training data set size n. Here, the population risk is given by an algorithm-dependent generalization bound.  The generalization gap is controlled by the cumulative training loss along the GD trajectory and is small for small training loss. The reference point technique also shows implicit regularization of GD favoring predictors with low generalization risk that remain close to initialization.  (The results are first proven for a reachability assumption defined by a function evaluation g(1/T) which is characterized for the NTK assumption in the appendix.)

For DP-GD (with clipping/projection after adding DP noise) shows a privacy-utility tradeoff Otilde(sqrt{d}/(n eps)) for the averaged population risk along the DP-GD trajectory where eps defines the (eps,delta)-DP guarantee and d is the number of model parameters. This trade-off holds if the width is at least polylog (necessary condition).

The theory has a number of practical implications: (1) Training and test accuracy saturate beyond a critical width of typically order polylog(n). Beyond this regime both training and test accuracy plateau. (2) Training accuracy may continue to improve for increasing T. This benefits optimization rather than generalization since (3) test accuracy saturates with increasing T (there is critical threshold T_{crit}). (4) Private utility is maximized when width and training iterations lie in an admissible range (characterized by the theoretical analysis: polylog width and T=Otilde(n eps/sqrt{d})).

These behaviors as predicted by the theoretical analysis are verified and confirmed by experiments.

**Compliance With Llm Reviewing Policy:**

Affirmed.

**Final Justification:**

I am satisfied with the author's answers and the reported results and the insights that come from their interpretation, and still recommend accept.

**Key Questions For Authors:**

Please, explain the effect of small epsilon on the parameter setting and whether this leads to practical scenarios.

How does the 2-layer KAN relate to a corresponding MLP?

Which math hurdles make the current analysis not suitable for multi-layer KANs?

**Limitations:**

yes

**Strengths And Weaknesses:**

Excellent exposition with intuitive explanations. Sound analysis. A first theoretical framework that analysis a privacy-utility tradeoff for KANs leading to insights that are confirmed empirically.

From a DP perspective, we do understand that a practical privacy-utility tradeoff is possible for low complex models and training/test data. The 2-layer KAN is low complex. The main question is how the privacy-utility tradeoff deteriorates for a multi-layer KAN (since multiple layers make the model training less stable against noise).

Related to a practical privacy-utility trade-off is that we require practical utility as well as a reasonable DP guarantee and eps=2.0 is *not* reasonable. One can verify that this reduces (in the DP proof) to an adversary who has a significant advantage in solving a corresponding hypothesis testing problem.  The advantage is significant in that the false positive and false negative curve is far below the random guessing line which defines perfect privacy. So, if eps=0.5 (still not reasonable from a privacy perspective) or eps=0.05 (much better!), then what parameters d, n, etc. will lead to practical utility? And do such parameters make sense in practice?

The motivation for KAN is their improvement over MLP. The paper does not compare against the utility of training a corresponding MLP with GD or DP-GD in order to verify that a two-layer KAN also is the preferred choice over a corresponding MLP in practice.

---

> ### Author Rebuttal · Authors · 2026-03-31
>
> Thank you for your careful reading and constructive comments.
>
> **Q1. Effect of small $\epsilon$ on practical utility.**
>
> We thank the reviewer for this important question. Stronger privacy budgets are clearly important in practice, and one goal of our theory is to make the dependence on $\epsilon$ explicit. In particular, Theorem 4.16 gives a utility bound of order $\widetilde O(\sqrt{d}/(n\epsilon))$ and suggests an effective training horizon $T$ of order $n\epsilon/\sqrt{d}$. Therefore, as $\epsilon$ becomes smaller, the utility bound becomes worse at rate $1/\epsilon$, and the admissible training horizon decreases proportionally. For example, relative to $\epsilon=2$, using $\epsilon=0.5$ changes both quantities by roughly a factor of $4$, while $\epsilon=0.05$ changes them by roughly a factor of $40$. This suggests that stronger privacy is still compatible with useful performance, but in a more favorable regime, namely with larger $n$, smaller effective dimension $d$, polylogarithmic width, and more conservative training. In this sense, the choice $\epsilon=2$ in our experiments is mainly illustrative, serving to validate the predicted tradeoff rather than to suggest a universal privacy target.
>
> **Q2. Relation between a two-layer KAN and a corresponding MLP.**
>
> We thank the reviewer for this important and valuable question. In the paper, KANs are motivated as a structured alternative to MLPs with growing empirical interest, while their optimization, generalization, and privacy behavior under GD training remains much less understood. Our main goal in this work is to help fill this theoretical gap by providing a unified GD/DP-GD analysis for two-layer KANs. In this sense, MLPs serve as the natural reference architecture, whereas KANs introduce a different inductive bias through learnable univariate edge functions, which makes them both practically interesting and analytically nontrivial. Our contribution is therefore to provide the first unified GD/DP-GD theory for this KAN setting.
>
> **Q3. Math hurdles for extending to multi-layer KANs.**
>
> Extending the current analysis to multi-layer KANs faces two main mathematical hurdles. First, our proof for two-layer KANs relies on precise Hessian control along the whole GD trajectory. Even in the two-layer case, the Hessian bounds already depend on the distance from initialization, so we need a self-consistent argument to keep the iterate in a region where the loss remains well behaved. In multi-layer KANs, this becomes much harder because the hidden features are nested compositions of learnable edge functions, and the Hessian then contains many cross-layer interaction terms that are no longer controlled by a single deviation quantity.
>
> Second, the DP analysis becomes significantly more difficult in deeper KANs. The privacy noise injected at each step affects all later layers through repeated composition, so both the gradient sensitivity and the accumulation of noise are amplified across layers. As a result, the two-layer sensitivity and utility argument does not extend directly. For these reasons, handling multi-layer KANs would require new curvature and sensitivity estimates beyond the present framework.

---

> > ### Author Rebuttal · Reviewer_y8GA · 2026-04-03
> >
> > Thank you for your satisfying answers.

---

### Official Review · Reviewer_a6YE · 2026-03-12

**Soundness:** 3
**Presentation:** 3
**Significance:** 1
**Originality:** 2
**Overall Recommendation:** 4
**Confidence:** 4

**Summary:**

This paper proposes a differentially private learning algorithm for Kolmogorov-Arnold Networks (KANs) and provides theoretical analysis of the optimization and generalization properties of gradient descent, along with privacy guarantees for the proposed method.
The theoretical analysis further provides practical guidance for achieving good inference performance. Experimental results illustrate the theoretical findings and show that the proposed method performs well in practice.

**Compliance With Llm Reviewing Policy:**

Affirmed.

**Final Justification:**

This paper has certain degree of novelty with theoretical results that are generally sound. The authors' rebuttal has satisfactorily addressed several of my main concerns, and I am willing to raise my score from 3 to 4.

**Key Questions For Authors:**

1. Does the gradient-based KANs method require any assumptions on the initialization distribution? In the analysis, the parameters $(\\mathbf{a}(0),\\mathbf{c}(0))$ are initialized from a zero-mean normal distribution. Is this assumption essential for the theoretical results, or would the analysis still hold under other initialization schemes?

2. The realizability assumption involves
$$
g(\\epsilon):=\\inf \\{\\|\\Theta - \\Theta(0)\\|_2: L_S(\\Theta) \\leq \\epsilon\\}.
$$
In the proof, $\\epsilon = 1/T$, which means the initialization must be close to a near-optimal solution when $T$ is sufficiently large. Is this assumption expected to hold in practical scenarios?

3. The selection of the reference point appears to be crucial for the resulting error bounds. How can one ensure the selection of a suitable reference point in practice? Could the authors provide some discussion or examples of situations in which
$$
\\inf_{\\Theta}\\left\\{\\mathcal{L}(\\Theta)+\\frac{1}{2 \\eta T}\\|\\Theta-\\Theta(0)\\|_2^2\\right\\}
$$
is small?

4. According to Assumption 1, the basis function $b$ is bounded. Does this assumption imply that the target function $f$ is also bounded?

5. In the simulation study, the projection radii are chosen as $R_1 = R_2 = 1$. Is this a standard or canonical choice in this setting? How sensitive are the empirical results to different values of $R_1$ and $R_2$? Some discussion on the role of these parameters would help interpret the experimental design.

**Limitations:**

Yes

**Strengths And Weaknesses:**

The paper presents a theoretically grounded study of differentially private training for Kolmogorov–Arnold Networks. A main strength of the work is that it provides a theoretical analysis covering optimization, generalization, and privacy guarantees within a unified framework. From a technical perspective, the proofs appear generally correct. The authors develop a theoretical framework analyzing the optimization and generalization behavior of gradient descent and extend the analysis to derive privacy and utility guarantees. The lemma arguments are clearly structured and the theoretical statements are presented rigorously. In addition, the experimental results provide empirical support for the theoretical findings.

However, the practical impact of the theoretical results may be limited. The analysis relies on several assumptions that are difficult to verify in practical scenarios, which could limit the applicability of the theoretical guarantees.

The paper follows an overview–detail structure that helps convey the main ideas before introducing the technical details. Nevertheless, certain parts of the exposition could be clarified. In particular, the connection between the earlier analyses of optimization and generalization errors and the later results on private learning (e.g., Theorems 4.14-4.16) is not fully transparent, and additional discussion on the motivation and role of these intermediate results would improve readability. There are also minor notational issues: For example, the use of the term $\\mathbb{E}_S \\mathcal{L}(\\Theta)$ in Theorems 4.10-4.12 may be confusing given that $\\mathcal{L}(\\Theta)$ is already defined as $\mathbb{E}\_{(\mathbf{x},y)\sim P}[\mathcal{L}_S(\Theta)]$.

The paper makes a substantial effort to analyze the proposed method from several theoretical perspectives and provides algorithm-dependent error bounds and utility guarantees. However, the strength of the theoretical conclusions depends heavily on quantities  appearing in the optimization and generalization bound, particularly the term $\\Lambda_{\\Theta^*}$, which relies on the choice of a reference point and the initialization. In finite-sample settings, if the initialization is not sufficiently close to the target parameter, the resulting bound may become loose. Furthermore, some constants appearing in the privacy noise scale, such as $B_b$ and $B_b'$, may be difficult to determine in practice.

Overall, while the paper offers useful insights into gradient descent behavior for two-layer KANs and applies Rényi differential privacy in this setting, the core algorithmic mechanism of adding noise to gradients for privacy follows existing approaches. More intrinsic novelty in the algorithm design or in the theoretical analysis (with respect to the initialization and reference point) would be expected.

---

> ### Author Rebuttal · Authors · 2026-03-31
>
> We thank the reviewer for the thoughtful comments. Before responding to the specific questions, we would like to briefly clarify the scope and contribution of the paper.
>
> **Contribution and novelty.**
>
> The paper is not only about a private training algorithm for KANs. Its central contribution is a unified framework for **both non-private GD and DP-GD** on two-layer KANs. On the non-private side, this yields, to the best of our knowledge, the first optimization and algorithm-dependent generalization analysis for gradient-based training of two-layer KANs, including optimization and population-risk guarantees in the polylogarithmic-width regime. On the private side, although the DP mechanism itself is standard, the resulting theory is new: to the best of our knowledge, this is the first utility analysis of DP-GD for KANs. More broadly, rigorous utility guarantees for private gradient methods on neural networks remain very limited, and existing results for MLPs typically rely on over-scaled regimes, convex reformulations, or stationarity-type guarantees rather than explicit utility bounds. By contrast, our results hold under the standard scaling, match the classical convex Lipschitz benchmark, and provide an essentially sharp width characterization. In this sense, the paper develops a KAN-specific theoretical framework that unifies optimization, algorithm-dependent generalization, and privacy utility, and identifies practically relevant regimes for width and training horizon.
>
> **Q1. Does the analysis of KANs require Gaussian initialization, or could it extend to other schemes?**
>
> The current analysis is developed under zero-mean Gaussian initialization, which is standard in practice and widely adopted in existing neural network theory, including recent KAN analyses. In our paper, some key estimates, especially the Hessian and GD trajectory controls, rely on Gaussian properties. That said, the main role of this assumption is to provide a well-controlled starting point for the local analysis. In principle, the same framework could be extended to other initialization schemes, although some technical estimates would need to be rederived.
>
> **Q2.  Does the realizability assumption mean that the initialization must already be close to a near-optimal solution, especially when $\epsilon=1/T$ and $T$ is large?**
>
> No. The condition with $\epsilon=1/T$ does *not* mean that the initialization itself must be close to a near-optimal solution. Rather, the assumption requires that, for each sufficiently small $\epsilon$, there exists a comparator $\Theta^\epsilon$ with $\mathcal L_S(\Theta^\epsilon)\le \epsilon$, while $g(\epsilon)$ measures its distance from initialization. Hence, taking $\epsilon=1/T$ only means that there exists a comparator with empirical risk at scale $1/T$. In our paper, this realizability condition is further verified under NTK separability, under which we show that $g(1/T)$ grows only logarithmically in $T$. Thus, this is a local reachability condition, not a near-optimality assumption on $\Theta(0)$. We also discuss the plausibility of the NTK assumption in our response to Reviewer bMLB.
>
> **Q3. How can one ensure a suitable reference point in practice, and when is the corresponding population-side quantity expected to be small?**
>
> We would like to clarify that the reference point is an analytical device and is *not* an explicit object that the algorithm must compute in practice. In the NTK-separable regime studied in the paper, the theory (Theorem 4.12) guarantees the existence of a suitable reference point for which the corresponding population-side quantity is small.
>
> **Q4. Does the basis function bounded assumption imply that the target function is also bounded?**
>
> No. The boundedness of the basis $b$ does not by itself imply that the target function is bounded, its role is to control the model class and related derivatives.
>
> **Q5. Are the choices $R_1=R_2=1$ standard or canonical, and how do different projection radii affect the empirical results and their interpretation?**
>
> The choices of $R_1$ and $R_2$ are not canonical, but simply one concrete choice used in our experiments. In the NTK-separable regime, the proof suggests that $R_1+R_2$ should be chosen on the scale of the reference-point complexity. Larger radii make the width condition more restrictive and can require more iterations $T$. We will clarify this dependence in the revised version.
>
>
> **Additional clarification on the noise scale.**
>
> In practice, the spline basis $b$ is fixed first, typically to a cubic basis. Once $b$ is fixed and the data are normalized, the corresponding constants $B_b$ and $B_b'$ are determined, and so is the privacy noise scale.

---

> > ### Author Rebuttal · Reviewer_a6YE · 2026-04-02
> >
> > I revisited the manuscript and found its logical structure as clarified. Overall, the authors’ rebuttal addressed most of my concerns. However, the presentation could be further improved if the authors explicitly clarified the specific form of the reference points before stating the instantiated theorems. For instance,  $\\Theta_\\tau$  could be introduced immediately after Theorem 4.11 as a possible choice of reference point.

---

> > > ### Author Response · Authors · 2026-04-03
> > >
> > > Thank you very much for revisiting the manuscript and for the helpful follow-up suggestion.
> > >
> > > We will revise the presentation so that a concrete candidate reference point is introduced immediately after Theorem 4.11 in the revised version. We agree that this would make the logical flow clearer for the reader.
> > >
> > > Thank you again for the constructive suggestion.

---

### Official Review · Reviewer_bMLB · 2026-03-15

**Soundness:** 3
**Presentation:** 3
**Significance:** 3
**Originality:** 3
**Overall Recommendation:** 4
**Confidence:** 4

**Summary:**

This paper provides the first theoretical analysis of gradient descent and differentially private gradient descent for training two-layer Kolmogorov-Arnold Networks with B-spline parameterization. Under NTK separability assumptions, the authors establish:

- Optimization: $O(1/T)$ convergence rate with polylogarithmic width
- Generalization: $O(1/n)$ population risk bound
- Privacy utility: $O(\sqrt{d}/(n\varepsilon))$ bound for $(\varepsilon,\delta)$-DP, matching classical lower bounds for convex Lipschitz problems

A key finding is that polylogarithmic width is both sufficient and necessary for DP-GD.

**Compliance With Llm Reviewing Policy:**

Affirmed.

**Final Justification:**

My questions are addressed and I will maintain the positive rating.

**Key Questions For Authors:**

Is NTK separability achievable for KAN in practical datasets? Are there any empirical gap between theoretical critical widths and practically optimal widths?

**Limitations:**

Yes, conclusion section acknowledge limitations such as extending the analysis beyond two-layer KANs, relaxing the smoothness assumption on σto cover non-smooth activations such as ReLU, and generalizing our guarantees from GD to SGD.

**Strengths And Weaknesses:**

Strengths:
1. First algorithm-dependent generalization bounds for KANs and first utility analysis of DP gradient methods for KANs
2. Proves that DP-GD needs polylogarithmic width (upper and lower bounded from both sides) to reach desired utility guarantee, by contrast non-private GD needs at least polylogarithmic width (no upper bound).
3. Reference-point analysis handles both GD and DP-GD with trajectory-wise curvature control


Weaknesses:
1. The NTK separability assumption: the authors note that it is weaker than positive-definiteness of gradient gram matrix, but more explanations on whether it is achievable under KAN would clarify the significance of the theoretical analysis.

---

> ### Author Rebuttal · Authors · 2026-03-31
>
> Thank you for your careful reading and constructive comments.
>
> **Q1. Practical Achievability of NTK Separability**
>
> We thank the reviewer for the thoughtful comment. We agree that the significance of the NTK separability assumption depends on whether it can plausibly arise for KANs. As already noted in the paper, this assumption is weaker than full NTK Gram-matrix positive-definiteness. Moreover, separability assumptions of this type are standard in the NTK theory of ReLU and smooth MLPs and have been rigorously verified in several settings.
>
> To further address achievability for KANs, we conducted additional experiments on synthetic *nonlinearly separable* datasets and compared the empirical NTK margin of two-layer KAN with that of a parameter matched two-layer ReLU MLP on the same data, see Table 1. We find that the KAN margin is positive, increases with width, and is consistently larger than that of the ReLU MLP. While this is not a complete characterization, it provides further support that NTK separability assumption is meaningful and plausible for structured data aligned with the KAN architecture.  We will add the details of these additional experiments in the revised version.
>
> | Dataset | Model | m = 8 | m = 64 | m = 512 |
> |---|---|---:|---:|---:|
> | Multi-interval 1D | ReLU | 0.0151 | 0.0191 | 0.0197 |
> | Multi-interval 1D | KAN | **0.0400** | **0.0883** | **0.1027** |
> | XOR-gap | ReLU | 0.0226 | 0.0285 | 0.0284 |
> | XOR-gap | KAN | **0.0243** | **0.0503** | **0.0599** |
> | Checkerboard-gap | ReLU | 0.0060 | 0.0104 | 0.0111 |
> | Checkerboard-gap | KAN | **0.0206** | **0.0451** | **0.0538** |
>
> **Table 1.** Empirical NTK margin on synthetic nonlinear datasets for two-layer ReLU and KAN at different widths.
>
>
> **Q2.  Are there any empirical gap between theoretical critical widths and practically optimal widths?**
>
> We thank the reviewer for this important question. A numerical gap between the theoretical critical width and the empirically best width can certainly occur, and we view this as natural. Our theory is intended to characterize the correct regime, namely that a moderate or polylogarithmic width is sufficient and that performance saturates beyond a critical range, rather than to predict the exact best width for each dataset. The experiments support precisely these qualitative trends. Thus, although the exact empirical optimum may differ because of hidden constants and dataset-specific effects, the observed behavior remains consistent with the theoretical prediction at the level of scaling and regime selection.

---

> > ### Author Rebuttal · Reviewer_bMLB · 2026-04-03
> >
> > Thanks for the clarifications. My questions are addressed and I will maintain the positive rating.

---

### Decision · Program_Chairs · 2026-04-30

**Decision:**

Accept (regular)

**Comment:**

This paper studies DP optimization for non-convex problems and provides new bounds for GD and DP-GD for training two-layer Kolmogorov-Arnold Networks. Under the NTK separability assumption, the paper shows that poly logarithmic width is sufficient to prove certain convergence rates for GD and DP-GD. The authors should revisit the reviewers' comments, especially regarding the strength of their NTK assumption, and provide certain examples\motivations in practice that show the feasibility of these assumptions.

Overall, the paper has nice contributions and I recommend accepting it.